# RECQ4-MUS81 interaction contributes to telomere maintenance with implications to Rothmund-Thomson syndrome

Raghib Ashraf[1,2], Hana Polasek-Sedlackova [1,3], Victoria Marini [2], Jana Prochazkova[2], Zdenka Hasanova[2], Magdalena Zacpalova [2], Michala Boudova[2] & Lumir Krejci [1,2] ✉

Replication stress, particularly in hard-to-replicate regions such as telomeres and centromeres, leads to the accumulation of replication intermediates that must be processed to ensure proper chromosome segregation. In this study, we identify a critical role for the interaction between RECQ4 and MUS81 in managing such stress. We show that RECQ4 physically interacts with MUS81, targeting it to specific DNA substrates and enhancing its endonuclease activity. Loss of this interaction, results in significant chromosomal segregation defects, including the accumulation of micronuclei, anaphase bridges, and ultrafine bridges (UFBs). Our data further demonstrate that the RECQ4-MUS81 interaction plays an important role in ALT-positive cells, where MUS81 foci primarily colocalise with telomeres, highlighting its role in telomere maintenance. We also observe that a mutation associated with Rothmund-Thomson syndrome, which produces a truncated RECQ4 unable to interact with MUS81, recapitulates these chromosome instability phenotypes. This underscores the importance of RECQ4-MUS81 in safeguarding genome integrity and suggests potential implications for human disease. Our findings demonstrate the RECQ4-MUS81 interaction as a key mechanism in alleviating replication stress at hard-to-replicate regions and highlight its relevance in pathological conditions such as RTS.

Cell division is a fundamental process for growth and maintenance in all living organisms, driven by the precise duplication of the genome. However, various intrinsic and extrinsic challenges can disrupt the replication progression, leading to a phenomenon known as replication stress[1]. Such stress can arise from regions of the DNA that are inherently difficult to replicate, including long genes, sparse replication origins, GC-rich repetitive sequences, G4 DNA, and other non-B-form DNA structures. These hard-to-replicate regions, often including telomeres, centromeres, and common fragile sites, can cause replication fork stalling, particularly under conditions of increased stress,

such as treatment with aphidicoline[2]. Additionally, replication stress can be triggered by the depletion of genes involved in DNA metabolism, such as DNA2[3] or 53BP1[4], or by the overexpression of oncogenes, like c-Myc[5], RAS[6], and BRAF[7]. These disruptions can lead to stalling of replication forks, resulting in the accumulation of replication intermediates that require processing to ensure proper chromosome segregation. Failure to process these intermediates can manifest as ultrafine and anaphase bridges between chromosomes during cell division, chromosome missegregation, and ultimately leading to chromosome breakage and the formation of micronuclei[8].

[1]National Centre for Biomolecular Research, Faculty of Science, Masaryk University, Kamenice 5/A4, 625 00 Brno, Czech Republic. [2]Department of Biology, Faculty of Medicine, Masaryk University, Kamenice 5/A7, Brno 62500, Czech Republic. [3]Institute of Biophysics, Czech Academy of Sciences, Kralovopolska 135, Brno 61200, Czech Republic. ✉e-mail: lkrejci@chemi.muni.cz

The effective resolution of these persistent replication intermediates, including branched DNA intermediates, Holliday junctions (HJs), and other recombination intermediates, is primarily mediated by the SLX1-SLX4-MUS81-EME1 (SLX-MUS complex), MUS81-EME2 complex, BLM-Topoisomerase IIIα-RMI1-RMI2 (BTR complex), GEN1[9] and DNA2[3]. Among these, the structure-specific endonuclease MUS81 is a key player in promoting faithful sister chromatid segregation during mitosis. MUS81 operates in distinct phases of the cell cycle, during S phase, MUS81 forms a complex with EME2, crucial for fork cleavage and telomere maintenance in cells dependent on the alternative lengthening of telomeres (ALT) mechanism[10]. In the G2/M phase, MUS81 operates in a complex with EME1 to cleave the branched DNA structures[10]. Previous studies have demonstrated that MUS81 frequently collaborates with helicases or translocases to effectively overcome replication stress[11,12]. Among these, the RECQ family of helicases, including WRN[13], BLM[14], RECQ1[15], and recently RECQ5[16], has been shown to be associated with MUS81 in maintaining genome integrity. Notably, RECQ4, another member of the RECQ family, has been linked with MUS81 in *Arabidopsis thaliana* for processing recombination-induced replication intermediates[17].

RECQ4 is highly conserved and essential to cells as knockout mice are embryonically lethal[18]. Besides the RECQ helicase domain, RECQ4 contains a specific N-terminal region that shares sequence homology with yeast Sld2, MCM10 interaction site, and binding domain to various branched DNA substrates[19]. It plays an important role in replication, recombination, and DNA repair and is a major player in conserving genome integrity[19,20]. Mutations in RECQ4 have been identified as a causative agent for three autosomal recessive diseases, Rothmund-Thomson (RTS), Baller-Gerold, and RAPADILINO syndromes characterised by growth deficiency, poikiloderma, and higher predisposition to osteosarcomas along with greater malignancy rate[21,22].

DNA intermediates generated by frequent replication fork pausing or stalling at hard-to-replicate genomic regions pose a threat to genome stability, a hallmark associated with the development of cancer and developmental diseases[23]. However, the mechanism responsible for generating and resolving intrinsic intermediates arising during the duplication of hard-to-replicate regions needs further investigation. In this study, we aim to elucidate how RECQ4 and MUS81 coordinate with each other to resolve replication intermediates and maintain the integrity of the human genome.

## Results

### RECQ4 stimulates MUS81 endonuclease activity

Since previous studies, including ours, reported that MUS81 nuclease activity can be stimulated by BLM[14] and RECQ5[16], we wished to compare the ability of various RECQ family members to promote cleavage of the 3′-flap DNA substrate representing typical branched DNA substrates resolved by MUS81[14,16]. To ensure consistency, we first confirmed the functional activities of BLM, RECQ1, and RECQ5 (Supplementary Fig. 1A–D). We found that RECQ4 showed the highest stimulation of MUS81-EME1, with >10-fold stimulation upon incubation with 25 nM protein concentration (Fig. 1A, B). We also observed the RECQ4-dependent stimulation of MUS81 nuclease in complex with EME2 (Supplementary Fig. 1E, F), suggesting the requirement of MUS81 protein in mediating this stimulation.

To test whether this stimulation requires direct protein interaction of RECQ4 with MUS81, we performed an in vitro pull-down of various MBP-tagged RECQ4 fragments (Supplementary Fig. 1G, H) incubated with the GST-MUS81-EME1 complex. We found that while the N-terminus (1-322) containing sequence homology with Sld2 and C-terminus (455-1208) consisting of helicase and RQC domains failed to interact with MUS81-EME1, fragments (1-492) and (1-400) were successfully retained on the GST beads (Supplementary Fig. 1I). These findings indicate that direct physical interaction between the MUS81-

EME1 complex and RECQ4 requires the N-terminal region of RECQ4 (322-400). Using a pull-down between RECQ4 and various GST-MUS81 fragments, we also determined that the interaction is mediated through the MUS81 subunit (Supplementary Fig. 1J) specifically through N-terminus of MUS81 (1-278) (Supplementary Fig. 1K).

To map this interaction at a single amino acid level, we performed multiple sequence alignment (Supplementary Fig. 1L) of the mapped MUS81 interaction region (322-400) within RECQ4 homologues and identified a highly conserved region spanning amino acids 353-355 (Supplementary Fig. 1M). To validate the importance of this domain for the interaction with MUS81, we generated a RECQ4 fragment (1-400) Δ3 by deleting three highly conserved amino acids and performed in vitro pull down. This assay revealed that fragment (1-400) Δ3 lost the ability to interact with the MUS81-EME1 complex (Fig. 1C).

Importantly, the MUS81 interaction site mapping correlates with the ability to stimulate MUS81-EME1 nuclease activity by RECQ4 (Fig. 1D, E). While interaction-proficient RECQ4 truncation (1-400 aa) was able to stimulate MUS81 nuclease activity to a similar level as full-length RECQ4, non-interacting truncation (1-322) along with (455-1208) only slightly promoted MUS81-EME1 cleavage (Fig. 1D, E). Similarly, RECQ4 fragment (1-400) Δ3 could not stimulate endonuclease activity (Fig. 1D, E). Of note, our previous work showed that the RECQ4 region, spanning residues 322-400, contains a branched DNA binding domain, however, deletion of 353-355 region did not affect DNA binding abilities (Supplementary Fig. 1N). Taken together, these findings indicate that stimulation of MUS81 activity requires functional protein interaction.

Next, given the RECQ4 ability to bind branched DNA substrates[20], we wished to elucidate whether RECQ4 DNA binding plays a role in stimulating MUS81-EME1 endonuclease activity by targeting MUS81 to such DNA substrate. We found that preincubating RECQ4 with DNA substrate before adding MUS81-EME1 resulted in almost 2-fold more product formation than when RECQ4 was simultaneously incubated with nuclease and DNA substrate (Fig. 1F and Supplementary Fig. 2A), confirming the ability of RECQ4 to target MUS81 to DNA substrate. Our findings suggest that RECQ4 interacts with MUS81 complexes, targets them to DNA substrate, and stimulates their nuclease activity.

### RECQ4 interacts with MUS81 ex vivo and safeguards proper chromosome segregation

To investigate the functional interplay between RECQ4 and MUS81 endonuclease in a cellular context, we established stable cell lines allowing simultaneous depletion of endogenous RECQ4 and the expression of siRNA-resistant exogenous EGFP-tagged RECQ4-WT, RECQ4-Δ3, and EGFP in Flp-In T-REx U2OS cells (Supplementary Fig. 2B, C). Immunoprecipitation assay confirmed that the minimal RECQ4 region required for MUS81 interaction ex vivo was located between amino acid residues 353-355 (Fig. 2A, B and Supplementary Fig. 2D). To further validate the functional specificity of this interaction, we also probed for RPA32, another RECQ4 interaction partner[24]. Notably, the deletion of these critical amino acids did not affect the binding to RPA32 (Fig. 2A), highlighting the selective nature of the RECQ4-MUS81 interaction. Additionally, substituting these residues with alanine resulted in a marked reduction in the interaction with MUS81 (Supplementary Fig. 2E, F), reinforcing the importance of this specific region. To determine the cell cycle stage at which the RECQ4-MUS81 interaction takes place, we synchronised cells at different phases and performed RECQ4 immunoprecipitation using anti-GFP beads. Our results revealed that the interaction between RECQ4 and MUS81 was predominantly observed at the G1 and S phases, with partial interaction observed in the G2/M phase (Fig. 2C and Supplementary Fig. 2G, H). Given the previously observed problems with chromosome segregation upon MUS81 depletion[10], we tested the effect of RECQ4 depletion. Monitoring cell transition from metaphase to anaphase, we observed a delay in mitotic progression and visible

chromosome misalignments upon RECQ4 depletion, consistent with a previous report[25] (Supplementary Fig. 3A, B). The expression of RECQ4-WT rescued this phenotype in contrast to the interaction-deficient mutant. These observations indicate that unresolved DNA intermediates may impair metaphase alignment and thus delay the anaphase entry. Further monitoring of downstream chromosome segregation events revealed a significant accumulation of micronuclei and bulky bridges upon RECQ4 depletion, similar to the effect observed with MUS81 depletion. While expression of RECQ4-WT rescued these phenotypes, RECQ4-Δ3 failed to do so (Fig. 2D, E and Supplementary Fig. 3C). Interestingly, co-depletion of RECQ4 and MUS81 did not result in an additive effect, indicating that they both might function in the same pathway. Our data thus suggest that RECQ4 and its interaction with MUS81 are required for proper chromosome segregation.

## Loss of RECQ4-MUS81 interaction leads to increased formation of ultrafine bridges

The observed segregation issues in RECQ4-depleted cells prompted us to measure the level of ultrafine bridges (UFBs) representing late replication or recombination intermediates. We employed specific staining techniques to identify their origin[26]. Immunofluorescence and FISH were used to monitor UFBs at fragile sites (FUB), centromeres (CUB), and telomeres (TUB), using anti-FANCD2, anti-CEN staining, and PNA-FISH, respectively (Fig. 3A, B). Depletion of RECQ4 resulted in more than 2-fold accumulation of both FUB and CUB, comparable to the accumulation observed with MUS81 or MUS81/RECQ4 co-depletion (Fig. 3C, D and Supplementary Fig. 3D–G). Interestingly, TUBs showed a more than 5-fold increase, suggesting a more critical requirement of RECQ4 in telomere maintenance (Fig. 3E and Supplementary Fig. 3H, I). Given that U2OS cells possess an active alternative lengthening of telomere (ALT) mechanism, the resolution of DNA structures arising from replication stress at telomeres appears to rely on the interaction between RECQ4 and MUS81. To further explore this, we also compared UFBs in the ALT-negative HT1080 cell line and observed a statistically significant increase in both CUBs and TUBs (Fig. 3F, G and Supplementary Fig. 3J, K). However, ALT-positive U2OS cells exhibited a higher fold change in TUBs compared to CUBs and FUBs in HT1080 (Supplementary Fig. 3D, E and Supplementary Fig. 3I–K), indicating comparable intrinsic stress at these genomic loci and pointing towards the higher requirement for MUS81-RECQ4 interaction at ALT telomeres. Importantly, expression of RECQ4-WT was able to mitigate the increase in all types of UFBs, in contrast to the interaction-deficient mutants RECQ4-Δ3 and RECQ4-AAA that

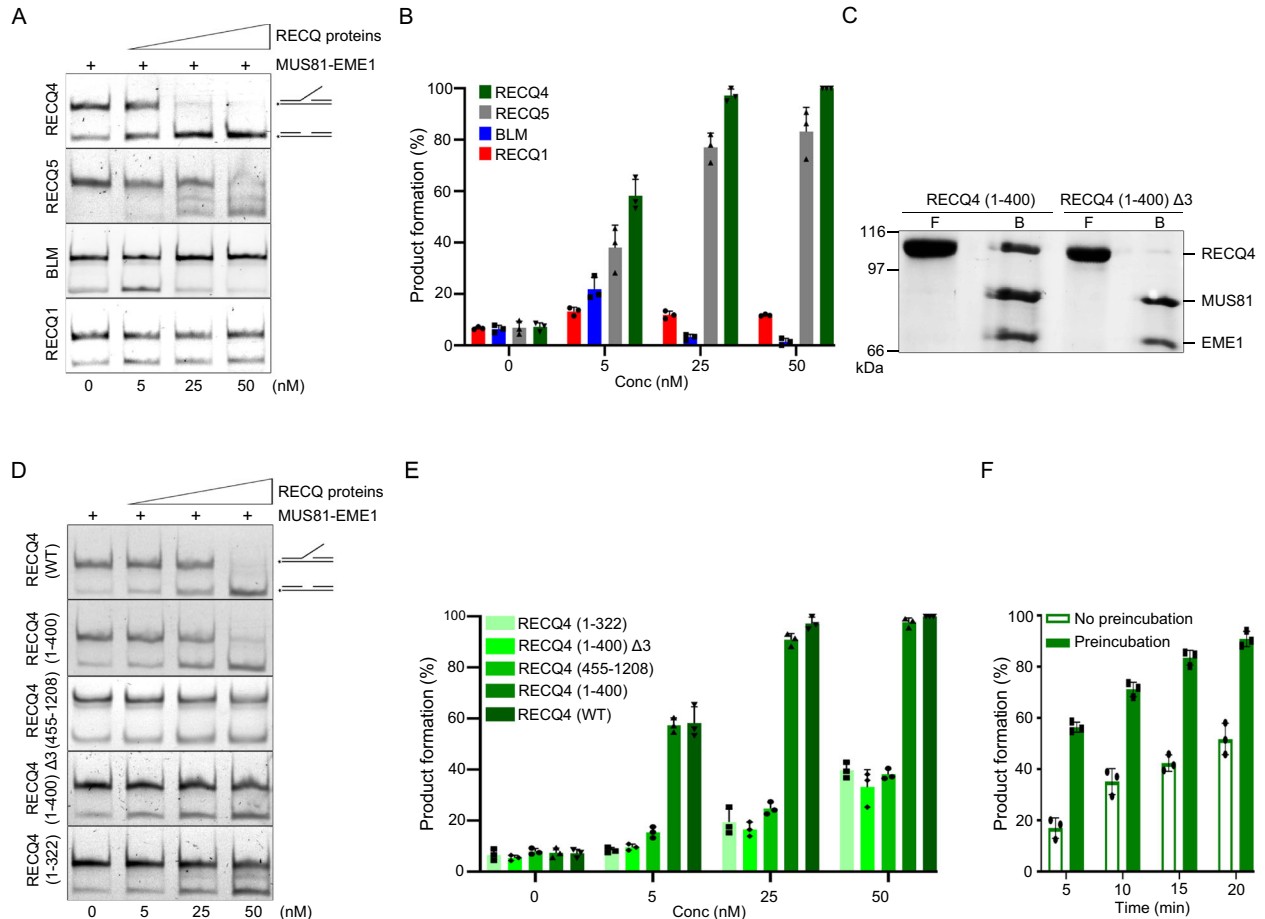

**Fig. 1 | RECQ4 interacts and stimulates MUS81-EME1 endonuclease activity.**
**A** Increasing concentrations of individual RECQ helicases (RECQ4, RECQ5, BLM, and RECQ1) were incubated with 3′-flap DNA substrate (6 nM) in the presence of MUS81-EME1 (0.5 nM) for 20 min. The reaction mixtures were resolved on a native PAGE gel. **B** Quantification of data in (A); $n = 3$ independent experiments; data are means ± SD. (**C**) Interaction of purified MBP-RECQ4 (1-400) and MBP-RECQ4 (1-400) Δ3 with GST-MUS81-EME1. The flow (F) and bound (B) fractions were analysed by SDS-PAGE, followed by Coomassie blue staining. **D** MUS81-EME1 (0.5 nM) and increasing amounts of RECQ4 (WT), RECQ4 (1-322), RECQ4 (1-400), RECQ4 (1-400) Δ3, and RECQ4 (455-1208) were incubated with 3′flap DNA substrate (6 nM) for 20 min at 37 °C before analysis by native gel electrophoresis. **E** Quantification of data in (D); $n = 3$ independent experiments; data are means ± SD. **F** RECQ4 (1-400) (25 nM) was either preincubated (Preincubation) with 3′flap DNA substrate (6 nM) followed by the addition of MUS81-EME1 (0.5 nM) or mixed with DNA and MUS81-EME1 without any preincubation (No preincubation). Reactions were incubated for the indicated time at 37 °C prior to analysis by native gel electrophoresis. $n = 3$ independent experiments; data are means ± SD. Source data are provided as a Source data file.

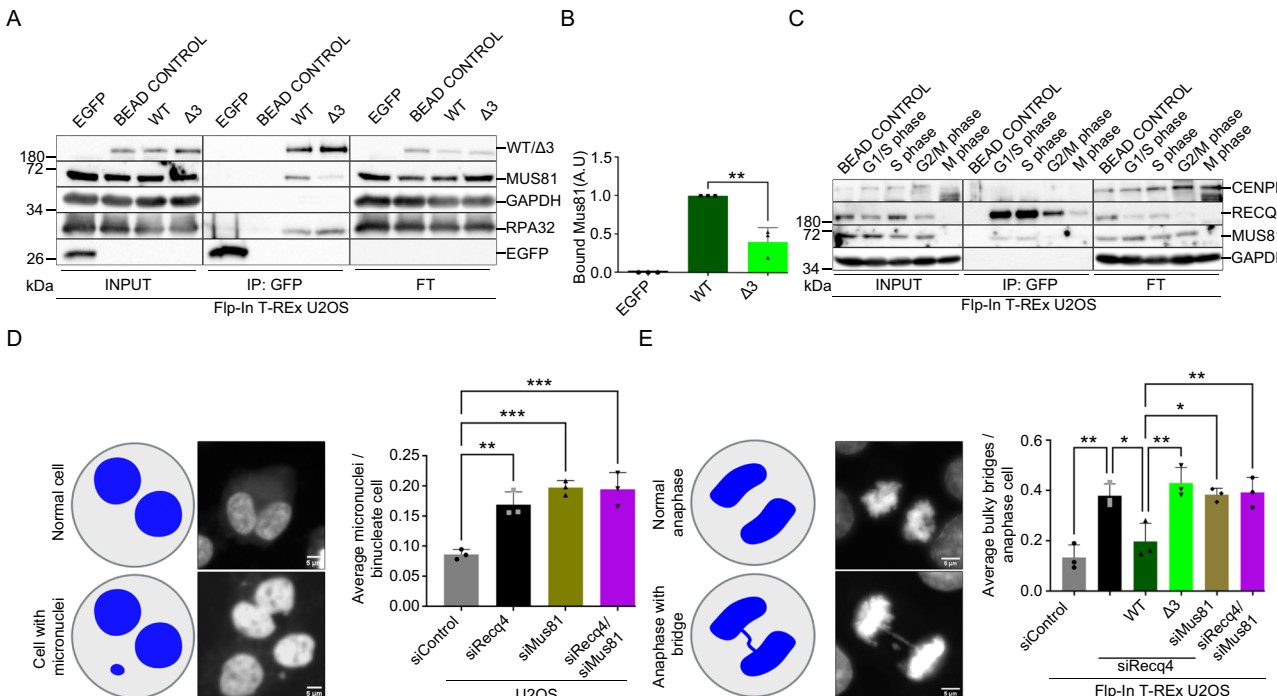

**Fig. 2 | RECQ4 interacts and is required for MUS81 function in cells.**
**A** Immunoprecipitation (IP) was done from whole cell extracts expressing EGFP, EGFP-RECQ4 -WT, and the mutant EGFP-REQ4-Δ3. Proteins from each cell extract (500 μg) were incubated with GFP beads for 1 h and then resolved on SDS-PAGE, followed by western blotting detecting corresponding proteins. Input (IP), and FT fractions were run on different gels respectively. **B** Quantification of MUS81 binding from data represented in (A); $n = 3$ independent experiments; data are means ± SD; one-way ANOVA followed by Tukey's multiple comparison test; $p$-value were adjusted by GraphPad Prism software; WT vs Δ3 **$p = 0.0011$. **C** Cell cycle-specific interaction between RECQ4 and MUS81. IP was done from whole cell extracts expressing EGFP-RECQ4-WT synchronised in different cell cycle phases. Protein from each extract (500 μg) was incubated with GFP beads for 1 h and then resolved on SDS-PAGE, followed by western blotting detecting corresponding proteins. **D** Graphical representation and representative images of binucleated cells by immunofluorescence with DAPI. Created in BioRender. Ashraf, R. (2025) https://

BioRender.com/n17i536. Quantification of the average number of micronuclei per binucleated cell for each treatment; $n = 3$ independent experiments; data are means ± SD; one-way ANOVA followed by Tukey's multiple comparison test; $p$-value were adjusted by GraphPad Prism software; siControl vs siRECQ4 **$p = 0.0018$; siControl vs siMUS81 ***$p = 0.0003$; siControl vs siMUS81/siRecq4 ***$p = 0.0003$. **E** Representative images and a graphical depiction of anaphase cells with bulky bridges. Created in BioRender. Ashraf, R. (2025) https://BioRender.com/n17i536. Quantification of the average number of bulky bridges per anaphase cell (DAPI positive bridges); $n = 3$ independent experiments; data are means ± SD; one-way ANOVA followed by Tukey's multiple comparison test; $p$-value were adjusted by GraphPad Prism software; siControl vs siRECQ4 **$p = 0.0015$; siRecq4 vs WT/siRecq4 *$p = 0.0155$; WT/siRecq4 vs Δ3/siRECQ4 **$p = 0.0025$; WT/siRecq4 vs siMus81 *$p = 0.0135$; WT/siRecq4 vs siRecq4/siMus81 **$p = 0.0094$. Source data are provided as a Source data file.

resembled RECQ4-depleted cells (Fig. 3C–E and Supplementary Fig. 3L). Our data thus indicate an essential role for RECQ4 and its interaction with MUS81 in processing late replication/recombination intermediates, ultimately minimizing UFB formation in mitosis, particularly at telomeres maintained using the ALT mechanism.

## ALT-dependent MUS81 foci require RECQ4 and colocalise with telomeres

The increased requirement for RECQ4 interaction with MUS81 in telomere maintenance in ALT cell lines, together with a previously described G2-specific MUS81 foci formation, prompted us to examine the importance of RECQ4-MUS81 interaction in MUS81 foci formation[27]. To validate MUS81 foci, we first performed siRNA depletion of MUS81 (Supplementary Fig. 4A). We then monitored MUS81 foci occurrence in a panel of both ALT-positive and ALT-negative cell lines. Notably, MUS81 foci were readily detected in all ALT-positive cells examined, whereas the majority of ALT-negative cells failed to form these foci (Fig. 4A and Supplementary Fig. 4B, C), despite comparable endogenous MUS81 protein levels confirmed by western blot (Supplementary Fig. 4D). Next, we investigated the effect of RECQ4 depletion on MUS81 foci formation in three ALT cell lines (U2OS, LM216J, and Saos-2). Using immunofluorescence, we found that RECQ4 knockdown significantly decreased MUS81 foci formation in U2OS and LM216J cell lines, but no significant effect was observed in

Saos-2 cell line (Fig. 4B–D and Supplementary Fig. 4E), possibly reflecting their intermediate ALT type[28]. To further test the effect of RECQ4-MUS81 interaction on MUS81 foci formation, we used quantitative image-based cytometry (QIBC) to assess MUS81 foci in our stable Flp-In T-REx U2OS cell lines. Depletion of endogenous RECQ4 in these cells led also to a significant decrease in MUS81 foci, which was rescued by the expression of RECQ4-WT but not by expression of RECQ4-Δ3 (Fig. 4E and Supplementary Fig. 4F, G), supporting the role of RECQ4 in MUS81 foci formation. We extended our study to the LM216J cell line by transiently transfecting plasmids expressing EGFP, EGFP-RECQ4-WT, and EGFP-RECQ4-Δ3. These experiments revealed similar results to those observed in Flp-In T-REx U2OS cells (Fig. 4F and Supplementary Fig. 4H).

Given the observed higher fold increase in telomeric ultrafine bridges (TUBs) upon RECQ4 depletion (Supplementary Fig. 3I), we investigated whether MUS81 foci formation is specific to telomeric regions compared to other genomic loci. To address this, we measured MUS81 foci formation and its colocalization with telomeres and centromeres in two ALT-positive cell lines (U2OS and LM216J) with endogenous levels of RECQ4. Notably, there was a strong colocalization of MUS81 foci with telomeric regions, which was almost 5-fold higher than that of MUS81 with centromeres (Fig. 4G–I). These findings are aligned with our in vitro experiments (Fig. 1F and Supplementary Fig. 1H) and indicate that RECQ4 may recognize DNA structures at

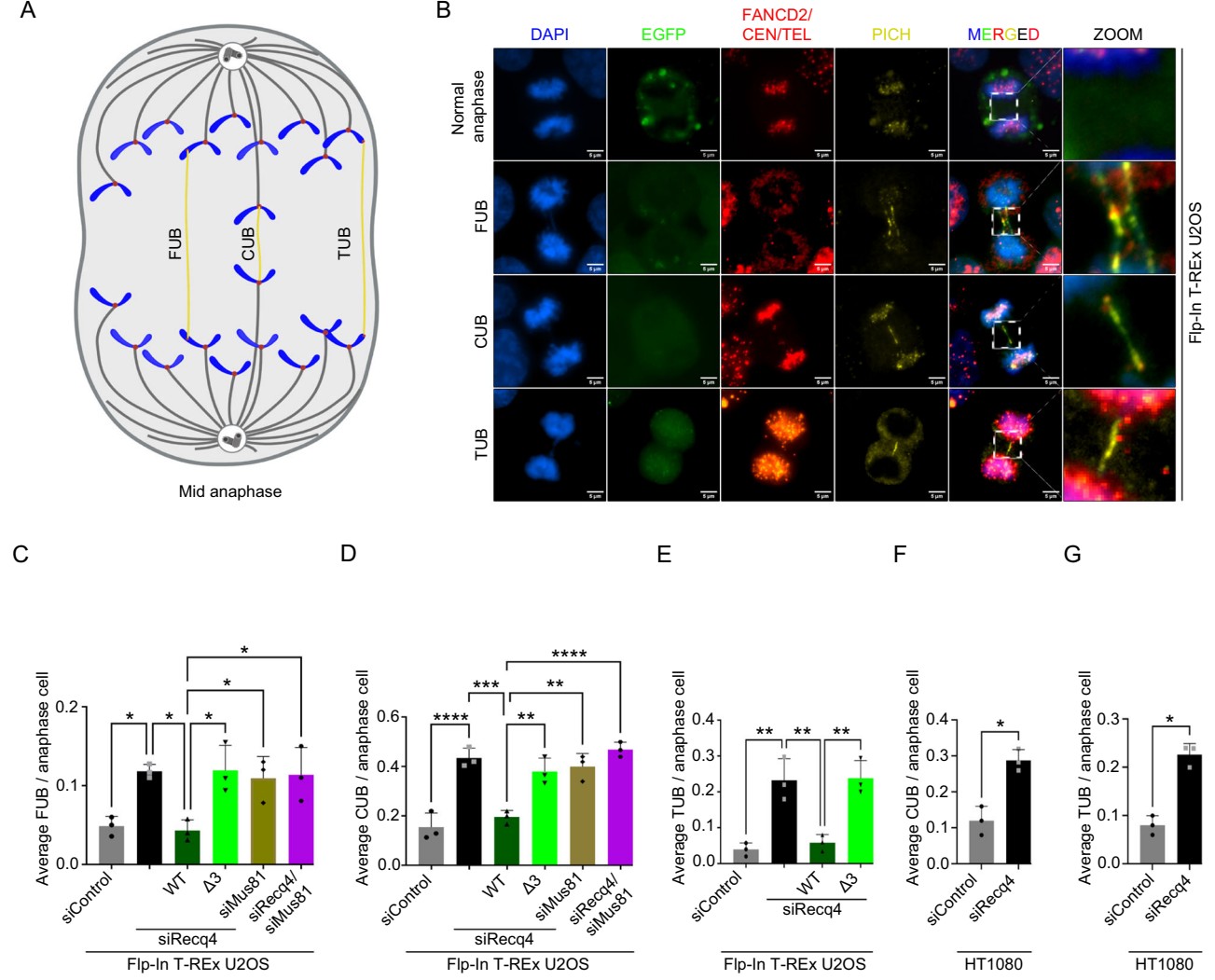

**Fig. 3 | RECQ4 cooperates with MUS81 to prevent ultrafine bridges formation.**
**A** Graphical depiction of ultrafine bridges. Created in BioRender. Ashraf, R. (2025)
https://BioRender.com/o36w298. **B** Representative immunofluorescence images
of UFBs stained by PICH, fragile sites visualised by FANCD2 staining, centromere
staining by Anti centromere antibodies (CEN), and telomere staining with telomere
PNA probe (TEL) in Flp-In T-REx U2OS cells. Scale bar = 5 μm. **C–E** Quantification of
the average number of UFBs per anaphase cell with different types of UFBs for each
treatment. (C) Fragile sites ultrafine bridges, FUB; $n = 3$ independent experiments;
data are means ± SD; one-way ANOVA followed by Tukey's multiple comparison
test; $p$-value were adjusted by GraphPad Prism software; siControl vs siRECQ4
*$p = 0.0343$; siRecq4 vs WT/siRecq4 *$p = 0.0211$; WT/siRecq4 vs Δ3/siRECQ4
*$p = 0.0191$; WT/siRecq4 vs siMus81 *$p = 0.0449$; WT/siRecq4 vs siRecq4/siMus81
*$p = 0.0312$. (D) Centromeric ultrafine bridges (CUB); $n = 3$ independent experi-
ments; data are means ± SD; one-way ANOVA followed by Tukey's multiple com-
parison test; $p$-value were adjusted by GraphPad Prism software; siControl vs

siRECQ4 ****$p < 0.0001$; siRecq4 vs WT/siRecq4 ***$p = 0.0003$; WT/siRecq4 vs Δ3/
siRECQ4 **$p = 0.0034$; WT/siRecq4 vs siMus81 **$p = 0.0014$; WT/siRecq4 vs siR-
ecq4/siMus81 ****$p < 0.0001$. (E) Telomeric ultrafine bridges (TUB); $n = 3$ indepen-
dent experiments; data are means ± SD; one-way ANOVA followed by Tukey's
multiple comparison test; $p$-value were adjusted by GraphPad Prism software;
siControl vs siRECQ4 **$p = 0.0020$; siRecq4 vs WT/siRecq4 **$p = 0.0039$; WT/
siRecq4 vs Δ3/siRECQ4 **$p = 0.0032$. Depletion of endogenous RECQ4 and
expression of RECQ4 constructs with doxycycline was done for 48 h.
**F**, **G** Quantification of the average number of UFBs per anaphase cell in wild-type
and RECQ4-depleted HT1080 (ALT-negative) cell line. **F** Centromeric ultrafine
bridges (CUB); $n = 3$ independent experiments; data are means ± SD;
Mann–Whitney test; one-tailed; *$p = 0.0500$. **G** Telomeric ultrafine bridges (TUB);
$n = 3$ independent experiments; data are means ± SD; Mann–Whitney test; one-
tailed; *$p = 0.0500$. Source data are provided as source data file.

hard-to-replicate genomic regions and target MUS81 to these loci to
resolve replication or recombination intermediates.

Moreover, we also assessed the effects RECQ4 depletion and its
complementation with either RECQ4-WT or RECQ4-Δ3 on the occur-
rence of ALT-associated promyelocytic leukaemia bodies (APBs).
Depletion of endogenous RECQ4 levels led to a marked decrease in
APBs foci (Fig. 4J, K). This decrease was reversed by the expression of
RECQ4-WT, while expression of RECQ4-Δ3 failed to restore APB foci
(Fig. 4J, K). Taken together, our data indicate that RECQ4-MUS81
interaction might be important for MUS81 foci formation at telomeric

regions in ALT-positive cells, reflecting possible increased endogenous
replication stress at telomeres in these cells.

## RECQ4/MUS81 pathway is distinct but coordinated with other resolution pathways

Our previous work demonstrated that RECQ4 selectively recognizes
branched DNA substrates, particularly HJs[20]. Considering that various
mechanisms have been described for HJ dissolution/resolution, we
aimed to investigate the potential interplay between MUS81-RECQ4
and the BLM and GEN1-mediated dissolution/resolution pathways[29]. To

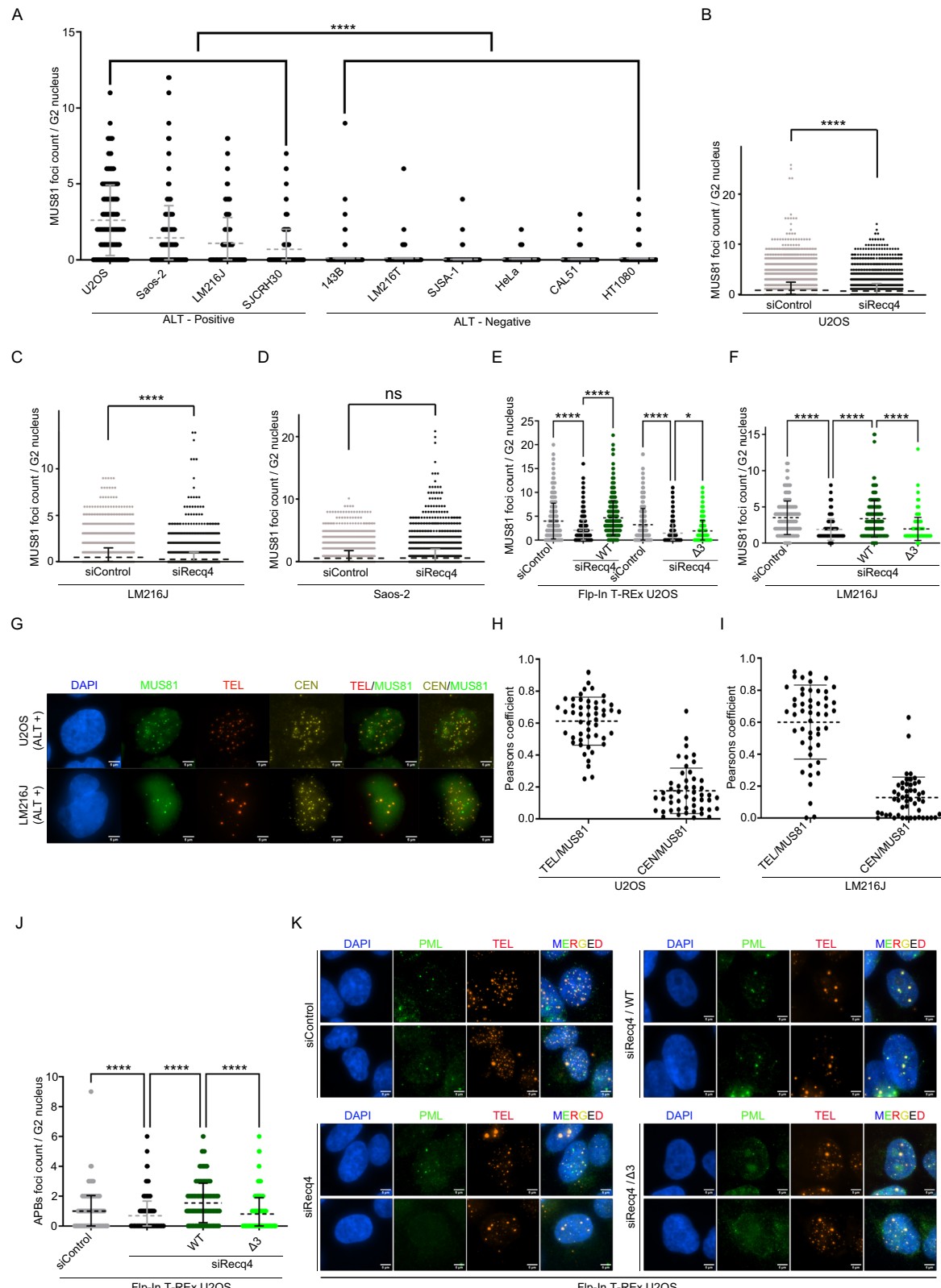

explore this, we depleted RECQ4 in GEN1 knockout cells[30] and observed an additive effect on the accumulation of bulky bridges (Fig. 5A, B). Increased accumulation of unresolved bulky bridges ultimately might be affecting the progression of cells through the cell cycle, particularly mitosis. Next, we co-depleted RECQ4 and BLM in U2OS cells and assessed UFB. We found that the percentage of cells having at least one ultrafine bridge remains unchanged (Fig. 5C and

Supplementary Fig. 4I). However, we observed an increase in the percentage of cells having multiple ultrafine bridges (more than one UFB) (Fig. 5C, D and Supplementary Fig. 4J). These results suggest that, in addition to direct role of BLM in resolution, RECQ4 contributes to the processing of replication intermediates to prevent the formation of UFBs. Our observations thus highlight the importance of RECQ4, GEN1, and BLM in maintaining genomic stability by efficiently resolving

**Fig. 4 | ALT-dependent MUS81 foci require RECQ4 and colocalise with telomeres. A** Quantification of MUS81 foci from IF analysis in a panel of ALT-positive and ALT-negative cell lines. $n$ = minimum 900 cells; data are means ± SD; one-way ANOVA followed by Tukey's multiple comparison test; $p$-value were adjusted by GraphPad Prism software; **** $p < 0.0001$. **B–D** IF analysis of MUS81 foci with indicated siRNA for 48 h. $n$ = minimum 5000 cells; data are means ± SD; Mann Whitney test; (B) U2OS; **** $p < 0.0001$. (C) LM216J; **** $p < 0.0001$. (D) Saos-2; ns $p = 0.13$. **E** QIBC analysis of MUS81 foci in Flp-In T-REx U2OS cells expressing either EGFP-REQ4-WT or EGFP-RECQ4-Δ3, with indicated siRNA (48 h). $n$ = minimum 5000 cells; data are means ± SD; one-way ANOVA followed by Tukey's multiple comparison test; $p$-value were adjusted by GraphPad Prism software; WT set: siControl vs siRECQ4 **** $p < 0.0001$; siRecq4 vs WT/siRecq4 **** $p < 0.0001$; Δ3 set: siControl vs siRECQ4 **** $p < 0.0001$; siRecq4 vs Δ3/siRECQ4 * $p = 0.0268$. **F** Quantification of MUS81 foci in LM216J cells transiently transfected with indicated constructs and treated with the specified siRNA for 48 h. $n$ = minimum 200 cells; data are means ± SD; one-way ANOVA followed by Tukey's multiple comparison test; $p$-value were adjusted by GraphPad Prism software; siControl vs siRECQ4 **** $p < 0.0001$; siRecq4 vs WT/siRecq4 **** $p < 0.0001$; WT/siRecq4 vs Δ3/siRECQ4 **** $p < 0.0001$. **G** Representative IF images of MUS81 foci and their colocalization with centromere (CEN) and telomere (TEL) in U2OS and LM216J cells. Scale bar = 5 μm. **H–I** Quantification of data in (G). $n$ = 50 cells in each cell line; data are means ± SD. **J** Quantification of (APBs) foci in Flp-In T-REx U2OS cells expressing either EGFP-REQ4-WT or EGFP-RECQ4-Δ3, with indicated siRNA (48 h). $n$ = minimum 500 cells; data are means ± SD; one-way ANOVA followed by Tukey's multiple comparison test; $p$-value were adjusted by GraphPad Prism software; siControl vs siRECQ4 **** $p < 0.0001$; siRecq4 vs WT/siRecq4 **** $p < 0.0001$; WT/siRecq4 vs Δ3/siRECQ4 **** $p < 0.0001$. **K** Representative IF images of APBs foci. Source data are provided as a Source data file.

and preventing the accumulation of replication/recombination intermediates, suggesting that these proteins play distinct but coordinated roles in their management.

## A patient-derived RECQ4 mutation phenocopies an interaction deficient mutant

Given the clinical relevance of RECQ4, we next investigated whether RECQ4-MUS81 interaction is also compromised in patient-derived mutations and could be one of the reasons underlying the disease. From known RECQ4 mutations linked to RTS near the identified MUS81 interaction region[31] (Supplementary Fig. 5A), we acquired fibroblasts from clinically affected son (RTS-CA, AG18371) and clinically unaffected mother (RTS-CU, AG18373) expressing a RECQ4 truncation. The RTS-CA fibroblast carries an 11 base pair deletion in intron 8, resulting in a premature stop codon generating truncated RECQ4 protein of 501 amino acids with mutated last six amino acids[32] (Supplementary Fig. 1M and Supplementary Fig. 5A). We confirmed previously reported reduced levels of full-length RECQ4 in RTS-CA fibroblasts[25], and to some extent in RTS-CU, compared to normal fibroblasts (Supplementary Fig. 5B). Further characterisation revealed a significant accumulation of average micronuclei and bulky bridges per anaphase in RTS-CA compared to normal fibroblasts (Supplementary Fig. 5C, D), along with a robust accumulation of spontaneous FUBs, CUBs, and TUBs, the CUBs and TUBs being more prominent with almost double the amount compared to FUBs (Fig. 6A-C). Despite carrying a mutation in one allele, the clinically unaffected RTS-CU fibroblast behaved similarly to normal fibroblast (Fig. 6A-C and Supplementary Fig. 5C, D), indicating that one wild-type allele is sufficient to prevent disease development, further confirming the recessive nature of RTS. These observations were supported by data showing that the presence of endogenous wild-type RECQ4 effectively compensates for the segregation defects observed in cells expressing the RECQ4-Δ3 mutant, specifically in terms of the average number of bulky bridges and UFBs (Supplementary Fig. 5E, F). To directly assess the RECQ4 and MUS81 interaction status in this clinically relevant setting, we created a stable Flp-In T-REx U2OS cell line carrying EGFP-tagged RTS-CA mutation (RECQ4-RTS-CA) in the above-described inducible system (Supplementary Fig. 1M, Supplementary Fig. 2B, C). Compared to RECQ4-WT, immunoprecipitation of RTS-CA significantly reduced the levels of MUS81 on GFP beads (Fig. 6D), indicating a defect in interaction. In addition, purified RECQ4 (RTS-CA) protein showed a partial defect in the stimulation of MUS81-EME1 endonuclease activity in vitro (Supplementary Fig. 5G, H). We also characterised levels of ultrafine bridges in Flp-In T-REx U2OS RTS-CA cells and observed a significant accumulation of all ultrafine bridges (Fig. 6E-G), similar to the RECQ4 knockdown. Additionally, the expression of RTS-CA could not complement the decrease of MUS81 foci formation upon RECQ4 depletion and was comparable to RECQ4-Δ3 mutant (Fig. 6H, Supplementary Fig. 4F, and Supplementary Fig. 5I), further supporting the defect in MUS81 interaction. Moreover, the RECQ4 mutation derived from RTS patient retains the MUS81 interaction motif identified in our study, the loss of MUS81 interaction could reflect large structural changes observed in RECQ4-RTS-CA using Alpha Fold compared to RECQ4-WT (Supplementary Fig. 6A-C). Overall, these results suggest that RTS patient-derived mutation in RECQ4 affects MUS81 interaction, is defective in chromosome segregation, and displays similar characteristics to the MUS81 interaction-deficient allele of RECQ4. This implies that loss of RECQ4-MUS81 interaction could play a role in developing Rothmund-Thomson syndrome in a subset of the patients.

## Discussion

The completion of DNA replication is a critical event in the cell cycle, and a proper resolution of replication intermediates is essential for preserving genome integrity. In this study, we focused on the biological role of RECQ4 and its interaction with MUS81, particularly in the context of chromosomal segregation during mitosis. Our key findings demonstrate that RECQ4 physically interacts with MUS81, targeting it to specific DNA substrates and enhancing its endonuclease activity. The depletion of RECQ4 protein or expression of a MUS81-interaction deficient RECQ4 mutant leads to significant chromosomal segregation defects, including the accumulation of micronuclei, anaphase bridges, and ultrafine bridges (UFBs).

Although the exact mechanism of UFB formation and resolution remains incompletely understood, UFBs are believed to arise from unresolved replication and recombination intermediates at hard-to-replicate genomic regions, such as centromeres, fragile sites, and telomeres[33,34]. Interestingly, the detection of various types of UFBs in untreated cells was unexpected, suggesting an intrinsic replication stress or a basal requirement for RECQ4/MUS81 activity even under normal conditions. While we are continuing to investigate this mechanism, similar observations have been reported in studies where the depletion of certain genes involved in DNA metabolism, such as 53BP1[4], RIF1[35], and FIRRM[36], lead to accumulation of the UFBs even in the absence of external replication stress. These findings reinforce the idea that endogenous replication challenges are sufficient to induce chromosomal abnormalities in the absence of adequate resolution mechanisms. Our study provides evidence that the interaction between RECQ4 and MUS81 plays an important role in the resolution of such DNA structures, later manifested as various chromosomal abnormalities at these regions. This is aligned with previously observed defects at centromeres[37] and telomeres[38] upon RECQ4 knockout/knockdown. Consistent with previous observations[39–41], we propose that the persistent accumulation of unresolved replication structures results in chromosome instability at specific loci. This is further supported by the requirement of RECQ4 for chromosome alignment and mitotic progression in a Xenopus cell-free system[25].

The RECQ4-MUS81 interaction occurs primarily during S phase, along with the ability of RECQ4 to stimulate also MUS81-EME2 complex, which is predominantly present in this phase[10], further supports their role in resolving stalled replication intermediates. Our ex vivo

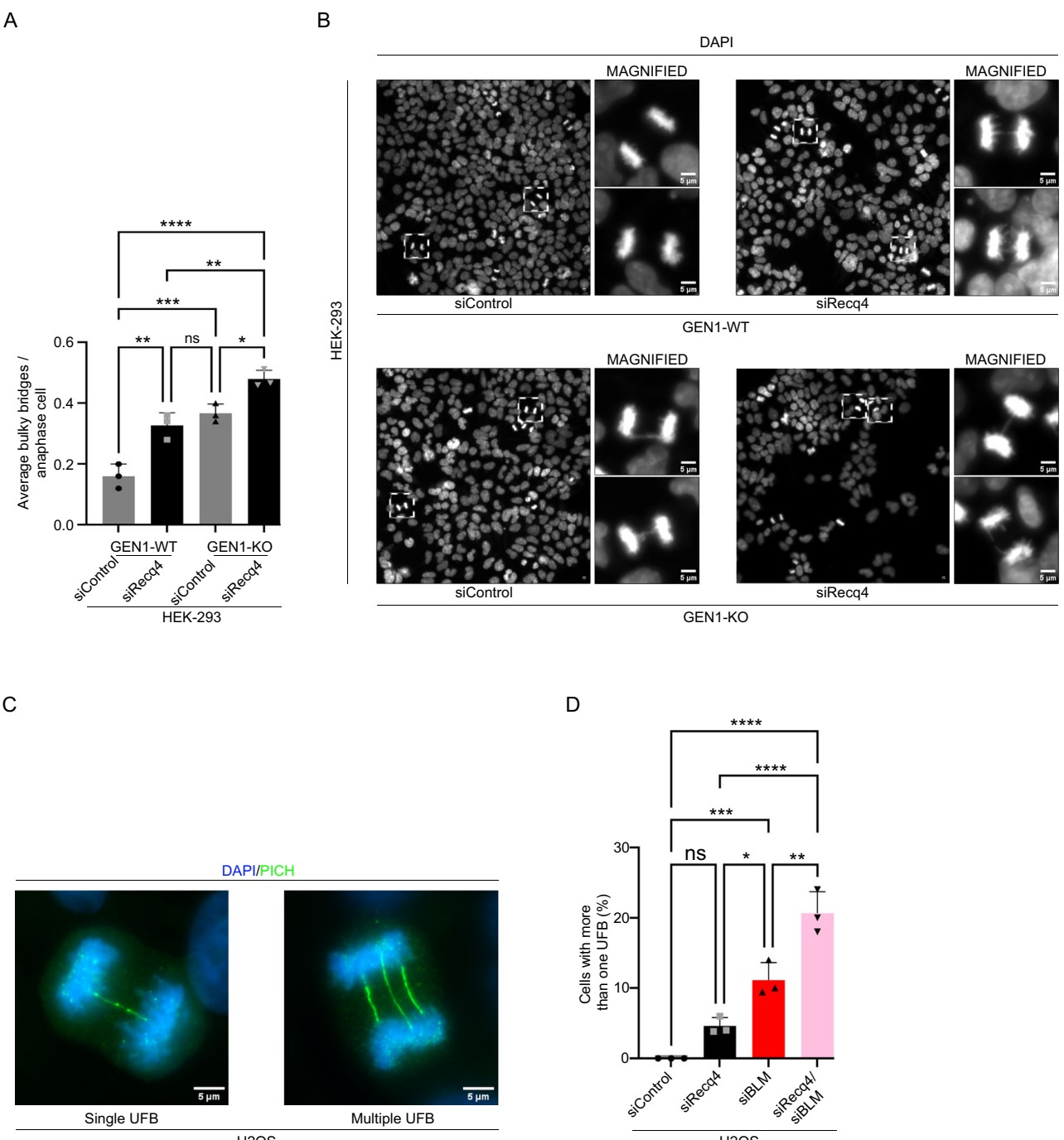

**Fig. 5 | RECQ4 exacerbates cellular function when challenged with other resolution pathways. A** Quantification of average bulky bridges per anaphase cell in HEK-293 (GEN1 -/-) and HEK-293 (wild-type) with siRNA as indicated. $n = 3$ independent experiments; data are means ± SD.; one-way ANOVA followed by Tukey's multiple comparison test; $p$-value were adjusted by GraphPad Prism software; GEN1-WT/siControl vs GEN1-WT/siRecq4 **$p = 0.0020$; GEN1-WT/siControl vs GEN1-KO/siControl ***$p = 0.0005$; GEN1-WT/siControl vs GEN-KO/siRecq4 ****$p < 0.0001$; GEN1-WT/siRecq4 vs GEN1-KO/siControl ns $p = 0.5461$; GEN1-WT/siRecq4 vs GEN-KO/siRecq4 **$p = 0.0034$; GEN1-KO/siControl vs GEN-KO/siRecq4 *$p = 0.0195$. **B** Representative images of HEK-293 (GEN1 -/-) and HEK-293 (wild-type) cells treated with siRNA as indicated. Magnified images of anaphase cells displaying bulky bridges are shown. Scale bar = 5 μm. **C** Representative images of single and multiple UFBs. Scale bar = 5 μm. **D** Quantitative analysis of UFBs in U2OS cells (more than one UFB) with siRNA as indicated. $n = 3$ independent experiments; data are means ± SD.; one-way ANOVA followed by Tukey's multiple comparison test; $p$-value were adjusted by GraphPad Prism software; siControl vs siRecq4 ns $p = 0.0955$; siControl vs siBLM ***$p = 0.0007$; siControl vs siRecq4/siBLM ****$p < 0.0001$; siRecq4 vs siBLM *$p = 0.0195$; siRecq4 vs siRecq4/siBLM ****$p < 0.0001$; siBLM vs sRecq4/siBLM **$p = 0.0021$. Source data are provided as a Source data file.

data indicate that this interaction is involved in the formation of MUS81 foci and their colocalization with hard-to-replicate loci. While MUS81 foci have been observed to colocalize with telomeres, partially centromeres, and possibly to CFSs[42], our findings suggest that this interaction is more pronounced in ALT-positive cells. These cells experience higher intrinsic replication stress within telomeric regions compared to telomerase-positive cells[43]. This is reminiscent of the increased intrinsic replication burden at telomeres observed in ALT cells upon TRF1 depletion, resulting in an increase of telomeric UFBs[44], similar to what we observe with RECQ4 knockdown or

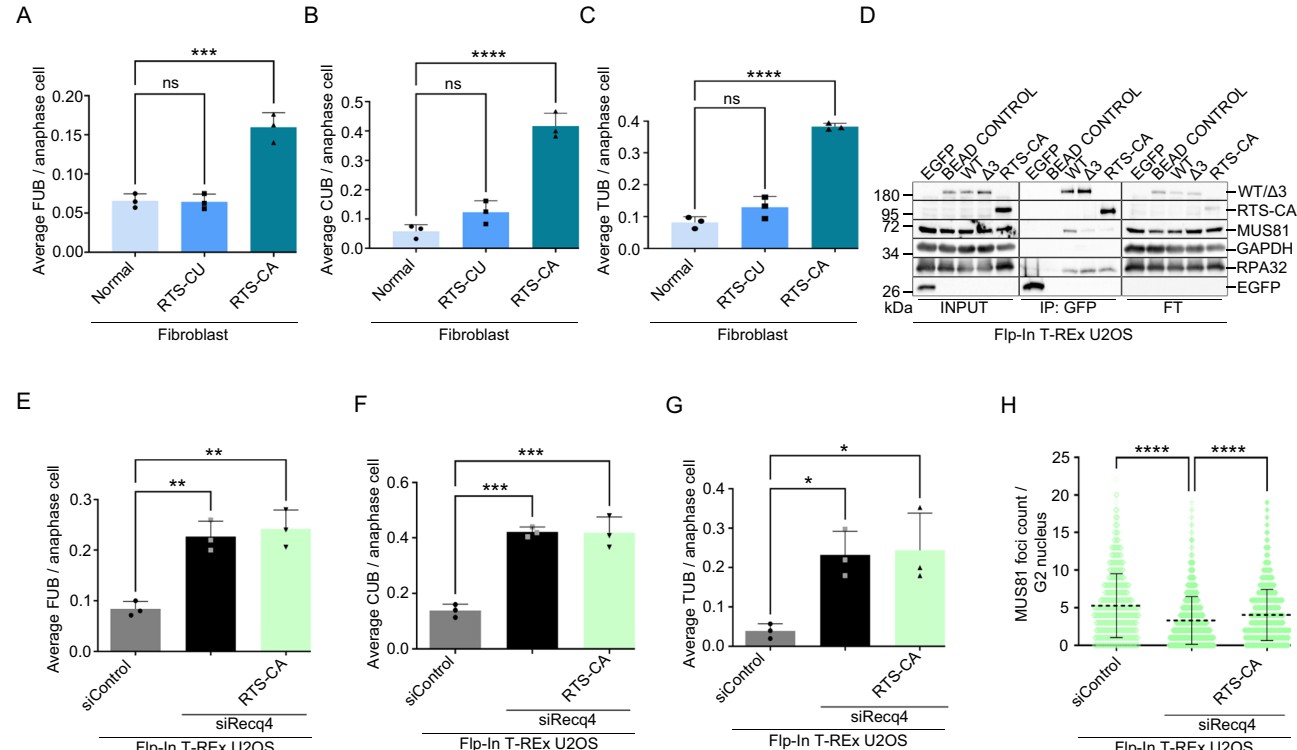

**Fig. 6 | Patient-derived mutation phenocopies MUS81-interaction deficient RECQ4 mutation. A–C** The average number of UFBs per anaphase cell in clinically affected (RTS-CA) and unaffected (RTS-CU) patient fibroblasts, along with normal fibroblasts as indicated. $n = 3$ independent experiments; data are means ± SD; one-way ANOVA followed by Tukey's multiple comparison test; $p$-value were adjusted by GraphPad Prism software. (A) Fragile sites; Normal vs RTS-CU ns $p = 0.9931$; normal vs RTS-CA ***$p = 0.0003$ (B) Centromeric; Normal vs RTS-CU ns $p = 0.1527$; normal vs RTS-CA ****$p < 0.0001$. (C) Telomeric; Normal vs RTS-CU ns $p = 0.0960$; normal vs RTS-CA ****$p < 0.0001$. **D** Immunoprecipitation (IP) from whole cell extracts as shown in Fig. 2A except EGFP-RECQ4-RTS-CA mutant was also included. Proteins from each cell extract (500 µg) were incubated with GFP beads for 1 h and then resolved on SDS-PAGE, followed by western blotting detecting corresponding proteins. Input, IP, and FT fractions were run on different gels respectively. **E–G** The average number of UFBs per anaphase cell in Flp-In T-REx U2OS cells expressing

EGFP-RECQ4-RTS-CA in combination with siControl or siRECQ4; $n = 3$ independent experiments; data are means ± SD.; one-way ANOVA followed by Tukey's multiple comparison test; $p$-value were adjusted by GraphPad Prism software. (E) Fragile sites; siControl vs siRecq4 **$p = 0.0023$; siRecq4 vs RTS-CA/siRecq4 **$p = 0.0013$. (F) Centromeric; siControl vs siRecq4 ***$p = 0.0002$; siRecq4 vs RTS-CA/siRecq4 ***$p = 0.0002$. (G) Telomeric; siControl vs siRecq4 *$p = 0.0259$; siRecq4 vs RTS-CA/siRecq4 *$p = 0.0201$. **H** Quantification of the MUS81 foci in Cyclin A positive Flp-In T-REx U2OS (EGFP-RECQ4-RTS-CA) cells in combination with siRNA as indicated from QIBC analysis in Supplementary Fig. 5I; $n =$ minimum 5000 cells; data are means ± SD; one-way ANOVA followed by Tukey's multiple comparison test; $p$-value were adjusted by GraphPad Prism software; siControl vs siRecq4 ****$p < 0.0001$; siControl vs RTS-CA/siRecq4 ****$p < 0.0001$. Source data are provided as a Source data file.

expression of MUS81-interaction deficient mutant. Further studies are needed to clarify the link between RECQ4-MUS81 interaction and ALT mechanism, to investigate the specific nature and genomic locations of these DNA substrates. Incorporating additional ALT-related pheno-types, such as extrachromosomal telomeric DNA circles and telomeric sister chromatid exchanges, would further enhance RECQ4 link to ALT mechanism.

We identified the MUS81-interaction region within the N-terminus of RECQ4 (353-355) and revealed its role in stimulating MUS81 nuclease activity. The N-terminus of RECQ4 possesses an Sld2-like domain[45], is involved in cell viability[46], and contains various protein interaction regions involved in replication and recombination, including MCM10[19] and BLM[47]. While deletion of the MUS81 binding domain did not abolish DNA binding abilities, RECQ4 devoid of these three amino acids was inefficient for MUS81 stimulation. We also confirmed that the conserved helicase and RQC domains are not involved in MUS81 interaction, but they might still help modulate substrate recognition. The depletion of the MUS81-interaction domain located near the conserved Sld2 domain might affect the function of RECQ4 in DNA replication. However, it has been shown that the replication rate remains unaffected when RECQ4 is depleted[48]. These findings further support the specific role of this complex in the resolution of replica-tion intermediates thereby preventing segregation defects.

Several pathways, including BLM/TOP3/RMI1 (BTR) complex, SLX1/SLX4, MUS81/EME1, XPF-ERCC1 nuclease complex (SMX), and GEN1, are required to cleave/resolve replication and recombination intermediates[9]. Their versatility seems to reflect requirements for processing of various DNA structures and stages of the cell cycle. The BTR pathway plays a crucial role in dissolving DNA intermediates and generating non-crossover (NCO) products, thereby avoiding sister chromatid exchanges (SCEs) and loss of heterozygosity (LOH) in mitotic cells[49]. On the other hand, SMX and GEN1 pathways represent the nucleolytic resolution of other DNA intermediates that persist till mitosis to ensure chromosome segregation[9]. Both pathways seem to work at either G2/M transition or at a later stage of mitosis, based on SMX activation in the form of MUS81-EME1[29] and accessibility of chromatin to cytoplasmatic-localised GEN1[50], respectively. To address the relationship of the RECQ4-MUS81 interaction with these pathways, we co-depleted RECQ4 and BLM, which resulted in a higher frequency of UFBs per cell. Given the primary role of BTR to prevent SCE for-mation, our data indicate that RECQ4/MUS81 might be required for processing a subset of these substrates. This claim is further supported by the observed higher number of SCEs per chromosome in BLM and RECQ4 co-deficient cells[47]. Depletion of RECQ4 in GEN1-KO cells resulted in an additive effect on the accumulation of bulky bridges. This observation is in agreement with the fact that both RECQ4[20] and

GEN1[30] are involved in the dissolution or resolution of branched DNA structures, their simultaneous depletion might lead to a more severe defect in the resolution process. These cooperative functions are further supported by the observed G2 arrest along with MUS81/GEN1 synthetic lethality[51], describing the necessity and coordination of these pathways. Our observations thus highlight the importance of RECQ4/MUS81, BLM, and GEN1 in maintaining genomic stability and that these proteins play distinct but coordinated roles by efficiently resolving and preventing the accumulation of ultrafine bridges.

The coordination between the MUS81 nuclease activity and DNA translocation is supported by previous studies involving RAD54[11] translocase, BLM[14], RECQ5[16], and yeast Srs2 helicases[12]. Interestingly, among various RECQ proteins, RECQ4 showed the highest stimulation efficiency toward MUS81, pointing to its specific and prominent role in facilitating the MUS81 activity in DNA resolution and repair. Although multiple stimulatory factors may reflect cell cycle requirements or the need to process specific intermediates. For instance, RECQ5 may be more selective for removing RAD51 from CFSs and promoting their cleavage by MUS81[16]. The specific function of RECQ5 could also stem from its role in regulating RNAPII and in the MUS81-mediated replication fork cleavage that is driven by collisions between replication and transcription complexes[52]. On the other hand, BLM might be particularly required for the processing of replication/recombination intermediates to prevent recombination ultrafine bridges (HR-UFBs) that accumulate upon loss of MUS81 and GEN1 and are decorated by BLM protein[51]. Separate interaction domains for BLM and RECQ4 within MUS81 may also indicate different mechanisms of interaction for MUS81. While the BLM-interaction domain was mapped to the region 125–244[14], we located the RECQ4-interaction domain to the first 1–125 amino acids of MUS81.

Furthermore, our study suggests a potential role for the RECQ4-MUS81 interaction in cellular stability, with implications for understanding Rothmund-Thomson syndrome. However, it is important to note that other RECQ4-dependent functions likely also contribute to RTS pathology. Notably, several RTS-associated mutations, including the RTS-CA mutation that results in a truncated form of RECQ4, are clustered near the region we identified as critical for RECQ4-MUS81 interaction[31]. In this study, we observed impaired RECQ4 interaction with MUS81, reduced MUS81 foci formation, and defects in chromosome segregation in both RTS patient-derived fibroblast and engineered Flp-In T-REx U2OS cells expressing RECQ4-RTS-CA. The similar segregation defects have been observed in cells with mutations in the ANAPC1[53], which cause RTS Type I, whereas RTS Type II is associated with mutations in the RECQ4 gene. Their possible mechanistic relationship is supported by the direct interaction between APC1 and RECQ4[54]. Our findings of accumulation of UFBs in RTS-derived cells along with chromosomal aberrations, and enhanced telomere fragility[38] observed in fibroblasts derived from RTS patients, reinforce this potential connection. Apart from RTS developmental disorder, we want to highlight the clinical relevance of our study in the context of cancer disease. Previous studies reported that RECQ4[55], MUS81[56], and GEN1[57] are frequently overexpressed in many cancers, potentially indicating that cancer cells rely more on these proteins, particularly in cases of oncogene-induced replication stress[58]. Therefore, we speculate that not only the inactivation of these genes but also their overexpression can be detrimental, leading to genome instability, potentially driving tumorigenesis.

In summary (Fig. 7), our data suggest that during S phase, RECQ4 recognises replication or recombination intermediates at hard-to-replicate regions and facilitates loading of MUS81 to these sites through a direct physical interaction to promote their resolution. Failure to resolve such replication intermediates leads to chromosome misalignment, delayed metaphase, and accumulation of UFBs in anaphase, resulting in aneuploidy. Similar phenotypes for RTS-associated mutation emphasize the importance of RECQ4-MUS81 interaction for proper chromosome segregation that may contribute to disease development.

## Methods

### Plasmid and stable cell line preparation

The EGFP-RECQ4 (WT) plasmid was generated by amplifying RECQ4 along with an EGFP tag from pET32a-RECQ4 and pEGFP-C1, respectively, using indicated primers (pR2572, pR2573 for EGFP and pR2574 and pR2575 for RECQ4, Supplementary Data 1). This amplified product was then inserted into the EcoRV site of the pAIO plasmid[59], generating an N-terminal fusion of EGFP with RECQ4, along with a C-terminal HA-tag. This plasmid contains a tetracycline-inducible system with a modified CMV promoter of medium strength, ensuring expression levels similar to endogenous RECQ4. Additionally, this plasmid includes the shRNA target sequence for RECQ4 (5'-TAGGAA-GAGCCTCATCTAAG-3') cloned between the BglII and HindIII sites using shRECQ4 FOR and shRECQ4 REV oligos (Supplementary Data 1). The exogenous RECQ4 sequence in pAIO plasmid was codon-optimised to resist shRNA and siRNA targeting endogenous RECQ4, using indicated primers (pR2576/pR2577, and pR3762/pR3763 respectively, Supplementary Data 1). The EGFP control, EGFP-RECQ4 (Δ3), EGFP-RECQ4 (RTS-CA), and EGFP-RECQ4 (AAA) mutant plasmids were generated by site-directed mutagenesis of EGFP-RECQ4 (WT) plasmid using corresponding primers (EGFP control (pR4250 and pR4251), EGFP-RECQ4 (Δ3) (pR1781 and pR1782), EGFP-RECQ4 (RTS-CA) (pR4175, pR4176, pR4177 and pR4178) and EGFP-RECQ4 (AAA) (pR5187 and pr5188), Supplementary Data 1). Sequence of all constructs was verified by sequencing. For stable cell line generation, the corresponding plasmids, along with Flp-In recombinase plasmid, were transfected into Flp-In T-REx U2OS cells using Lipofectamine™ 3000 Transfection Reagent (L3000001, Invitrogen) according to the manufacturer's protocol. Stable cell lines were selected with 100 µg/mL of hygromycin.

### Cell culture

The U2OS (92022711), and HeLa (93021013) cell lines were obtained from the European Collection of Authenticated Cell Cultures (ECACC). HT1080 cells was obtained from Dr. Soucek (Institute of Biophysics, Brno). CAL51 was obtained from Dr. Janscak (University of Zurich, Zurich). Saos-2 was obtained from Dr. Uldrijan (Masaryk University, Brno). SJCRH30 (CRL-2061), SJSA-1 (CRL-2098), and 143B (CRL-8303) were acquired from American Type Culture Collection (ATCC). Clinically unaffected RTS-CU fibroblast, female (AG18373), and Clinically affected RTS-CA fibroblast, male (AG18371) were obtained from Coriell Institute for Medical Research. Flp-In T-REx U2OS cell line was acquired from MRC PPU Reagents (University of Dundee). LM216J (ALT-positive) and LM216T (ALT-negative) cell lines were obtained from Dr. Roderick J. O'Sullivan (University of Pittsburgh) and were originally described by J.P Murnane[60]. HEK-293 (GEN1 -/-) and HEK-293 (wild-type) were obtained from Dr. Stephen C. West (The Francis Crick Institute). All cell lines were maintained in DMEM (LM-D1108/500, Biosera) supplemented with 10% FBS (FB-1001T/500, Biosera), Glutamine (XC-T1715/100, Biosera), and penicillin-streptomycin (XC-A412/100, Biosera). Flp-In T-REx U2OS stable cell lines were cultured in DMEM with tetracycline-free FBS (FB-1001T/500, Biosera), along with additional antibiotics: 100 µg/mL hygromycin (10687010, Invitrogen) and 15 µg/mL blasticidin (A1113903, Thermo Fisher Scientific). AG18371 and AG18371 (RTS patient fibroblasts) were cultured with 15% FBS instead of 10%. For expression of RECQ4 constructs, 1 µg/mL of doxycycline (D9891, Sigma-Aldrich) was added to the growth medium for 48 h. To ensure a significant reduction of endogenous RECQ4 levels, siRNA targeting RECQ4 was employed in conjunction with shRNA expressed from the pAIO plasmid. Respective siRNA (20 nM) was transfected with lipofectamine RNAiMAX (13778075, Invitrogen). The siRNA used in this study include: siControl (siGENOME non-Targeting siRNA #3,

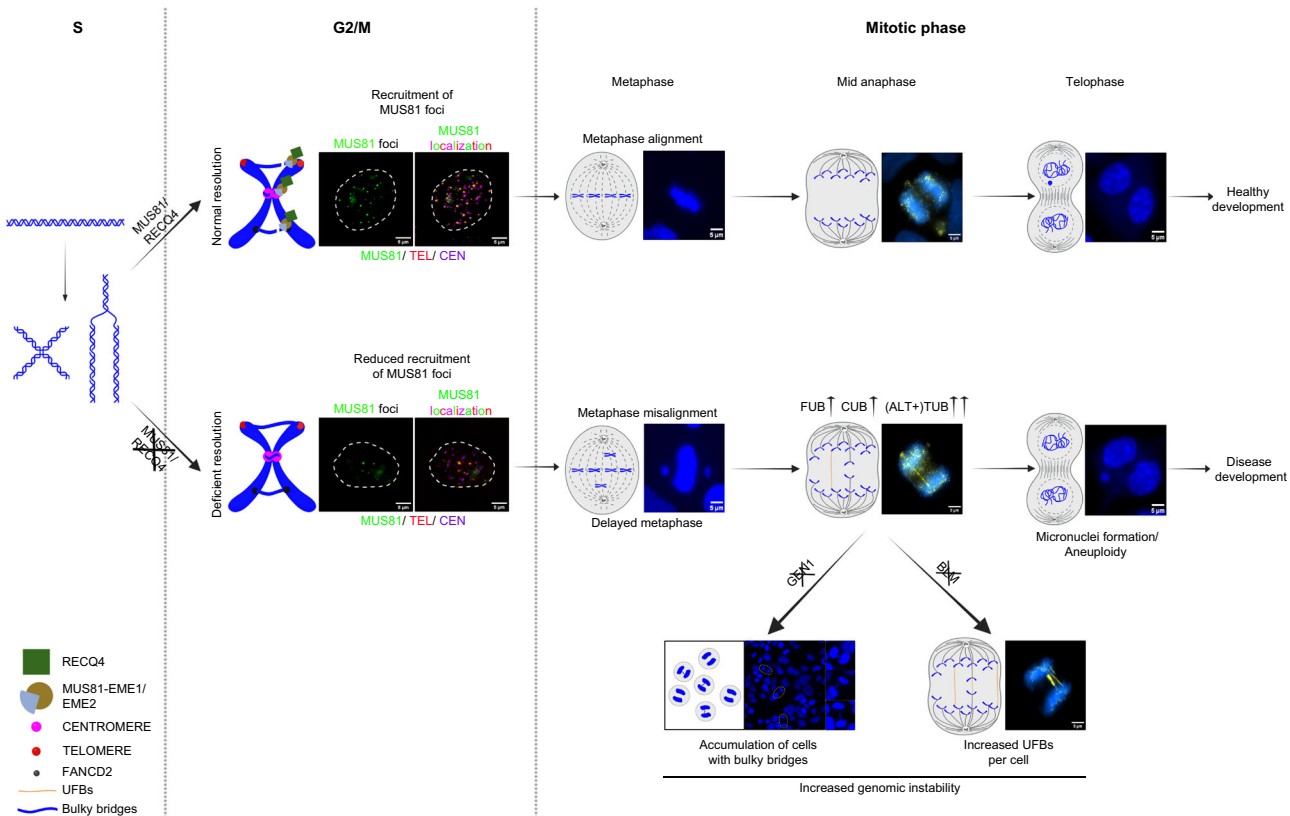

**Fig. 7 | Model of RECQ4 coordination with MUS81 in processing replication/ recombination intermediates.** MUS81 interacts with RECQ4 in the S phase and stimulates MUS81-EME2 to resolve replication, recombination intermediates/stalled replication forks. During the G2/M transition, RECQ4 further activates MUS81 in the form of MUS81-EME1 which helps in the resolution of Holliday junction/late replication intermediates, at this stage MUS81 foci are visible and colocalise predominantly with telomeres and partially with centromeres. The proper operation and activation of MUS81 complexes play a crucial role in ensuring a seamless transition from prometaphase to anaphase by preventing misalignment and the

build-up of UFBs (with TUBs being more vulnerable in ALT-positive cells). These actions ultimately prevent the formation of micronuclei, which can lead to aneuploidy and disease development, including RTS. Loss of RECQ4 in the background of GEN1 knockout makes the cells very sick further proposing GEN1 as an essential resolution pathway which possibly is the final resolution check during the progression of the cell cycle as GEN1 is only accessible to DNA during the mitotic phase. Conversely, the depletion of both BLM and RECQ4 leads to an increase in unresolved SCEs, as evidenced by the presence of more UFBs per cell. Scale bar = 5 μm. Created in BioRender. Ashraf, R. (2025) https://BioRender.com/o78n251.

D-001210-03-05, Dharmacon); siMUS81 (target sequence, 5'-CGCGCTTCGTATTTCAGAA-3', S37038, Thermo Fisher Scientific); siRECQ4 (target sequence, 5'-GGCTCAACATGAAGCAGAA-3', S17991, Thermo Fisher Scientific) and siBLM (SMARTpool, M-007287-02-0005, Dharmacon).

## Co-Immunoprecipitation
GFP-Trap® Magnetic Particles M-270 Kit supplied with lysis and washing buffer (gtdk-20, Chromotek) was used for immunoprecipitation of 500 μg of the whole cell lysate as described by the manufacturer's protocol. Briefly, cells were lysed in cell lysis buffer, and the lysate was incubated with 20 μL of beads at 4 °C for 1 h on a rotator. A 10% aliquot of the lysate was kept as input. After incubation, the beads were separated on a magnetic separation stand (Z5342, Promega), and 10% of the flow-through (FT) was retained for further analysis. The beads were washed three times with 500 μL of washing buffer, and bound proteins were eluted with 2x SDS Laemmli buffer. All samples were heated at 100 °C for 5 min and analysed by SDS-PAGE followed by western blotting.

## Western Blot
Cells were seeded in 60 mm dishes and allowed to express EGFP-RECQ4 constructs or EGFP control by treatment with doxycycline (1 μg/mL) together with reverse transfection with the indicated siRNA for 48 h After this period, the cells were harvested and lysed in RIPA

buffer (R0278, Sigma-Aldrich). Cell samples were prepared in 4X SDS Laemmli buffer, heated at 100 °C for 5 min, and 80 μg of protein was loaded on a 10% SDS-PAGE gel. Proteins were then transferred to a nitrocellulose membrane using a semi-dry Trans-blot turbo Transfer system (1704150, Biorad). After transfer, membranes were blocked in 5% milk/TBST or BSA/TBST for 1 h at room temperature, followed by incubation with a corresponding primary antibody overnight at 4 °C on a rocker. The primary antibodies used were: (Anti-RECQ4, 1:700 dilution, 2814, CST); Anti-GAPDH (1:1000 dilution, 3683, CST); Anti-GFP (1:2500, ab290, Abcam); Anti-MUS81 (1:1000 dilution, ab14387, Abcam) and Anti-CENPF (1:1000 dilution, 58982S, CST). The following day, membranes were washed with TBST and incubated with the corresponding secondary antibodies: Anti-Rabbit IgG (1:10000 dilution, A6154, Sigma-Aldrich), and Anti-Mouse IgG (1:15000, A0168, Sigma-Aldrich) for 1 h at room temperature. The blots were developed by Immobilon Western Chemiluminescent HRP Substrate (WBKLS0500, MERCK Millipore), and images were acquired using a Luminescent Image Analyzer (ImageQuant™ LAS 4000, FUJIFILM corporation).

## Cell synchronisation
Cells were seeded in 100 mm dishes and subjected to a double thymidine block using 2 mM thymidine (T1895, Sigma-Aldrich). The first block lasted for 18 h, after which the cells were released into fresh media for 8 h. Subsequently, a second thymidine block was applied for another 18 h. Following this, an aliquot of cells synchronised in G1/S

was collected. To obtain cells in the S phase, cells were released into fresh media for 6 h before harvesting. For G2/M phase synchronisation, after allowing cells to progress through the S phase for 8 h post-G1/S release, RO3306 (9 μM) was added for 3 h to arrest cells in the G2M phase, and the cells were then collected. To obtain cells in M phase, the G2/M-arrested cells were further released into fresh media containing 100 nM nocodazole for 2 h. Mitotic cells were subsequently collected by shake-off.

## Micronuclei/bulky bridges staining

To visualise micronuclei, cells were seeded in a 96-well plate and subjected to reverse siRNA transfection for 48 h. Cytochalasin B (2 μg/mL, C6762, Sigma-Aldrich) was added for the last 16 h to inhibit cytokinesis. Cells were then fixed and stained with DAPI (A1001, Appli-Chem) at a final concentration of 2 μg/mL. At least 60 binucleate cells were visualised using Nikon Eclipse Ti inverted fluorescence microscope with a Plan Fluor ELWD 40x Ph2 DM objective (numerical aperture 0.6) and were analysed for micronuclei as average micronuclei per binucleated cells. For bulky bridge analysis, cells were seeded and treated similarly, but without cytochalasin B. Mid-anaphase cells were visualised using the same microscope. At least 30–50 cells were analysed for bulky bridges, as average bulky bridges per anaphase cell in each condition.

## Mitotic progression

Cells were seeded in 35 mm dishes (81156, Ibidi) with the indicated reverse transfection, and expression of RECQ4 constructs was induced by doxycycline (1 μg/mL) for 48 h. Cells were then stained with SiR-DNA dye (SC007, Spirochrome) at a 1:1000 dilution and incubated under the same culture conditions for 1 h. Live cell imaging was done at 37 °C with a continuous supply of 5% $CO_2$. Images were acquired every 10 min for 3 h using Nikon Eclipse Ti inverted fluorescence microscope with a Plan Fluor ELWD 40x Ph2 DM objective (numerical aperture 0.6), using a Cy5 filter. After image acquisition, all mitotic cells were analysed for mitotic progression from metaphase to anaphase, with at least 45 cells examined in each condition.

## Ultrafine bridge staining

The protocol for ultrafine bridges UFBs staining was adapted from Dr. Chan group[61]. Briefly, cells were seeded in 6-well plates and subjected to reverse transfection with siRNA and doxycycline for expression of RECQ4 constructs for 24 h. After that, cells were trypsinised, reseeded onto coverslips (22 × 22 mm, 0107052, Marienfeld) at a density of 0.22 million cells per coverslip, and allowed to adhere with siRNA reverse transfection and doxycycline for further 24 h. The cells were then fixed with 4% methanol-free formaldehyde (28908, Thermo Fisher Scientific) for 10 min at room temperature. Slides were washed three times with PBS, permeabilised in a buffer containing 0.5% Triton X-100 (T8787, Sigma-Aldrich) and 5% FBS in PBS at 4 °C for 20 min, and washed three times with PBS. Blocking was performed by 5% FBS at room temperature for another 20 min. After this, slides were incubated with indicated primary antibodies: Anti-ERCC6L (1:100 dilution, H00054821-B01P, Abnova), Anti-Centromere (1:400 dilution, HCT-0100, Immuniovision), and Anti-FANCD2 (1:500 dilution, NB100-182, Novus Biologicals) at 37 °C for 80 min. After three washes with PBS, slides were incubated with corresponding secondary antibody: Anti-Mouse IgG, Alexa flour 488 (1:500 dilution, A11001, Thermo Fisher Scientific); Anti-Mouse IgG, Alexa flour 546 (1:500 dilution, A21123, Thermo Fisher Scientific); Anti-Rabbit, Alexa flour 555 (1:500 dilution, A31572, Thermo Fisher Scientific); Anti-Human IgG, DyLight®650 (1:500 dilution, ab98622, Abcam); Anti-Rabbit IgG, Alexa Fluor Plus 647 (1:500 dilution, A32733, Thermo Fisher Scientific); Anti-Mouse IgG (H + L), Alexa flour 594 (1:500 dilution, A11005, Thermo Fisher Scientific) and Anti-Mouse IgG (H + L), Alexa Fluor™ 568 (1:500 dilution, A11061, Thermo Fisher Scientific) for 25 min at 37 °C. Then slides were

washed three times with PBS. If the FISH was to be combined, slides were prepared accordingly, otherwise, they were washed with MiliQ water, air-dried in the dark, and mounted with Vectashield/DAPI (H-1200, Vector Laboratories) on glass slides. Imaging was done on mid-anaphase cells, with 30–50 cells acquired for analysis as average UFB per cell using a Nikon Eclipse Ti inverted fluorescence microscope equipped with a Plan Apo VC 60 x A WI DIC N2 objective (numerical aperture 1.2) and appropriate filters. DAPI-positive DNA bridges (bulky bridges) were also analysed from these datasets.

## FISH

After staining the coverslips with primary and secondary antibodies, they were fixed with 2 % formaldehyde for 5 min at room temperature in the dark. The coverslips were then washed twice with PBS and subjected to denaturation for 40 min at 72 °C in 2xSSC buffer (15557044, Thermo Fisher Scientific). Following a rinse with PBS, the coverslips were dehydrated gradually with 70%, 90%, and 100% ethanol (1.00983.2500, Merck Millipore), with 5-min interval for each concentration, and air dried. Next, the coverslips were incubated in a hybridisation buffer consisting of 70% formamide (A0937, Panreac Applichem), 0.5% maleic acid (11096176001, Sigma-Aldrich) as a blocking reagent, 10 mM Tris pH 7.5, telomere probe Tel-Cy3 (1:250 dilution, F1002, Pangene), and 2 x SSC. This mixture was denatured at 85 °C for 3 min, followed by incubation at room temperature for 90 min. Finally, the coverslips were washed twice with 2xSSC at 37 °C with gentle rocking, once with PBS, and then rinsed with MiliQ water. The coverslips were air-dried in the dark and mounted with Vectashield/DAPI.

## Analysis of MUS81 foci

For QIBC analysis of MUS81 foci (Figs. 4E, 6H, Supplementary Fig. 4F, G and Supplementary Fig. 5I), cells were grown on round glass coverslips (12 mm diameter, 1.5 mm thickness, 6307356, Menzel-Glaser). Ectopic expression of different RECQ4 proteins was induced by doxycycline for 48 h, and the endogenous RECQ4 was depleted for 48 h. siGENOME non-Targeting siRNA #3 (4457287, Thermo Fisher Scientific) was used as the control siRNA. After treatment, cells were fixed with 4 % buffered formaldehyde for 15 min at room temperature, followed by cell permeabilization using PBS containing 0.2 % TritonX-100 for 5 min at room temperature. Primary antibodies against MUS81 (1:1000, mouse, sc-53382, Santa Cruz) and cyclin A (1:1000, rabbit, sc-596, Santa Cruz) were diluted in DMEM medium containing 10% FBS and 0.05% sodium azide (filtered through a 0.2 μm filter) and incubated at room temperature for 90 min. Respective secondary antibodies and DAPI (0.5 μg/mL) were diluted in DMEM and set at room temperature for 30 min. After staining, coverslips were washed three times with PBS and additionally twice in distilled water, dried, and mounted with a Mowiol-based mounting medium containing 12% Mowiol 4-88 (81381, Sigma-Aldrich), 30% glycerol (G5516, Sigma-Aldrich), 0.12 M Tris-HCl pH 8.5 (RES3098T-B701X, Sigma-Aldrich). QIBC (Quantitative image-based cytometry) was performed as described[62]. Images were acquired using Scan R inverted high-content screening microscope (Olympus) equipped with wide-field optics, UPLSAPO dry objective (20X, 0.75 numerical aperture), fast excitation and emission filter-wheel devices for DAPI, FITC, Cy3 and Cy5 wavelengths, an MT20 illumination system and a digital monochrome Hamamatsu ORCA-R2 CCD camera. Image acquisition was performed using automated scanR acquisition software (Olympus, v.2.7.1) at non-saturating conditions (12-bit dynamic range). Approximately 5000 cells per condition were processed and analysed with scan R analysis software (Olympus, v.2.7.1). An automated dynamic background correction (thresholding at least fivefold pixel intensity above background levels) was applied to all images for each channel separately but maintaining similar parameters for all the treatments within an experiment. Individual cell nuclei were segmented based on the DAPI signal as the main objects and MUS81 foci as

sub-objects to obtain various measurements, such as mean and total intensities, foci count, and foci intensities. Then, a table with values was exported and analysed in Spotfire software (TIBCO, V10.5.0.72). To visualise overlapping markers, low y-axis jittering was applied in scatter plots (random displacement of objects along the y-axis).

Due to the unavailability of the scanR microscope for some of the experiments, we used Nikon Eclipse Ti inverted fluorescence microscope and Image J to analyse MUS81 foci in panel of ALT positive and negative cell lines (Figs. 4A, 4F, Supplementary Fig. 4B, C and Supplementary Fig. 4H). Cells were seeded as described earlier, briefly, cells on coverslips were either left untreated or subjected to reverse transfection with indicated siRNA along with transient transfection with indicated plasmids for 48 h and were then fixed and incubated with respective primary and secondary antibodies. Fluorescence images were processed in ImageJ (Java 1.8.0_172). Regions of interest (ROIs) were defined based on DAPI staining, and images were converted to 8-bit grayscale. Background subtraction was applied, followed by thresholding to isolate foci. Foci detection was carried out using the "Find Maxima" function, with the Prominence threshold adjusted to minimize background detection. The number of foci within each ROI was recorded and analysed, with results depicted as MUS81 foci per individual cell. To ensure accuracy, foci counts were also verified through manual counting.

To study the effect of RECQ4 knockdown on MUS81 foci in U2OS, LM216J and Saos-2 cell lines (Fig. 4B–D and Supplementary Fig. 4A), cells were seeded in 6-well plates and either left untreated or subjected to reverse transfection with siControl or siRNA targeting RECQ4 for 24 h. Subsequently, cells were trypsinised and plated in a 96 well plate (6055300, Perkin Elmer) for another 24 h. Following this, cells were fixed and stained as described above. After staining, plates were washed three times with PBS and stained with DAPI (R37606, Thermo Fisher Scientific). Images were acquired using a scanR inverted high-content screening microscope (Olympus) as described above. MUS81 foci were detected and quantified using CellProfiler software version 4.2.1 (Broad Institute of MIT and Harvard)[63]. The analysed data were plotted in RStudio software using R programming language (Team, R., 2020. RStudio: Integrated Development for R. RStudio, PBC, Boston, MA; Team, R.C., 2022. R: A Language and Environment for Statistical Computing), with minimally 5000 cells analysed.

### Analysis of ALT-associated PML bodies (APBs)
Cells were seeded in 6-well plates followed by reverse transfection with indicated siRNA and doxycycline for expression of RECQ4 constructs for 24 h. After that, cells were trypsinised, reseeded onto coverslips (22 × 22 mm) at a density of 0.22 million cells per coverslip, and allowed to adhere with siRNA reverse transfection and doxycycline for further 24 h. The cells were then fixed with 4% methanol-free formaldehyde for 10 min at room temperature. Slides were washed three times with PBS, permeabilised in a buffer containing 0.5% Triton X-100 and 5% FBS in PBS at 4 °C for 20 min, and washed three more times with PBS. Blocking was performed by 5% FBS at room temperature. After this, slides were incubated with the primary antibody: Anti-PML (1:200 dilution, sc-966, Santa Cruz Biotechnology) at 37 °C for 80 min followed by secondary antibody: Anti-Mouse IgG2a, Alexa flour 647 (1:500 dilution, A21241, Thermo Fisher Scientific) for 25 min at 37 °C. Slides were washed three times with PBS followed by FISH as described earlier. Imaging was done using Nikon Eclipse Ti inverted fluorescence microscope equipped with a Plan Apo VC 60xA WI DIC N2 objective (numerical aperture 1.2) and appropriate filters. By using Image J (Java 1.8.0_172), fluorescent images were split into different channels for PML and Telomere staining and then merged to get a composite image. Colocalised PML/ Telomere as APBs were counted manually and represented as the number of APBs foci per individual cell and a minimum of 500 cells were analysed per condition.

### Colocalization analysis
Slides were prepared and imaged for MUS81, Centromere, and Telomere staining as described above for Ultrafine bridges staining and FISH sections. Colocalization was performed using Image J with the JACoP (Just Another Co-localisation Plugin)[64]. We applied the Costes automatic thresholding method to determine colocalization. The extent of colocalization was quantified using the Pearson correlation coefficient. A minimum of 50 cells were analysed for each parameter.

### Cell cycle analysis by FACS
Following cell synchronization, cells were harvested, washed with PBS, trypsinised, and fixed in 70% ethanol. The fixed cells were stored at -20 °C for further processing. Ethanol was removed by centrifugation at 1500 rpm, and the cells were washed again with PBS, discarding the supernatant. The pelleted cells were resuspended in a staining solution containing Propidium Iodide (PI, 40 µg/mL, P4170, Sigma-Aldrich), RNase A (100 µg/mL, A2760.0100, Panreac Applichem), and 0.1 % Triton X-100. The samples were incubated at 37 °C for 40 min and then centrifuged again. The resulting pellet was dissolved in 500 µL PBS. Subsequently, cell cycle acquisition was performed using a BD FACS verse Flow Cytometer (BD Biosciences), and a minimum of 10,000 events were recorded for each sample. Cells were gated to exclude debris and aggregates based on forward and side scatter (FSC and SSC) profiles. PI fluorescence was evaluated on a linear scale to differentiate between cells in the G0/G1 (2 N DNA content), S (intermediate DNA content), and G2/M (4 N DNA content) phases. The data were analysed using BD FACSuite software V1.0.6 (BD Biosciences).

### Multiple sequence alignment
Amino acid sequence from the various organisms, as indicated, were obtained from the UniProt database. These sequences were aligned using T-COFFEE[65] (Version_11.00) to generate multiple sequence alignment. The resulting alignment was then visualised using ESPript 3.0[66] to produce a high-quality image displaying residues according to the conservation.

### 3D protein structure prediction
The protein structures were predicted using AlphaFold[67], an advanced deep learning-based tool for protein structure prediction. Amino acid sequences of the target proteins were provided as input to AlphaFold. The software utilized these sequences to predict three-dimensional protein structures, employing its highly accurate models trained on a diverse dataset of known protein structures. The predicted structures were then analysed, and images were generated using UCSF Chimera[68].

### Cloning of RECQ4 fragments and mutants
Cloning of RECQ4 constructs was performed as previously described[20]. Briefly, individual RECQ4 constructs were amplified with specific primers RECQ4 (1-269) (pR405 and pR826), RECQ4 (1-322) (pR405 and pR1011), RECQ4 (1-400) (pR405 and pR577), RECQ4 (1-492) (pR405 and pR705), RECQ4 (455-1208) (pR1013 and pR401) (Supplementary Data 1) using pET32a-RECQ4-9XHis as the template. The resulting PCR products, which included the desired fragment with a C-terminal 9xHis-tag, were inserted into the EcoRI site of the pMAL-c2x expression vector. The RECQ4 deletion and substitution mutants, pMAL-c2x-RECQ4 (1-400) Δ3 and pMAL-c2x-RECQ4-RTS-CA, were created by site-directed mutagenesis of pMAL-c2x-RECQ4 (1-400) and pMAL-c2x-RECQ4 (1-500)[20], respectively using Hifi PCR premix (639298, Clontech) with indicated primers, pMAL-c2x-RECQ4 (1-400) Δ3 (pR1781 and pR1782) and pMAL-c2x-RECQ4-RTS-CA (pR5258, pR5259, pR5260 and pR5261) (Supplementary Data 1). All constructs were verified by sequencing to ensure accuracy.

### Expression and purification of RECQ4 and its fragments
Full-length RECQ4 (WT) was purified as described earlier[69]. Briefly, the pGEX-RECQ4-9xHis plasmid, with RECQ4 fused to GST at the

N-terminus, was transformed into *E. coli* Rossetta (DE3) cells. A single bacterial colony was transferred into 2x TY medium containing ampicillin (0.1 mg/mL) and grown overnight at 37 °C with constant shaking. Protein expression was induced by 0.3 mM IPTG (A1008,0100, Panreac Applichem) for 18 h at 16 °C. Cells were harvested by centrifugation, resuspended, and sonicated in CBB buffer (50 mM tris-HCl pH 7.5, 10% sucrose, 2 mM EDTA) containing 200 mM KCl (131494.1211, Panreac Applichem), 0.01% NP40 (A1694,0500, Panreac Applichem), 1 mM DTT (A1101,0100, Panreac Applichem), and a cocktail of protease inhibitors (aprotinin, chymostatin, leupeptin, pepstatin A, and benzamide hydrochloride, each at 5 µg/mL each). The crude extract was clarified by ultracentrifugation (100,000x g, 90 min, 4 °C) and passed through the Q Sepharose column (17-0510-05, GE Healthcare, 40 mL) and then onto SP Sepharose column (17-0729-05, GE Healthcare, 30 mL). The SP column was eluted with a gradient of KCl (150-1000 mM) in buffer (20 mM K2HPO4, 10% glycerol, 0.5 mM EDTA, 150 mM KCl, 0.01% NP40, 1 mM DTT). Fractions containing RECQ4 (WT) were mixed with 2 mL Glutathione Sepharose 4 Fast Flow (17-5132-03, GE Healthcare) for 2 h at 4 °C. The matrix was washed with buffer 1X K containing 500 mM KCl, 0.01% NP40, and 1 mM DTT, and the bound protein was eluted in steps with 10, 50, 100, and 200 mM glutathione (A2084,0025, A2084,0025, Panreac Applichem) in buffer 1X K containing 500 mM KCl, 0.01% NP40, and 1 mM DTT. Peak fractions eluting in 50 mM glutathione, were dialysed against buffer 1X K to remove glutathione and then incubated with PreScission Protease overnight at 4 °C to cleave off the GST tag. The cleaved protein was mixed with His-Select nickel affinity gel (P6611, Sigma-Aldrich, 0.5 mL) for 2 h at 4 °C. The beads were washed with buffer 1X K containing 500 mM KCl, 0.01% NP40, 1 mM β-mercaptoethanol (A4338,0250, Panreac Applichem), and 10 mM imidazole (A1073,0500, Panreac Applichem), and the bound protein was eluted increasing concentrations of imidazole (150-1000 mM). Fractions containing purified RECQ4 (WT) were concentrated using VivaSpin centricon.

Full-length RECQ4 (MBP-RECQ4-His) was also expressed using a baculovirus expression vector system-BEVS (ProTech, Vienna Biocentre Core facilities) in High Five insect cells. The cells were lysed by sonication in lysis buffer (50 mM Tris–HCl pH 7.5, 10% sucrose, 0.5 mM EDTA, 600 mM KCl, 1 mM BME, 0.1% NP40, protease inhibitors) and cleared lysate was incubated with Ni-NTA affinity resin. Unbound proteins were washed out by lysis buffer followed by a wash in 25 mM Tris–HCl pH 7.5, 300 mM KCl, 10% glycerol, 0.5 mM EDTA, 1 mM BME, and 0.01% NP40. The protein was eluted in the same buffer supplemented with 400 mM imidazole.

RECQ4 fragments were purified as previously described[20]. Bacterial strain E. coli BL21(DE3) pLysS was transformed with individual plasmids encoding the various RECQ4 fragments. Individual bacterial colonies from LB plates were grown overnight in 2X TY medium containing ampicillin (0.1 mg/mL) at 37 °C with shaking. Protein overexpression was induced by the addition of 0.1 mM IPTG for 18 h at 16 °C. The cell pellet was sonicated in CBB buffer supplemented with 200 mM KCl, 0.01% NP40, 1 mM β-mercaptoethanol, and a cocktail of protease inhibitors (aprotinin, chymostatin, leupeptin, pepstatin A and benzamide hydrochloride, each at 5 µg/mL). The crude lysate was clarified by ultracentrifugation (100,000x g for 1 h at 4 °C) and the supernatant was incubated with His-Select nickel affinity gel (1 mL) overnight at 4 °C. The beads were washed with buffer 1X K containing 150 mM KCl, 0.01% NP40, 1 mM β-mercaptoethanol, and 10 mM imidazole. Bound proteins were eluted with buffer 1X K supplemented with increasing concentrations of imidazole (150-1000 mM). Fractions containing RECQ4 fragments were mixed with 0.5 mL Amylose Resin High Flow for 1 h at 4 °C. The bound protein was washed with buffer 1X K and eluted with buffer 1X K supplemented by 10 mM maltose. Pooled fractions containing RECQ4 (1-322), (1-400) and (1-400) Δ3, and RTS-CA proteins were loaded onto MonoS (0.5 mL) and eluted using a KCl gradient (150-1000 mM) in buffer 1X K. The RECQ4 (455-1208)

fragment was purified using a MonoQ column (0.5 mL) with a KCl gradient (150-1000 mM) in buffer 1X K. Purified RECQ4 fragments were concentrated using VivaSpin centricon and stored in small aliquots at -80 °C.

## Expression and purification of MUS81-EME1

The bacterial *E. coli* Rossetta (DE3) pLysS strain was transformed with plasmids having pDEST15 as the backbone encoding full length GST-MUS81 and its fragments GST-MUS81 (1-278), GST-MUS81 (125-278), GST-MUS81 (244-551), and with pGEX-6P-1-MUS81-HisEME1 encoding MUS81-EME1 complex (a kind gift from Dr. Stephen West and Dr. Curtis Harris). Individual bacterial colonies from LB plates were inoculated into 2X TY medium containing ampicillin (0.1 mg/mL). Cells were grown overnight with constant shaking at 37 °C, and protein overexpression was induced by adding 0.1 mM IPTG for 18 h at 16 °C. Corresponding cell pellets were sonicated in CBB buffer supplemented with 200 mM KCl, 0.01% NP40, 1 mM DTT, and a cocktail of protease inhibitors (aprotinin, chymostatin, leupeptin, pepstatin A and benzamide hydrochloride at 5 µg/mL each). The crude extract was clarified by ultracentrifugation at 100,000x g for 1 h at 4 °C, and the supernatant was mixed with 2 ml of Glutathione Sepharose 4 Fast Flow overnight at 4 °C. The matrix was washed with buffer 1X K containing 150 mM KCl, 0.01% NP40, and 1 mM DTT. The bound protein was eluted in two steps with 150 and 250 mM glutathione in buffer 1X K containing 150 mM KCl, 0.01% NP40, and 1 mM DTT. The peak fractions were loaded onto the Heparin column (1 mL) and eluted using a KCl gradient (100–1000 mM) in buffer 1X K. After the Heparin column, GST-MUS81, GST-MUS81 (1-278), GST-MUS81 (125-278), and GST-MUS81 (244-551) fragments were concentrated using VivaSpin centricon. Pooled fractions containing GST-MUS81-EME1 complex were further purified on a MonoS column (0.5 mL) and eluted using a KCl gradient (100-1000 mM) in buffer 1X K. Purified MUS81-EME1 complex was concentrated using VivaSpin centricon and stored in small aliquots at -80 °C.

## In vitro pull-down assay

Purified GST-MUS81-EME1, GST-MUS81, and GST-MUS81 fragments (5 µg each) were incubated with MBP-tagged RECQ4 fragments (5 µg each) and Glutathione Sepharose 4 Fast Flow beads in 1X K buffer containing 150 mM KCl, 0.01% NP40, and 1 mM DTT for 30 min at 4 °C. Flow-through fractions were collected, and the beads were washed three times with the same buffer. SDS Laemmli buffer was added to flow-through and bead fractions, and the samples were analysed by SDS-PAGE on a 10% gel, followed by Coomassie blue staining.

## Nuclease and targeting assay

GST-MUS81-EME1 (0.5 nM) was incubated with RECQ1, BLM, RECQ5, RECQ4 or its various truncations (5, 25 and 50 nM) in 1X ME buffer (50 mM Tris-HCl (pH 7.5), 1 mM DTT, 5 mM MgCl2) and 6 nM 3'-flap DNA substrate for 20 min at 37 °C. BLM and RECCQ1 proteins were provided by Patrick Sung and Alessandro Vindigni respectively. RECQ5 protein was purified as described previously[70]. Following incubation, each reaction was incubated with 0.1% SDS and 500 µg/mL of proteinase K for 5 min at 37 °C for deproteinization. The reactions were resolved on 10% native polyacrylamide gels at 110 V for 45 min, scanned by Image Reader FLA-9000, and analysed by Multi Gauge V3.2 software. For the targeting time course experiment, 25 nM RECQ4 (1-400) was first preincubated in the presence of 1X ME buffer and 6 nM DNA 3'-flap substrate for 10 min at 37 °C. MUS81-EME1 (0.5 nM) was then added, and the mixture was incubated for 5, 10, 15 and 20 min at 37 °C. Corresponding reactions were analysed on 10% native polyacrylamide gel as described above. The gels were scanned with Image Reader FLA-9000 Starion (Fujifilm) or Amersham™ Typhoon™ RGB (Cytiva). The product formation was calculated using MultiGauge software (Fujifilm) as the percentage of the sum of the fluorescent signals (area under the peaks) from the product and the uncleaved substrate.

## Preparation of 3'flap DNA substrate

Synthetic oligonucleotides were purchased from Eurofins Genomics. Equimolar amounts of the corresponding oligonucleotides (Supplementary Data 1) were mixed in hybridising buffer (50 mM Tris, 100 mM NaCl, and 3 mM MgCl$_2$), heated to 75 °C for 3 min and cooled slowly to room temperature for annealing. The substrates were then purified by HPLC using a 1-mL Mono Q column (GE Healthcare Life Sciences) and a 20 mL gradient in 10 mM Tris buffer containing up to 1 M NaCl. The purity was checked on native PAGE. The corresponding fractions were then concentrated on a Vivaspin Concentrator 5000 MWCO and washed with buffer W (25 mM Tris and 3 mM MgCl$_2$). The concentrations were determined using the absorbance at 260 nm and the corresponding molar extinction coefficients.

## Electrophoretic mobility shift assay (EMSA)

RECQ4 (1-400) and RECQ4 (1-400) Δ3 were incubated with 3 nM FITC-labelled HJ DNA substrate in a reaction buffer containing 30 mM Tris [pH 7.5], 1 mM DTT, 100 mM KCl, and 100 µg/mL BSA at 37 °C for 20 min. Following incubation, the reaction mixtures were resolved on 7.5% native polyacrylamide gel at 110 V for 45 min. The final gels were subsequently scanned by an Image Reader FLA-9000 Starion (Fujifilm).

## Statistical analysis

All graphs and statistical analyses were performed using GraphPad Prism software version 7 (GraphPad Software).

## Reporting summary

Further information on research design is available in the Nature Portfolio Reporting Summary linked to this article.

## Data availability

All data supporting the findings of this study are available within the paper and its Supplementary Information. Requests for any additional data or materials should be addressed to the corresponding author. Source data are provided with this paper.

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

## Acknowledgements

We would like to thank F. Nikulenkov for his assistance with data analysis; J. Cibulka for purification of RECQ4(WT) from insect cells and his help with data analysis; K.L. Chan for training and support in setting up the UFB staining technique; S. Narasimhan and A. Jekabsons for their help in running AlphaFold and generating images using UCSF Chimera; Biorender.com for creating schematics images; Patrick Sung for sharing DNA constructs encoding RECQ4 construct and providing BLM protein; Alessandro Vindigni for the RECQ1 protein; Curtis Harris for sharing DNA constructs containing MUS81 fragments; Tomas Loja, Genomics and the CF Bioinformatics supported by the NCMG research infrastructure (LM2023067 funded by MEYS CR) for their support with obtaining scientific FACS data presented in this paper; CIISB, Instruct-CZ Centre of Instruct-ERIC EU consortium, funded by MEYS CR infrastructure project LM2018127, for their support of the measurements of mass spectrometry at the CEITEC Proteomics Core Facility, and all members of the Krejci lab for their discussions and comments on the manuscript. This work was supported by the Czech Science Foundation (21-22593X), and the Wellcome Trust collaborative grant 206292/E/17/Z. H.P.-S. was supported by the Czech Science Foundation Junior Star (grant no. 22-20303 M), the European Union's Horizon 2022 Widera Talent program

(ERA grant agreement no. 101090292), and EMBO Installation Grant (grant no. IG-5689-2024). R.A. was supported by a fellowship offered to foreign students enroled in the Ph.D. programme by JCMM.

## Author contributions

L.K. designed the study. R.A. generated stable cell lines, performed immunofluorescence and UFB staining along with analysis, micronuclei and bulky bridges detection, live cell imaging, ex vivo immunoprecipitation assay and RECQ4 (RTS-CA) protein purification; H.P-S. performed protein purifications, in vitro nuclease, pull-down, EMSA assays, and QIBC analysis; V.M. and M.Z. performed a nuclease assay with MUS81/EME2 complex, RECQ4 (RTS-CA) protein, and prepared DNA substrates; J.P. and M.B. performed the MUS81 foci analysis by IF; Z.H. performed initial in vitro observations uncovering the cooperation of RECQ4 with the MUS81-EME1 complex. R.A., H.P-S., and L.K analysed the data. R.A. and L.K wrote the manuscript. All authors commented on the manuscript.

## Competing interests

The authors declare no competing interests.
