## [Transparent Peer Review file · Nature Communications]

RECQ4-MUS81 interaction contributes to telomere maintenance with implications to Rothmund-Thomson syndrome

Corresponding Author: Dr Lumir Krejci

Version 0:

Reviewer comments:

Reviewer #1

(Remarks to the Author)

In this manuscript entitled "RECQ4-MUS81 interaction safeguards hard-to-replicate regions to prevent genome instability and Rothmund-Thomson syndrome" by Ashraf, R. et al, the authors aimed to determine if RECQ4 modulates MUS81 function. The authors found that RECQ4 physically interacts with MUS81 and stimulates MUS81 endonuclease activity in vitro. The authors show that loss RECQ4 results in chromosomal segregation defects, including formation of increased micronuclei and ultrafine bridges. The authors provide evidence that RECQ4/MUS81 process replication/recombination intermediates, in particular in ALT-positive cells. Finally, the authors show that a mutation in RECQ4 that results in RTS is defective in interacting with MUS81 and in the processing of ultrafine bridges. Overall, the experiments in the manuscript were well controlled and provide evidence supporting much of the conclusions. However, I do think a number of points need to be addressed before I recommend publication in Nature Communications, which I have listed below.

1. A previous study (PMID: 30718377) studied the function of RQ4 in DNA replication/mitosis and found that RQ4 is required for chromosome alignment and mitotic progression. I believe the authors need to mention and discuss this study in their manuscript.
2. Figure 1 and Sup. Figure 1. The authors identify a segment of RQ4 that mediates its interaction with MUS81 (aa 322-400) and using amino acid alignment of RQ4 homologues they putatively identified the amino acids (Y353, V354, and R355) responsible for the RQ4-MUS81 interaction. The authors' biochemical studies clearly shows that deletion ($\Delta 3$) of these three amino acids disrupts the RQ4-MUS81 interaction. As the authors predict these are a separation of function amino acids, it would be helpful if the authors show that deletion of these three amino acids does not result in misfolding of the protein. AlphaFold predictions are fine since the authors used it later in the paper. Also, this would be supported by experiment(s) that shows that other RQ4 interaction(s) and/or functions are retained in $\Delta 3$ complemented cells outside of its function with MUS81.
3. The immunoblotting in Figure 2A are of poor quality and I recommend new immunoblots for this figure.
4. In Figure 2C, the authors test the interaction between RQ4 and MUS81 in different phases of the cell cycle. The authors state that the "interaction was mainly in S phase and partially in G2/M phase" but the data shows interaction in G1 phase, which appears to be higher than in G2/M. Please explain.
5. The authors show that RQ4-MUS81 functions in cancer cells that use ALT for telomere maintenance (Fig. 4) but then tries to correlate the RQ4-MUS81 function with RTS development (Fig. 6) in which the patient fibroblasts are non-cancerous. How does the ALT function associate with RTS development since ALT maintenance is not going to play a role in RTS? Does RQ4 modulate MUS81 in non-cancerous cells?
6. RQ4 mutations found in RTS patients results in a loss of multiple RQ4-dependent functions; therefore, it is likely the authors are overstating in the title that the RQ4-MUS81 interactions "prevents RTS". How can the authors differentiate these functions from the other lost functions in the RQ4 mutations found in RTS patients?

(Remarks to the Author)

In the manuscript 'RECQ4-MUS81 interaction safeguards hard-to-replicate regions to prevent genome instability and Rothmund-Thomson syndrome' (NCOMMS-23-54424), Ashraf et al investigated whether the structure-specific endonuclease MUS81-EME1 interacts with and could be stimulated by RECQ4. They then assessed whether the loss of this interaction could have detrimental consequences in human cells. Although the initial in vitro analysis of RECQ4 and MUS81-EME1 is logical and convincing (although with some weaknesses; see below), the follow-up analysis in human cells with over expression of EGFP tagged RECQ4-WT or the RECQ4- Δ 3 constructs has significant weaknesses in experimental design and interpretation of the results. Furthermore, the text needs to be improved, as the quality of the English is very variable, and is quite poor in parts.

Specific comments are listed below:

A, Introduction section:

- The concept of 'replication stress' was not properly introduced. For example, 'difficult-to-replicate' regions including telomeres, centromeres, and common fragile sites do not generally experience extensive replication problems until agents such as aphidicolin are used to alter their replication timing [1]. Also, it would be helpful to make the Introduction more accessible to non-specialists rather than being focused on the enzymes under study.

B, Results of 'RECQ4 stimulates MUS81 endonuclease activity' (Figures 1, S1):

- How were the purified fragments of RECQ4 verified? There is a need for western blot or mass spectrometry analysis in the purification steps.
- Western blot analysis also needs to be provided for the RECQ1, BLM, RECQ5, and RECQ4 proteins.
- The information of the sequence and length of 3'-flap DNA substrate is missing.
- Because the RECQ4-353-355 motif is highly conserved, in the interaction assays these three residues should be changed to amino acids with a similar hydrophobicity (and geometry ideally), so that the 3D structure of the mutant protein is not disrupted. Full deletion of the residues is a rather crude way to analyze their significance. Therefore, the analysis with RECQ4- Δ 3 construct needs to be performed with a construct in which these three residues are substituted, but not deleted.

C, Results of 'RECQ4 interacts with MUS81 ex vivo and safeguards the proper segregation of chromosomes' (Figures 2, S2, S3)

- To determine the timing of RECQ4 and MUS81 interaction in cell cycle, the authors synchronized cells and then performed an immunoprecipitation with GFP to pull-down MUS81 with the overexpression of RECQ4- Δ 3. The cells were apparently not well-synchronized, as judged by the data in Figure S2C. At best, there was an enrichment in the different phases. In the Methods section it was stated that: 'Cells were seeded in 100 mm dishes and subjected to a double Thymidine (2 mM thymidine, T1895, Sigma-Aldrich) block. Between the two blocks, consisting of 18 hours, the cells were released for 8 hours into fresh media. After that, an aliquot of G1/S synchronised cells was taken, or cells were released into fresh media for 6 hours further, and then cells were harvested (S phase). For G2/M and M phases, cells were treated with RO3306 (SML0569, Sigma-Aldrich) at a concentration of 9 μ M for 20 hours. After 20 hours, cells were harvested (G2/M synchronised cells) or further released in fresh media with 100 nM Nocodazole (M1404, Sigma-Aldrich) for 2 hours and then harvested by mitotic shake-off (M phase).' Because RO3306 is a CDK1 inhibitor, a 20-hour treatment of the cells would inevitably have a lot of side effects, particularly on replication origin firing in S phase. If this drug is to be used purely for synchronization of cells, it should be added for only 1-3 hours after cells have already progressed through most of S phase. Potentially, this perturbation of DNA replication timing would affect not only the interaction between RECQ4 and MUS81, but also the chromosome phenotypes analyzed in mitosis.
- In Figure S3, the authors quantified mitotic events by measuring the progression time between prometaphase to anaphase in cells with or without RECQ4 depletion, and the frequency of misalignment of chromosomes in metaphase. However, it is not clear how 'misalignment' was quantified. Was this by the number of non-aligned chromosomes for each metaphase cell, or by the thickness of the metaphase plate? Judging by Figure S3B, the 'proper alignment' and 'misalignment' samples do not appear to be very different. The timepoint at which the misalignment was measured is also very important. How was this chosen and what was the time chosen relative to? Generally, in such analyses, time 0 would be defined by a key mitotic event – most commonly, the timing of nuclear envelope breakdown. More rigorous analysis of this phenotype is required to assess if there is a true delay in the progression of mitosis in RECQ4 depleted cells.
- In Figure 2D, because there could be more than one micronucleus in each set of binucleated cells, the quantification of micronuclei needs to be presented as the average number of micronuclei per binucleated cells to be more accurate. The same quantification should be applied to quantifications of anaphase DNA bulky bridges, and UFBs.

D, Results of 'RECQ4/MUS81 interaction is essential for the processing of ultrafine bridges' (Figure 3)

- It is well-established that while centromeric UFBs might be part of a normal mitosis and be composed of catenated double-stranded DNAs, the other type of UFBs including common fragile site associated UFBs, telomere associated UFBs, rDNA associated UFBs, or homologous recombination generated UFBs, generally arise from pathological conditions, e.g., treatment with APH, oncogene activation or DSB induction (cisplatin, camptothecin) [2-8]. Therefore, it is very surprising to see that these different types of UFBs were detected in untreated cells (Figure 3A, B). In addition, the procedure for detecting UFBs was different from most published protocols (for example [6]), and it would be helpful for the authors to state why they used this protocol.
- In Figure 3C-D, it is not clear what the average number of UFBs is per cell following each treatment, and there were no IF images to illustrate the quality of the UFBs detected. It is therefore impossible to draw a firm conclusion from these data.
- Even if the data presented here were correct, these data could not support the statement that 'RECQ4/MUS81 interaction is

essential for the processing of ultrafine bridges'. How do the authors define 'processing' here. Are the authors suggesting that these enzymes resolve bridges in some way, because this wasn't studied as far as I could tell. Presumably, to process UFBs, these proteins would have to bind to the UFBs during mitosis (anaphase). Curiously though, it wasn't shown that either protein is found on UFBs, and it was shown that RECQ4 and MUS81 interact mainly in S phase and not in M phase. It would be more logical to conclude that the interaction between RECQ4 and MUS81 might in some way remove DNA structures in interphase (S phase), which can lead to reduced chance of forming UFBs in mitosis.

• In summary, the data presented do not fully support the conclusions of this section.

E, Results of 'ALT-dependent MUS81 foci requires RECQ4' (Figure 4)

• The authors used QIBC to analyze MUS81 foci in ALT or non-ALT cells with or without RECQ4 expression. It was not clear how this analysis was performed and how many cells were analyzed for each condition, as it was not described in the Methods. Only 2 ALT cell lines were analyzed, which clearly isn't sufficient to make general conclusions about a role for RECQ4/MUS81 in telomere maintenance pathways. This analysis is too superficial.

F, Results of 'Patient-derived RECQ4 mutation phenocopies interaction deficient mutant' (Figures 6, S1F, S2, S5, S6)

• In Figure S5A, clearly, there is a band of a similar size to that of the normal RECQ4 protein in the RTS-CU and RTS-CA samples. It is puzzling, therefore, why the authors stated: 'We found reduced and almost absent levels of full-length RECQ4 in RTS-CU and RTS-CA fibroblasts, respectively (Figure S5A) compared to normal fibroblasts.'

• It is not clear how representative of the RTS-CA mutation is in the RTS patients. This needs to be introduced and discussed. If the data presented in this part were to be correct, then the title of this section should be altered to something like 'A patient derived RECQ4 mutation phenocopies an interaction deficient mutant'.

G, regarding the 'Discussion'

• This part suffers from a lack of a summary of the key findings and many over statements or speculations. One example is: 'We investigated the biological role of RECQ4 and its interaction with MUS81 in the resolution of replication/recombination DNA intermediates' (lines 279-281). This is not correct as this study has only touched upon the effect of RECQ4 depletion on the frequency of UFBs in mitotic cells. Another example is found on lines 297-308, in which incorrect statements were made on the MUS81 foci and their location to 'hard to replicate' regions, which has not been verified in this study.

In conclusion, although the authors have described a potentially interesting interaction between RECQ4 and MUS81, the data presented here cannot support many of the conclusions stated due to inappropriate experimental design and data of limited quality. In my opinion, this manuscript is not ready for publication in Nature Communications.

References

1. Zeman, M.K. and K.A. Cimprich, Causes and consequences of replication stress. *Nat Cell Biol*, 2014. 16(1): p. 2-9.
2. Chan, K.L., T. Palmal-Pallag, S. Ying, and I.D. Hickson, Replication stress induces sister-chromatid bridging at fragile site loci in mitosis. *Nat Cell Biol*, 2009. 11(6): p. 753-60.
3. Lukas, C., V. Savic, S. Bekker-Jensen, C. Doil, B. Neumann, R.S. Pedersen, M. Grofte, K.L. Chan, I.D. Hickson, J. Bartek, and J. Lukas, 53BP1 nuclear bodies form around DNA lesions generated by mitotic transmission of chromosomes under replication stress. *Nat Cell Biol*, 2011. 13(3): p. 243-53.
4. Liu, Y., C.F. Nielsen, Q. Yao, and I.D. Hickson, The origins and processing of ultra fine anaphase DNA bridges. *Curr Opin Genet Dev*, 2014. 26: p. 1-5.
5. Nielsen, C.F., D. Huttner, A.H. Bizard, S. Hirano, T.N. Li, T. Palmal-Pallag, V.A. Bjerregaard, Y. Liu, E.A. Nigg, L.H. Wang, and I.D. Hickson, PICH promotes sister chromatid disjunction and co-operates with topoisomerase II in mitosis. *Nat Commun*, 2015. 6: p. 8962.
6. Bizard, A.H., C.F. Nielsen, and I.D. Hickson, Detection of Ultrafine Anaphase Bridges. *Methods Mol Biol*, 2018. 1672: p. 495-508.
7. Chan, Y.W., K. Fugger, and S.C. West, Unresolved recombination intermediates lead to ultra-fine anaphase bridges, chromosome breaks and aberrations. *Nat Cell Biol*, 2018. 20(1): p. 92-103.
8. Ozer, O., R. Bhowmick, Y. Liu, and I.D. Hickson, Human cancer cells utilize mitotic DNA synthesis to resist replication stress at telomeres regardless of their telomere maintenance mechanism. *Oncotarget*, 2018. 9(22): p. 15836-15846.

Reviewer #3

(Remarks to the Author)

In this manuscript Ashraf et al show that the RECQ4 helicase interacts with MUS81 and activates its nuclease activity and that this plays a role in chromosomal segregation. This is an interesting study that describes an unidentified role of RECQ4. Nevertheless, I have major concerns with the manuscript, and at its current state there are several issues that need to be addressed in order for this work to be published in Nature Communications.

Major comments:

1) According to Fig 1E, RECQ4 (1-400) and RECQ4 (WT) have similar effect on cleavage of the 3'-flap DNA substrate, which is close to 100% product formation. I understand that the %product formation is the % ratio of the bottom band to the sum of the top and bottom bands. If indeed this is the case, the representative images of Fig 1 do not depict that, as the RECQ4 (1-400) is clearly cleaving less of the substrate than the RECQ4 (WT) at 5, 25 and 50 nM. The same also applies for RECQ4 and RECQ5 samples in Fig 1A. The authors need to clarify what they measure as % product formation in the

materials and methods and also make sure that their measurements are correct and that the representative images support their plots.

2) I am not convinced that the statement 'Our results revealed that the interaction between RECQ4 and MUS81 was observed mainly in the S phase and partially also in the G2/M phase (Figures 2C and S2C)' is completely true. The reasons are:

- i) The IP experiments in Figure 2A and 2C are lacking an IgG control. The authors need to repeat their IP experiments and include this.
- ii) The MUS81 band in the G2/M phase lane of the IP:GFP is very small to support a significant interaction and the authors rightly state that there is a partial interaction. Since, the M phase lane of the IP:GFP is clearly underloaded I am wondering if it was indeed properly equally loaded, whether it would produce a MUS81 band similarly small as the G2/M phase sample. If that is true, then the ensuing conclusion would change. The authors need to produce an IP that the samples are equally loaded.
- iii) What the authors considered to be the G1/S phase sample is mostly G1. This changes the argument that the interaction happens primarily in S phase. The authors need to rewrite this part of the text.
- iv) The authors could also immunoblot for cell cycle markers such as CDT1 (specific to G1), CYCLIN A (specific to S/G2) and CENPF (specific to G2) to further support the successful synchronisation of their samples. The authors need to address these concerns.

3) It would be interesting to examine how the siRNA-RECQ4-induced micronuclei and anaphase bridges are affected by simultaneous overexpression of WT-RECQ4 and RECQ4- Δ 3.

4) I find the association between RECQ4 and ALT to be very weak.

The authors state '...we also compared UFBs in the ALT-negative HT1080 cell line and observed almost no difference in fold changes for CUB and TUB (Figure 3F-G)'. This statement is completely wrong and misleading to the reader. The presented plots (Figures 3F and 3G) to support this statement clearly show that knockdown of RECQ4 by siRNA induces an increase of CUBs and TUBs that the authors show to be statistically significant. The authors need to rewrite this section of the manuscript and address the ensuing conclusions about effect of RECQ4 on cells that use ALT, that they deduce based on their wrong statement.

In addition, if the authors want to explore a correlation between their RECQ4 findings and ALT status, they should test several ALT positive and telomerase positive cell lines. Using just U2OS and HT1080 is not enough to reach any legitimate conclusions.

5) In order for the authors to investigate a possible role of RECQ4 and ALT phenotype they need to test the effect of RECQ4 knockdown with and without simultaneous overexpression of WT-RECQ4 and RECQ4- Δ 3 in various ALT-related phenotypes (ALT-associated PML bodies, presence of extrachromosomal telomeric DNA circles and telomeric sister chromatid exchanges) in ALT-positive as well as in telomerase-positive cell lines

6) Based on the text, I deduce that in Figure 4C, RECQ4 has been deleted/knocked down. If this is indeed true, then it needs to be clearly stated in the text. If that is not the case, then I do not see proof for a correlation between RECQ4, MUS81 and ALT status.

In addition, as a control, the authors should show that MUS81 foci can indeed successfully form in response to DSBs in the HT1080 and LM216T cell lines. In this way they can be sure that the lack of MUS81 foci in these cell lines is not due to a particular defect these cell lines might have.

7) The IP experiments in Figure 6D are lacking an IgG control. The authors need to include this.

Also, the immunoblots for GFP should be the same for all samples and not cropped for the GFP control and the WT-RECQ4-GFP, RECQ4- Δ 3-GFP and RTS-CA-GFP samples. In this way, the difference in the molecular weight heights between these samples is clearly shown.

8) The authors should test the effect of the RECQ4-RTS-CA mutation on the MUS81-EME1 endonuclease activity.

9) In order to provide clinical significance, the authors engineered U2OS cells carrying the RECQ4-RTS-CA mutation and performed several experiments to test i) the interaction between RECQ4-MUS81 (Fig 6D), ii) FUBs (Fig 6E), iii) CUBs (Fig 6F), iv) TUBs (Fig 6G) and v) MUS81 foci (Fig 6H). It would be much more informative and clinically relevant if the authors perform these experiments directly on the patient RTS-CU and 252 RTS-CA fibroblasts (and normal fibroblasts as control) that they have acquired, in the same way they did in experiments in Figures 6A-C. On top of these experiments, I would also ask the authors to assess bulky bridges in the patient-derived RTS fibroblasts.

10) I am very confused with the immunoblot of Fig S2B.

In the text the authors claim: '...we established stable cell lines allowing simultaneous depletion of endogenous RECQ4 and expression of siRNA-resistant exogenous EGFP-tagged RECQ4-WT and RECQ4- Δ 3 in Flp-In T-REx U2OS cells (Figure S2A-B)'.

And in the figure legend one reads '(B) Western blot confirming the expression of the various constructs EGFP, EGFP-RECQ4-WT, EGFP-RECQ4- Δ 3, and EGFP1125 RECQ4-RTS-CA in Flp-In T-REx U2OS cells, alongside endogenous RECQ4 knockdown using SiRecQ4 or SiControl shown for reference.

According to the information in Materials and Methods the engineered U2OS express pAIO plasmids that simultaneously express an shRNA against RECQ4 and siRNA-resistant RECQ4 (WT, Δ 3 or RTS-CA).

This is not supported by the western blot in Figure S2B, where you can clearly see that in the WT and Δ 3 lanes, the

endogenous RECQ4 is present, but in the RTS-CA lane, endogenous REC4 is absent.

Also, the authors claim that on top of the expression of the shRNA against RECQ4 of the pAIO system the authors transfect the cells an siRNA against RECQ4 to achieve knockdown. How do the authors control that this siRNA will not knockdown the various RECQ4 plasmids (WT, $\Delta 3$ or RTS-CA) that are simultaneously expressed in the pAIO system that are only siRNA-resistant but not shRNA-resistant?

The authors need to clarify what do the engineered U2OS cells express, how RECQ4 knockdown is achieved and what is shown in the immunoblot of Fig S2B.

Minor comments:

1) The authors should make sure to have at least 3 independent biological repeats in all of their experiments and also to add statistical significance tests in them.

2) Figure 5B does not provide any substantial information to support the plot in Figure 4A. The only thing the readers see are some cells in anaphase, but none type of bridges (bulky or not) are visible. The authors should provide representative magnified images of what they consider to be bulky bridges.

Version 1:

Reviewer comments:

Reviewer #1

(Remarks to the Author)

In this revision entitled "RECQ4-MUS81 interaction contributes to safeguarding hard-to-replicate regions to alleviate genome instability and Rothmund-Thomson syndrome development" by Ashraf, R et al, the authors appropriately responded to my comments. My only remaining issue is the interaction of RQ4 and MUS81 shown in Fig. 2C is quite weak and the MUS81 blot is of low quality. I would only recommend a better MUS81 immunoblot. Regardless, I now recommend this paper for publication in Nature Communications.

Reviewer #2

(Remarks to the Author)

This revised manuscript is substantially improved. I am happy with most of the experimental and textual revisions. However, there are a few points, listed below, need to be addressed before this manuscript could be published:

1) The data in Figure 1A are crucial for the demonstration that RECQ4 gives the highest stimulation of MUS81-EME1 amongst the RECQ helicases. Therefore, it is important for the authors to provide evidence to show that the proteins used in this assay are equally pure and equally active as DNA helicases. The stimulation should be analyzed using an equivalent number of 'unwinding units' for each helicase. Also, the authors have given the references for the source of these proteins in the rebuttal letter, and it is clear now that the proteins used in this assay are a mixture of those obtained externally and those produced in-house. This makes it even more important to show that the purity of these proteins is similar, and that they have not been affected during transport or storage etc. In addition, the references for the source of the proteins used need to be provided in the manuscript text as well.

2) In the current Figures 4 and S4, the authors have included a panel of ALT+ve and ATL-ve cell lines in the MUS81 focus study. Two issues with this new analysis need to be addressed:

a) The references or sources, and culture conditions for all of the cell lines used need to be included in the Materials & Methods (currently, this information is only shown for some of the cell lines).

b) In Fig S4A-B, the IF staining of MUS81 analysis has a very strong background, and it is actually very difficult to separate the foci from the background. We have to take on faith that the foci amongst the mass of background staining are really the MUS81 foci? More strangely, the MUS81 staining in HT1080 cells was mostly outside of the nucleus. There is a concern that this IF did not work well in this assay. For example, the MUS81 IF images in Figure S4F look quite clear and distinct. The authors need to address the discrepancies between different experiments and provide quantification of MUS81 foci in all the cell lines analyzed in Fig S4A-B. In addition, in Figure S4A, a quantification of MUS81 foci in all the cell lines analyzed need to be provided.

3) The data shown in this manuscript support the notion that RECQ4-MUS81 interaction contributes to telomere maintenance but are not so strong with regard to 'other hard-to replicate regions'. Also, it is an overstatement to say that this interaction contributes to 'Rothmund-Thomson syndrome development'. I would suggest change the title to 'RECQ4-MUS81 interaction contributes to telomere maintenance in human cells', so that it reflects more precisely on the content and conclusion drawn from the data. This means the Abstract and Discussion needs to be modified somewhat also.

Minor point: the table of a list of the mutation found in Rothmund-Thomson syndrome patients are very helpful for the readers to have a sense of the scale of the mutation spectrum of RECQ4 in these patients. This table could be included in the Supplementary data in the manuscript.

Reviewer #3

(Remarks to the Author)

This revised version is a great improvement from the initial submission. Nevertheless, there are a few of my initial concerns that have not been addressed adequately, in order to recommend this revised manuscript for publication in Nature Communications.

Major comments:

1) According to Fig 1E, RECQ4 (1-400) and RECQ4 (WT) have similar effect on cleavage of the 3'-flap DNA substrate, which is close to 100% product formation. I understand that the %product formation is the % ratio of the bottom band to the sum of the top and bottom bands. If indeed this is the case, the representative images of Fig 1 do not depict that, as the RECQ4 (1-400) is clearly cleaving less of the substrate than the RECQ4 (WT) at 5, 25 and 50 nM. The same also applies for RECQ4 and RECQ5 samples in Fig 1A. The authors need to clarify what they measure as % product formation in the materials and methods and also make sure that their measurements are correct and that the representative images support their plots.

Response: We apologise for any confusion and lack of clarity. We have now included other representative images for RECQ5 in Figure 1A and RECQ4 (WT) as well as RECQ4 (1-400) in Figure 1D) in the revised version of the manuscript. We have also provided a detailed description of the quantification of % product formation in the corresponding Material and Method section.

I thank the authors for addressing my comment.

2) I am not convinced that the statement 'Our results revealed that the interaction between RECQ4 and MUS81 was observed mainly in the S phase and partially also in the G2/M phase (Figures 2C and S2C).' is completely true. The reasons are: i) The IP experiments in Figure 2A and 2C are lacking an IgG control. The authors need to repeat their IP experiments and include this.

Response: We have repeated the experiment and included a bead control, as we used anti-GFP trap beads for co-immunoprecipitation. The new data confirm our previous observations and are now included in the revised manuscript (Figures 2A and C, S2D and 6D).

I thank the authors for addressing my comment.

ii) The MUS81 band in the G2/M phase lane of the IP:GFP is very small to support a significant interaction and the authors rightly state that there is a partial interaction. Since, the M phase lane of the IP:GFP is clearly underloaded I am wondering if it was indeed properly equally loaded, whether it would produce a MUS81 band similarly small as the G2/M phase sample. If that is true, then the ensuing conclusion would change. The authors need to produce an IP that the samples are equally loaded.

Response: We appreciate the reviewer's comment, we have repeated the experiment to ensure equal loading and modified the statement in the results section. Our new results confirmed that the interaction between RECQ4 and MUS81 was predominantly observed in the G1 and S phases, with partial interaction observed in the G2/M phase (Figures 2C and S2E).

I thank the authors for addressing my comment.

iii) What the authors considered to be the G1/S phase sample is mostly G1. This changes the argument that the interaction happens primarily in S phase. The authors need to rewrite this part of the text.

Response: We acknowledge the reviewer's concern and have revised the corresponding statement in the results section. The corrected statement now reflects that the interaction was predominantly observed in the G1 and S phases, with partial interaction observed in the G2/M phase (see page 4, lines 173-175).

I thank the authors for addressing my comment.

iv) The authors could also immunoblot for cell cycle markers such as CDT1 (specific to G1), CYCLIN A (specific to S/G2) and CENPF (specific to G2) to further support the successful synchronisation of their samples. The authors need to address these concerns.

Response: We have repeated the synchronisation experiment as described for reviewer #2 (page 5, Figure 2C), included new FACS analysis (Figure S2E), as well as western blot for CENPF (Figure 2C) in the revised version of the manuscript.

I thank the authors for addressing my comment.

3) It would be interesting to examine how the siRNA-RECQ4-induced micronuclei and anaphase bridges are affected by simultaneous overexpression of WT-RECQ4 and RECQ4-Δ3.

Response: We thank the reviewer for this suggestion. In response, we extended our study using Flp-In T-REx U2OS cell line to examine the effect of expressing exogenous RECQ4-Δ3 in the presence or absence of endogenous wild-type RECQ4, by employing siControl or siRECQ4, respectively. Our data indicate that presence of wild-type RECQ4 effectively compensate for the segregation defects observed in cell expressing only RECQ4-Δ3, specifically in terms of the average number of bulky bridges and UFBs (Figure S5D-E). These findings are consistent with results from fibroblasts derived from clinically unaffected individual, which expressed higher level of wild-type RECQ4 protein compared to clinically affected fibroblasts (Figure S5A-C).

The authors should make it clearer that in the plots in Figures S5D-E, that the grey bars are representing cells

overexpressing the wild-type RECQ4 and the brown ones are representing cells overexpressing RECQ4-Δ3. The way the plot is depicted and the explanation in the figure legend is confusing.

4) I find the association between RECQ4 and ALT to be very weak. The authors state '...we also compared UFBs in the ALT-negative HT1080 cell line and observed almost no difference in fold changes for CUB and TUB (Figure 3F-G)'. This statement is completely wrong and misleading to the reader. The presented plots (Figures 3F and 3G) to support this statement clearly show that knockdown of RECQL4 by siRNA induces an increase of CUBs and TUBs that the authors show to be statistically significant. The authors need to rewrite this section of the manuscript and address the ensuing conclusions about effect of RECQ4 on cells that use ALT, that they deduce based on their wrong statement. In addition, if the authors want to explore a correlation between their RECQ4 findings and ALT status, they should test several ALT positive and telomerase positive cell lines. Using just U2OS and HT1080 is not enough to reach any legitimate conclusions. Response: We apologize for the unclear and misleading statement regarding the association between RECQ4 and ALT, which we have now corrected in the revised manuscript. We have rewritten this section to accurately reflect the statistically significant increase in both CUBs and TUBs upon RECQ4 knockdown in the ALT-negative HT1080 cell line (see page 5, lines 203-205; Figures 3F-G and S3J-K). In addition, ALT-positive U2OS cells exhibit higher fold change for TUBs compared to CUBs and FUBs in H1080 (see page 5, lines 205-208; Figures S3D-E, and S3I-K). Additionally, as suggested by the reviewer, we have extended our studies to include a panel of ALT-positive and ALT-negative cell lines. This comprehensive analysis is now included in Figures 4A-D, 4F, 4J-K, S4A-B, and S4F of the revised manuscript. These data provide a more robust basis for our conclusions regarding the effect of RECQ4 on cells that use ALT mechanism.

The authors should provide a western blot of MUS81 for the various ALT+ and telomerase positive cell lines they use in Figure 4A.

The data in Figure 4A must be explained more thoroughly. The authors state that MUS81 foci were completely absent from the telomerase+ cell lines, but this needs to be shown with some sort of analysis. Therefore, the authors should provide MUS81 foci quantification analysis data for the data presented in Figure 4A.

5) In order for the authors to investigate a possible role of RECQ4 and ALT phenotype they need to test the effect of RECQ4 knockdown with and without simultaneous overexpression of WT-RECQ4 and RECQ4-Δ3 in various ALT-related phenotypes (ALT-associated PML bodies, presence of extrachromosomal telomeric DNA circles and telomeric sister chromatid exchanges) in ALT-positive as well as in telomerase-positive cell lines

Response: To address this point, we extended our study cells by transiently transfecting LM216J cells with plasmids expressing RECQ4-WT and RECQ4-Δ3, and subsequently monitored MUS81 foci. Our findings indicate that the expression of RECQ4-WT successfully rescued the decrease in MUS81 foci observed upon RECQ4 knockdown, whereas expression of RECQ4-Δ3 did not (Figure 4F and S4F).

Furthermore, we also investigated the effect of RECQ4 knockdown, both with and without simultaneous expression of RECQ4-WT and RECQ4-Δ3, on the formation of ALT-associated PML bodies (APBs) (Figure 4J-K). These additional experiments, detailed in the revised manuscript, further confirm that RECQ4 depletion impacts APB formation, a defect that can be complemented by RECQ4-WT expression but not by RECQ4-Δ3. Our new data thus support role of RECQ4-MUS81 interaction in ALT-related phenotypes. The ALT-related phenotypes are not detectable in telomerase-positive cells lines.

Since the authors only assessed APBs and not the other ALT-related phenotypes I suggested (presence of extrachromosomal telomeric DNA circles and telomeric sister chromatid exchanges) in only one ALT+ cell line and none telomerase+ cell lines (which would have been the negative control), I suggest they significantly tone down any claims about an RECQ4 having an effect on ALT. Without the experiments I suggested, any definitive association of RECQ4 and ALT is farfetched and misleading.

6) Based on the text, I deduce that in Figure 4C, RECQ4 has been deleted/knocked down. If this is indeed true, then it needs to be clearly stated in the text. If that is not the case, then I do not see proof for a correlation between RECQ4, MUS81 and ALT status.

Response: We apologise for possible unclarity. We have now explicitly stated that status of RECQ4 in these experiments. Figure 4G and the newly included Figure S4A-B demonstrate that MUS81 foci are indeed readily detected in ALT-positive cells while they are absent in ALT-negative cells. In addition, these MUS81 foci, at least in the two tested ALT-positive cell lines (U2OS and LM216J), are dependent on RECQ4 and its interaction with MUS81 (Figure 4E-F). Furthermore, we show that these foci exhibit very high colocalization with telomeres in both U2OS and LM216J cell lines (Figure 4G-I).

As mentioned in a previous comment, the authors should provide a western blot of MUS81, as well as a quantification analysis of MUS81 foci for the various ALT+ and telomerase positive cell lines they use in Figure 4A.

In addition, as a control, the authors should show that MUS81 foci can indeed successfully form in response to DSBs in the HT1080 and LM216T cell lines. In this way they can be sure that the lack of MUS81 foci in these cell lines is not due to a particular defect these cell lines might have.

Response: We have tried to induce MUS81 foci formation in both ALT-positive (U2OS, LM216J) and ALT-negative (HeLa) cells using various agents, including camptothecin, cisplatin, aphidicolin, hydroxyurea, and pyridostatin. However, none of these treatments resulted in the induction of MUS81 foci. There are indeed conflicting reports regarding MUS81 foci formation: some studies have shown an increase in MUS81 foci after replication stress (Chappidi et al 2020, PMID: 31759821; Noumora et al 2007, PMID: 17903171; Saugar et al 2017, PMID: 28813668), while others have reported no increase (Elbakry et al 2021, PMID: 33431668; Deshpande et al 2022, PMID: 36099913 and Panichnantakul et al 2021, PMID: 34139663).

This again makes it important that the authors assess MUS81 levels in these cell lines. Since the authors rely their conclusions on MUS81 foci it is imperative that MUS81 controls have been assessed before making any claims.

7) The IP experiments in Figure 6D are lacking an IgG control. The authors need to include this. Also, the immunoblots for GFP should be the same for all samples and not cropped for the GFP control and the WT-RECQ4-GFP, RECQ4- Δ 3-GFP and RTS-CA-GFP samples. In this way, the difference in the molecular weight heights between these samples is clearly shown.

Response: We appreciate the reviewer's comment regarding the need for appropriate control and the presentation of immunoblot data. To address this, we have repeated the IP experiment, this time including a bead-only control, as we used GFP trap beads for co-immunoprecipitation. The new data confirm our previous conclusion and are now included into the revised manuscript (Figure 6D).

I thank the authors for addressing my comment.

Additionally, we have provided uncropped EGFP immunoblot images (Figure R3) to clearly demonstrate the differences in molecular weight between EGFP control, EGFP-RECQ4-WT, EGFP-RECQ4- Δ 3 and EGFP-RTS-CA samples. For clarity and space limitations in the main figures, we would like to continue to use cropped images, however, the uncropped images for all immunoblots will be uploaded to the publisher site and available to research community.

If the authors want to continue using the cropped GFP western blots, they should at least indicate on the side of the cropped blot what is depicted when they write 'EGFP', otherwise the reader will be confused with two different cropped blots being described as 'EGFP'.

Also, I would kindly ask the authors to include the uncropped GFP western blots in the supplementary data of the manuscript.

8) The authors should test the effect of the RECQ4-RTS-CA mutation on the MUS81-EME1 endonuclease activity.

Response: We have expressed and purified RECQ4-RTS-CA mutant protein and tested its effect on the MUS81-EME1 endonuclease activity. While we observe ability of this mutant to stimulate the activity of MUS81-EME1 endonuclease, however, it indeed shows reproducibly partial defect compared RECQ4-WT. The results of these experiments are now included in the revised manuscript and Figure S4G-H.

I thank the authors for addressing my comment.

9) In order to provide clinical significance, the authors engineered U2OS cells carrying the RECQ4-RTS-CA mutation and performed several experiments to test i) the interaction between RECQ4-MUS81 (Fig 6D), ii) FUBs (Fig 6E), iii) CUBs (Fig 6F), iv) TUBs (Fig 6G) and v) MUS81 foci (Fig 6H). It would be much more informative and clinically relevant if the authors perform these experiments directly on the patient RTS-CU and 252 RTS-CA fibroblasts (and normal fibroblasts as control) that they have acquired, in the same way they did in experiments in Figures 6A-C. On top of these experiments, I would also ask the authors to assess bulky bridges in the patient-derived RTS fibroblasts.

Response: We have conducted the following experiments directly on RTS-CU and RTS-CA fibroblasts: analysis of UFBs (FUB, CUB and TUB in Figures 6A, B, and C, respectively); analysis of micronuclei (Figure S5B); and bulky bridges (Figure S5C). Unfortunately, we could not perform the co-IP experiments in fibroblast as there are no IP-compatible RECQ4 antibodies currently available on the market. Nevertheless, our results reveal increased levels of damage in fibroblast derived from a clinically affected individual (see page 7, lines 282-292). In addition, while we attempted to monitor MUS81 foci in the patient-derived fibroblasts, we were unable to detect spontaneous or DNA damage-induced Mus81 foci in these cells.

I thank the authors for addressing my comment.

10) I am very confused with the immunoblot of Fig S2B. In the text the authors claim: '...we established stable cell lines allowing simultaneous depletion of endogenous RECQ4 and expression of siRNA-resistant exogenous EGFP-tagged RECQ4-WT and RECQ4- Δ 3 in Flp-In T-REx U2OS cells (Figure S2A-B)'. And in the figure legend one reads '(B) Western blot confirming the expression of the various constructs EGFP, EGFP-RECQ4-WT, EGFP-RECQ4- Δ 3, and EGFP1125 RECQ4-RTS-CA in Flp-In T-REx U2OS cells, alongside endogenous RECQ4 knockdown using SiRecQ4 or SiControl shown for reference. According to the information in Materials and Methods the engineered U2OS express pAIO plasmids that simultaneously express an shRNA against RECQ4 and siRNA-resistant RECQ4 (WT, Δ 3 or RTS-CA). This is not supported by the western blot in Figure S2B, where you can clearly see that in the WT and Δ 3 lanes, the endogenous RECQ4 is present, but in the RTS-CA lane, endogenous REC4 is absent. Also, the authors claim that on top of the expression of the shRNA against RECQ4 of the pAIO system the authors transfect the cells an siRNA against RECQ4 to achieve knockdown. How do the authors control that this siRNA will not knockdown the various RECQ4 plasmids (WT, Δ 3 or RTS-CA) that are simultaneously expressed in the pAIO system that are only siRNA-resistant but not shRNA-resistant? The authors need to clarify what do the engineered U2OS cells express, how RECQ4 knockdown is achieved and what is shown in the immunoblot of Fig S2B.

Response: The bands observed in the lanes for EGFP-RECQ4-WT and EGFP-RECQ4- Δ 3 proteins in previous Figure S2B are likely degradation products of these proteins. To confirm this, we repeated the experiment using 6% SDS-PAGE gel for better separation of protein bands, followed by Western blot analysis (Figure S2C). This updated blot confirms that the endogenous RECQ4 band is absent in the siRECQ4, WT, Δ 3, and RTS-CA samples, indicating successful knockdown of endogenous RECQ4. We also provide a western blot using GFP antibodies that show similar multiple bands in cell

expressing exogenous RECQ4 constructs (Figure R4).

We apologise for the lack of clarity describing the expression of engineered U2OS cells. We have modified the text in the Material and Method section to clarify the text. Briefly, upon doxycycline induction, our engineered U2OS cells express both the shRNA targeting endogenous RECQ4 and the exogenous RECQ4 constructs (WT, $\Delta 3$, and RTS-CA). The exogenous RECQ4 sequences are codon optimised to be resistant to the shRNA and siRECQ4 also used for the knockdown to ensure maximal downregulation of endogenous RECQ4.

I thank the authors for addressing my comment.

Minor comments: 1) The authors should make sure to have at least 3 independent biological repeats in all of their experiments and also to add statistical significance tests in them.

Response: We have now included the results of triplicate experiments and their statistical significance for all remaining figures (Figures 5D, 6C).

I thank the authors for addressing my comment.

2) Figure 5B does not provide any substantial information to support the plot in Figure 4A. The only thing the readers see are some cells in anaphase, but none type of bridges (bulky or not) are visible. The authors should provide representative magnified images of what they consider to be bulky bridges.

Response: We apologize for not providing more clear images. We have now included representative magnified images of the bulky bridges in Figure 5B to provide substantial support for the plot in Figure 5A.

I thank the authors for addressing my comment. I would kindly ask that the new provided images are presented at a higher magnification.

Version 2:

Reviewer comments:

Reviewer #2

(Remarks to the Author)

This 2nd revised manuscript by Ashra R. et al has addressed all the three points raised last time.

I am happy with the content and quality of the figures provided, namely Figure S1A, Figure R1, Figure R2 and Figure S5A. I am also happy with the changes in the Title, Abstract, Material/methods and Discussion about following my comment.

The remaining minor modifications I would like to ask for before its publication in Nature Communication are: to include Figures R1 and R2 in the manuscript. This is because, regarding Figure R1, it is essential to show that the proteins used in this study has equal helicase activity and not just by their action in previous studies; regarding Figure R2, it would be an excellent addition to show MUS81 foci changes with siRNA MUS81 KD in a cell line with zoomed images. Both figures will bring more credibility to this manuscript. Please note, after including these two figures in the MS, the text regarding figure orders, Materials/methods, etc will need to be changed accordingly. I am happy to read a final version again.

Reviewer #3

(Remarks to the Author)

This revised version is a great improvement from the previous submission. My concerns have been met and I am happy to recommend this manuscript for publication in Nature Communications.

REVIEWER COMMENTS

Reviewer #1 (Remarks to the Author):

In this manuscript entitled "RECQ4-MUS81 interaction safeguards hard-to-replicate regions to prevent genome instability and Rothmund-Thomson syndrome" by Ashraf, R. et al, the authors aimed to determine if RECQ4 modules MUS81 function. The authors found that RECQ4 physically interacts with MUS81 and stimulates MUS81 endonuclease activity in vitro. The authors show that loss RECQ4 results in chromosomal segregation defects, including formation of increased micronuclei and ultrafine bridges. The authors provide evidence that RECQ4/MUS81 process replication/recombination intermediates, in particular in ALT-positive cells. Finally, the authors show that a mutation in RECQ4 that results in RTS is defective in interacting with MUS81 and the processing of ultrafine bridges. Overall, the experiments in the manuscript were well controlled and provide evidence supporting much of the conclusions. However, I do think a number of points need to be addressed before I recommend publication in Nature Communications, which I have listed below.

Response: We appreciate that the reviewer finds our study well-controlled and supportive of our conclusions. We value the detailed feedback and constructive comments, which helped us to enhance our manuscript. Below, we address each point raised:

1. A previous study (PMID: 30718377) studied the function of RQ4 in DNA replication/mitosis and found that RQ4 is required for chromosome alignment and mitotic progression. I believe the authors need to mention and discuss this study in their manuscript.

Response: We apologise for omitting this study and have now included and discussed it in the revised version of the manuscript (see page 4, lines 177-180 and page 8, lines 337-338).

2. Figure 1 and Sup. Figure 1. The authors identify a segment of RQ4 that mediates its interaction with MUS81 (aa 322-400) and using amino acid alignment of RQ4 homologues they putatively identified the amino acids (Y353, V354, and R355) responsible for the RQ4-MUS81 interaction. The authors' biochemical studies clearly shows that deletion ($\Delta 3$) of these three amino acids disrupts the RQ4-MUS81 interaction. As the authors predict these are a separation of function amino acids, it would be helpful if the authors show that deletion of the these three amino acids does not result in misfolding of the protein. AlphaFold predictions are fine since the authors used it later in the paper. Also, this would be supported by experiment(s) that shows that other RQ4 interaction(s) and/or functions are retained in $\Delta 3$ complemented cells outside of its function with MUS81.

Response: We provide evidence that the RECQ4 $\Delta 3$ variant does not result in protein misfolding, as it retains its DNA-binding ability comparable to the wild-type version of the protein (Figure S1J). To further support the functionality of this mutant, we also tested the recently reported interaction of RECQ4 with RPA32 (Papageorgiou *et al* 2023, PMID: 37875529) using co-IP experiment. Our results show that the deletion of these three amino acids does not alter the RPA32 and RECQ4 interaction (Figure 2A). Additionally, we observed a similar defect in interaction with MUS81 when these residues were substituted with alanine (Figure S2D).

3. The immunoblotting in Figure 2A are of poor quality and I recommend new immunoblots for this figure.

Response: We have repeated the experiment and included a new version of the immunoblot in the revised manuscript (Figure 2A).

4. In Figure 2C, the authors test the interaction between RQ4 and MUS81 in different phases of the cell cycle. The authors state that the "interaction was mainly in S phase and partially in G2/M phase" but the data shows interaction in G1 phase, which appears to be higher than in G2/M. Please explain.

Response: Given that we used a double thymidine block for synchronisation, which arrest the cells at the G1/S boundary, we have corrected the statement in the result section to: "the interaction between RECQ4 and MUS81 was predominantly observed at the G1 and S phases, with partial interaction observed in the G2/M phase (Figures 2C and S2E)". This adjustment more accurately reflects our synchronization and the corresponding cell cycle phases.

5. The authors show that RQ4-MUS81 functions in cancer cells that use ALT for telomere maintenance (Fig. 4) but then tries to correlate the RQ4-MUS81 function with RTS development (Fig. 6) in which the patient fibroblasts are non-cancerous. How does the ALT function associate with RTS development since ALT maintenance is not going to play a role in RTS? Does RQ4 modulate MUS81 in non-cancerous cells?

Response: We appreciate the reviewer's insightful comments. We have not been trying to directly correlate ALT and RTS development but rather to characterise the role of RECQ4-MUS81 interaction in various biological processes. Based on our results, we hypothesise that the RECQ4-MUS81 complex is required for processing replication intermediates in both ALT cells and RTS patient fibroblasts to prevent replication stress and chromosomal segregation defects. Thus, the role of RECQ4-MUS81 in non-cancerous cells extends beyond telomere maintenance and is essential for overall genome integrity. There is substantial evidence supporting the role of replication stress in both ALT and RTS (Flynn *et al* 2015, PMID: 25593184; Siitonen *et al* 2003, PMID: 12952869; Sangrithi *et al* 2005, PMID: 15960976).

6. RQ4 mutations found in RTS patients results in a loss of multiple RQ4-dependent functions; therefore, it is likely the authors are overstating in the title that the RQ4-MUS81 interactions "prevents RTS". How can the authors differentiate these functions from the other lost functions in the RQ4 mutations found in RTS patients?

Response: We acknowledge that the RECQ4 mutations found in RTS patients result in the loss of multiple RECQ4-dependent functions, and this complexity can make it challenging to attribute the prevention of RTS solely to the RECQ4-MUS81 interaction.

To address this, we have revised the title to more accurately reflect our findings and avoid overstatement. Additionally, we have included further discussion in the manuscript to clarify the distinct roles of RECQ4 and how we attempted to differentiate the specific contribution of the RECQ4-MUS81 interaction from other RECQ4-dependent functions (see page 4, lines 165-169 and page 9, lines 408-411. Our experimental design included the use of other specific interaction (i.e. with DNA and RPA) to test proficiency of the mutant affecting the interaction with MUS81 (Figures 2A and S1J).

Reviewer #2 (Remarks to the Author):

In the manuscript ‘RECQ4-MUS81 interaction safeguards hard-to-replicate regions to prevent genome instability and Rothmund-Thomson syndrome’ (NCOMMS-23-54424), Ashraf et al investigated whether the structure-specific endonuclease MUS81-EME1 interacts with and could be stimulated by RECQ4. They then assessed whether the loss of this interaction could have detrimental consequences in human cells. Although the initial *in vitro* analysis of RECQ4 and MUS81-EME1 is logical and convincing (although with some weaknesses; see below), the follow-up analysis in human cells with over expression of EGFP tagged RECQ4-WT or the RECQ4- $\Delta 3$ constructs has significant weaknesses in experimental design and interpretation of the results. Furthermore, the text needs to be improved, as the quality of the English is very variable, and is quite poor in parts.

Response: We appreciate the reviewer’s positive feedback on our *in vitro* analysis of RECQ4 and MUS81-EME1. We are grateful for the recognition of the logical and convincing nature of this aspect of our work. We hope we have addressed the identified weaknesses in the experimental design and interpretation of our follow-up analysis in human cells and improved the text to enhance the clarity and quality of the English throughout the manuscript.

Specific comments are listed below:

A, Introduction section:

- The concept of ‘replication stress’ was not properly introduced. For example, ‘difficult-to-replicate’ regions including telomeres, centromeres, and common fragile sites do not generally experience extensive replication problems until agents such as aphidicolin are used to alter their replication timing [1]. Also, it would be helpful to make the Introduction more accessible to non-specialists rather than being focused on the enzymes under study.

Response: Thank you for your comment, we have modified the introduction part to make it more general and better introduce the concept of replication stress. Please see page 2, lines 68-82.

B, Results of ‘RECQ4 stimulates MUS81 endonuclease activity’ (Figures 1, S1):

- How were the purified fragments of RECQ4 verified? There is a need for western blot or mass spectrometry analysis in the purification steps.

Response: The fragments and full-length RECQ4 used in this study were previously characterised in mapping the DNA binding region within RECQ4 (Sedlackova *et al.*, 2015; see Supplementary Figure 1B). As requested, we now provide western blot data for all fragments, except for the RECQ4 (455-1208) fragment, which was confirmed by mass spectrometry due to the RECQ4 antibody not detecting the C-terminus of RECQ4. We have included the western blot and summary of MS data in Figure S1C-D.

- Western blot analysis also needs to be provided for the RECQ1, BLM, RECQ5, and RECQ4 proteins.

Response: BLM and RECQ1 were kind gifts from Patrick Sung (Daley *et al* 2020, PMID: 32555206) and Alessandro Vindigni (Cui *et al* 2004, PMID: 15096578), respectively. We have characterised RECQ5 protein previously (Di Marco *et al* 2017, PMID: 28575661). Western blot for RECQ4 proteins is now included in Figure S1C.

- The information of the sequence and length of 3'-flap DNA substrate is missing.

Response: We apologise for omitting this information in the manuscript. We have now included the sequences of individual oligoes for the 3'-flap DNA substrate in TABLE 1.

- Because the RECQ4-353-355 motif is highly conserved, in the interaction assays these three residues should be changed to amino acids with a similar hydrophobicity (and geometry ideally), so that the 3D structure of the mutant protein is not disrupted. Full deletion of the residues is a rather crude way to analyze their significance. Therefore, the analysis with RECQ4- Δ 3 construct needs to be performed with a construct in which these three residues are substituted, but not deleted.

Response: We appreciate this insightful comment. Although this region of RECQ4 appears to be disordered, deleting three amino acids could indeed have a significant impact on protein's organisation. To address this concern and as suggested by the reviewer, we have generated a stable cell line with a construct where the three residues were substituted with alanine instead of being deleted. The analysis of this cell line showed a similar phenotype to that observed with RECQ4- Δ 3. Specifically, immunoprecipitation (IP) experiments revealed a clear defect in the interaction with MUS81 (Figure S2D) and increased levels of ultrafine bridges (UFBs) (Figure S3L).

In addition, we also provide evidence that the RECQ4 Δ 3 variant does not result in protein misfolding, as it retains its DNA-binding ability comparable to the wild-type version of the protein (Figure S1J). To further support the functionality of this mutant, we also tested the recently reported interaction of RECQ4 with RPA32 (Papageorgiou *et al* 2023, PMID: 37875529) using co-IP experiment. Our results show that the deletion of these three amino acids does not alter the RPA32 and RECQ4 interaction (Figure 2A).

C, Results of 'RECQ4 interacts with MUS81 ex vivo and safeguards the proper segregation of chromosomes' (Figures 2, S2, S3)

- To determine the timing of RECQ4 and MUS81 interaction in cell cycle, the authors synchronized cells and then performed an immunoprecipitation with GFP to pull-down MUS81 with the overexpression of RECQ4- Δ 3. The cells were apparently not well-synchronized, as judged by the data in Figure S2C. At best, there was an enrichment in the different phases. In the Methods section it was stated that: 'Cells were seeded in 100 mm dishes and subjected to a double Thymidine (2 mM thymidine, T1895, Sigma-Aldrich) block. Between the two blocks, consisting of 18 hours, the cells were released for 8 hours into fresh media. After that, an aliquot of G1/S synchronised cells was taken, or cells were released into fresh media for 6 hours further, and then cells were harvested (S phase). For G2/M and M phases, cells were treated with RO3306 (SML0569, Sigma-Aldrich) at a concentration of 9 μ M for 20 hours. After 20 hours, cells were harvested (G2/M synchronised cells) or further released in fresh media with 100 nM Nocodazole (M1404, Sigma-Aldrich) for 2 hours and then harvested by mitotic shake-off (M phase).' Because RO3306 is a CDK1 inhibitor, a 20-hour treatment of the cells would inevitably have a lot of side effects, particularly on replication origin firing in S phase. If this drug is to be used purely for synchronization of cells, it should be added for only 1-3 hours after cells have already progressed through most of S phase. Potentially, this perturbation of DNA replication timing would affect not only the interaction between RECQ4 and MUS81, but also the chromosome phenotypes analyzed in mitosis.

Response: We have repeated this experiment as suggested by the reviewer. Cells were seeded in 100 mm dishes and subjected to a double thymidine block using 2 mM thymidine (T1895, Sigma-Aldrich). The first block lasted for 18 hours, after which the cells were released into fresh media for 8 hours. Subsequently, a second thymidine block was applied for another 18 hours. Following this, an aliquot of cells synchronised in G1/S was collected. To obtain cells in the S phase, cells were released into fresh media for 6 hours before harvesting. For G2/M phase synchronisation, after allowing cells to progress through the S phase for 8 hours post-G1/S release, RO3306 (9 μ M) was added for 3 hours to arrest cells in the G2M phase, and the cells were then collected. To obtain cells in M phase, the G2/M-arrested cells were further released into fresh media containing 100 nM nocodazole for 2 hours. Mitotic cells were subsequently collected by shake-off (page 12, lines 518-527). The interaction pattern between RECQ4 and MUS81 was consistent with our previous observation and is now included in the revised version of the manuscript (Figures 2C and S2E). These artificial synchronisations were only done for the Immunoprecipitation experiment, apart from this, all other experiments (UFB and bulky bridges) were performed without any treatment.

- In Figure S3, the authors quantified mitotic events by measuring the progression time between prometaphase to anaphase in cells with or without RECQ4 depletion, and the frequency of misalignment of chromosomes in metaphase. However, it is not clear how ‘misalignment’ was quantified. Was this by the number of non-aligned chromosomes for each metaphase cell, or by the thickness of the metaphase plate? Judging by Figure S3B, the ‘proper alignment’ and ‘misalignment’ samples do not appear to be very different. The timepoint at which the misalignment was measured is also very important. How was this chosen and what was the time chosen relative to? Generally, in such analyses, time 0 would be defined by a key mitotic event – most commonly, the timing of nuclear envelope breakdown. More rigorous analysis of this phenotype is required to assess if there is a true delay in the progression of mitosis in RECQ4 depleted cells.

Response: Thank you for your insightful comment. We appreciate your feedback regarding the quantification of chromosome misalignment in Figure S3. To clarify, misalignment was assessed by examining the number of chromosomes that failed to properly align at the metaphase plate in each cell. Cells that took more than 50 minutes to progress from prometaphase to anaphase, coupled with incomplete alignment of chromosomes, were classified as having misalignment.

To address reviewer’s concern, we have removed the misalignment data to avoid confusion. We have now recalculated the mitotic transition time based on the complete disappearance of the nuclear envelope, which marks the transition from metaphase to anaphase, as shown in the revised Figure S3A-B. This approach is consistent with recent literature (Yokoyama *et al* 2019, PMID: 30718377), providing a more rigorous assessment of mitotic progression. These changes are reflected in the updated result section of the manuscript.

- In Figure 2D, because there could be more than one micronucleus in each set of binucleated cells, the quantification of micronuclei needs to be presented as the average number of micronuclei per binucleated cells to be more accurate. The same quantification should be applied to quantifications of anaphase DNA bulky bridges, and UFBs.

Response: We now provide the average number of micronuclei and bulky bridges per binucleated cell and anaphase cells, respectively (Figures 2D-E and S3C). However, we need to emphasise that most of the binucleated cells or anaphase cells which are positive for

micronuclei or bulky bridges generally showed single micronuclei or bulky bridge under the given conditions. Similarly, average number of micronuclei, bulky bridges or UFBs per binucleate / anaphase cell are now depicted in corresponding figures.

D, Results of ‘RECQ4/MUS81 interaction is essential for the processing of ultrafine bridges’ (Figure 3)

- It is well-established that while centromeric UFBs might be part of a normal mitosis and be composed of catenated double-stranded DNAs, the other type of UFBs including common fragile site associated UFBs, telomere associated UFBs, rDNA associated UFBs, or homologous recombination generated UFBs, generally arise from pathological conditions, e.g., treatment with APH, oncogene activation or DSB induction (cisplatin, camptothecin) [2-8]. Therefore, it is very surprising to see that these different types of UFBs were detected in untreated cells (Figure 3A, B). In addition, the procedure for detecting UFBs was different from most published protocols (for example [6]), and it would be helpful for the authors to state why they used this protocol.

Response: We agree that the detection of various types of ultrafine bridges in untreated cells is indeed surprising, while we are continuing to investigate this mechanism, similar observations have been reported in studies where depletion of certain genes involved in DNA metabolism such as 53BP1 (Tiwari *et al* 2018, PMID: 29445165), FIRMM (Stok *et al* 2023, PMID: 37347663), and RIF1 (Hengeveld *et al*, 2015, PMID: 26256213), led to the formation of bulky bridges and UFBs even in the absence of external replication stress. To address reviewer’s

concern, we conducted additional experiments comparing UFBs accumulation in RECQ4-depleted cells both with and without the replication stressor aphidicolin (0.4 uM for 24 hr). Our data show increased UFBs accumulation in aphidicolin-treated cells, comparable to RECQ4 knockdown cells (Figure R1). Moreover, combination of RECQ4 depletion with aphidicolin treatment showed further slight enhanced accumulation of UFBs compared to single treatments.

Figure R1: The average number of UFBs per anaphase U2OS cells treated with corresponding siRNA in the presence or absence of Aphidicolin. n=2 independent experiments.

Regarding the UFB staining protocol, we followed a previously published method (Addis Jones *et al* 2019, PMID: 31253795), which we were trained at lab (Dr. Chan's Lab - University of Sussex)

- In Figure 3C-D, it is not clear what the average number of UFBs is per cell following each treatment, and there were no IF images to illustrate the quality of the UFBs detected. It is therefore impossible to draw a firm conclusion from these data.

Response: We apologise for lack of clarity in the initial presentation. The percentage of cells with UFBs was calculated based on the presence of any UFB, irrespective of the number per

anaphase cell, classifying these cells as UFB-positive. However, as requested, we have now provided the average number of UFBs per cell for each condition (Figure 3C-G). Additionally, we have included representative IF images in Figure S3F-H to illustrate the quality and morphology of the UFBs detected. Consistent with the trends observed for micronuclei and bulky bridges, most of UFB-positive anaphase cells contained a single UFB. Notably, multiple UFBs per anaphase cell were observed primarily in conditions where defects in other resolution pathways, especially involving BLM, were present (Figure 5D and S4H).

- Even if the data presented here were correct, these data could not support the statement that ‘RECQ4/MUS81 interaction is essential for the processing of ultrafine bridges’. How do the authors define ‘processing’ here. Are the authors suggesting that these enzymes resolve bridges in some way, because this wasn’t studied as far as I could tell. Presumably, to process UFBs, these proteins would have to bind to the UFBs during mitosis (anaphase). Curiously though, it wasn’t shown that either protein is found on UFBs, and it was shown that RECQ4 and MUS81 interact mainly in S phase and not in M phase. It would be more logical to conclude that the interaction between RECQ4 and MUS81 might in some way remove DNA structures in interphase (S phase), which can lead to reduced chance of forming UFBs in mitosis.
- In summary, the data presented do not fully support the conclusions of this section.

Response: We thank the reviewer for pointing out this unclarity. We have now revised the text of the manuscript to clarify that UFBs, anaphase bridges and micronuclei are manifestations of unresolved replication intermediates. Our findings suggest that the interaction between RECQ4 and MUS81 is crucial for preventing the occurrence of UFBs and anaphase bridges, rather than directly resolving them during mitosis.

E, Results of ‘ALT-dependent MUS81 foci requires RECQ4’ (Figure 4)

- The authors used QIBC to analyze MUS81 foci in ALT or non-ALT cells with or without RECQ4 expression. It was not clear how this analysis was performed and how many cells were analyzed for each condition, as it was not described in the Methods. Only 2 ALT cell lines were analyzed, which clearly isn’t sufficient to make general conclusions about a role for RECQ4/MUS81 in telomere maintenance pathways. This analysis is too superficial.

Response: We apologise for the lack of details in our initial description of the MUS81 foci analysis. We have now revised the Material and Methods section to include a more detailed description of the experimental procedure. In addition, the number of cells analysed was now included in the corresponding figure legends. Specifically, QIBC was utilised to analyse MUS81 primarily in Flp-In T-REx U2OS cells (Figure 4E). Due to limited access to the ScanR microscope, additional experiments were conducted using a Nikon Eclipse Ti inverted fluorescence microscope, with the analysis performed using Image J software (Figure 4F, S4F).

As requested, we extended the study to LM216J cells by transiently transfecting plasmids expressing RECQ4-WT and RECQ4- Δ 3 (Figure 4F and S4F), confirming the data obtained in Flp-In T-REx U2OS cells.

Additionally, we also investigated the effect of RECQ4 knockdown on the occurrence of ALT-associated promyelocytic leukemia bodies (APBs) and complemented these experiments by expressing RECQ4-WT and RECQ4- Δ 3 in these Flp-In T-REx U2OS cells (Figure 4J-K).

We have expanded our analysis to include a broader panel of ALT-positive and ALT-negative cell lines. Using immunofluorescence, we found that RECQ4 knockdown significantly decreased MUS81 foci formation in U2OS and LM216J cell lines, but no significant effect was observed in Saos-2 cell line (Figure 4B-D and S4C), reflecting their “intermediate ALT type” (Scheel et al 2001, PMID: 11439347). This suggests that other factors or compensatory mechanisms may be at play in Saos-2 cells that mitigate the impact of RECQ4 depletion, even within the context of the ALT pathway.

These additional experiments provide a more comprehensive evaluation of the role of RECQ4-MUS81 interaction in telomere maintenance pathways.

F, Results of ‘Patient-derived RECQ4 mutation phenocopies interaction deficient mutant’ (Figures 6, S1F, S2, S5, S6)

- In Figure S5A, clearly, there is a band of a similar size to that of the normal RECQ4 protein in the RTS-CU and RTS-CA samples. It is puzzling, therefore, why the authors stated: ‘We found reduced and almost absent levels of full-length RECQ4 in RTS-CU and RTS-CA fibroblasts, respectively (Figure S5A) compared to normal fibroblasts.’

Response: We have repeated the WB of the patient's fibroblast multiple times using RECQ4 antibody but have not achieved better resolution. However, our observation is consistent with previously published WB of the same fibroblasts (Yokoyama *et al* 2019, PMID: 30718377, Figure 2A see also below). In all cases, we observe a reduction of full-length RECQ4 in RTS-CU fibroblasts (AG18373) and a more pronounced reduction in RTS-CA fibroblast (AG18371). Therefore, we now state (Page 7, lines 280-281): “We confirmed previously reported reduced levels of full-length RECQ4 in RTS-CA fibroblasts²⁵, and to some extent in RTS-CU, compared to normal fibroblasts (Figure S5A).”

Figure R2: Expression of RECQ4 from various cell lines including healthy fibroblasts (GM0023 and GM01864), and RTS fibroblasts (AG05013 and AG18371 (RTS-CA cell line used in this study)). (Taken from Yokoyama *et al* 2019, PMID: 30718377).

- It is not clear how representative of the RTS-CA mutation is in the RTS patients. This need to be introduced and discussed. If the data presented in this part were to be correct, then the title of this section should be altered to something like ‘A patient derived RECQ4 mutation phenocopies an interaction deficient mutant’.

Response: We appreciate the reviewer’s suggestion to clarify the representation of the RTS-CA mutation in Rothmund-Thomson Syndrome (RTS) patients. Similar to RTS-CA mutation,

which produces a truncated version of RECQ4 through mis-splicing, six other mutations (Table T1) have been identified in RTS patients that affects the region of amino acids 350-500. These mutations either lead to nucleotide substitution or nucleotide deletion which further leads to premature stop codon or mis-splicing (Xu *et al* 2021, PMID: 34869606), resulting in a truncation. However, we acknowledge that our specific mutation does not represent the full spectrum of RECQ4 mutations observed in RTS patients. Therefore, we have changed the heading of this section as suggested and modified the discussion to reflect this aspect (see page 6, line 271; and page 9, lines 406-411).

Table T1: RECQ4 mutations with clinical implications. (del) deletion; (>) nucleotide change from; (X) premature stop codon; (fs) frameshift.

Mutation	Effect	Mutation location	Syndrome	Cancer type	References
c.1048_1049delAG	p.Arg350fsX	N-terminus	RTS	Lymphoma	Wang et al. (2003), Suter et al. (2016), van Rij et al. (2017), Colombo et al. (2018)
c.1078C > T	p.Gln360X	N-terminus	RTS	Squamous cell carcinoma, basal cell carcinoma	Suter et al. (2016)
c.1132-2A > G	Missplicing	N-terminus	RTS	Not Known	Yadav et al. (2019)
c.1222C > T	p.Gln408X	ZnK (Zink knuckle)	RTS	Not Known	Suter et al. (2016)
c.1391-1G > A	Missplicing	NTS (Nuclear targeting signal), MTE (mitochondrial exclusion)	RTS	Osteosarcoma	Lindor et al. (2000), van Rij et al. (2017)
c.1483 + 25del11	Missplicing	SF2 helicase domain	RTS	Osteosarcoma	(Wang et al., 2002; Wang et al., 2003)

G, regarding the ‘Discussion’

- This part suffers from a lack of a summary of the key findings and many over statements or speculations. One example is: ‘We investigated the biological role of RECQ4 and its interaction with MUS81 in the resolution of replication/recombination DNA intermediates’ (lines 279-

281). This is not correct as this study has only touched upon the effect of RECQ4 depletion on the frequency of *UFBets* in mitotic cells. Another example is found on lines 297-308, in which incorrect statements were made on the MUS81 foci and their location to ‘hard to replicate’ regions, which has not been verified in this study.

Response: We thank the reviewer for the feedback on the discussion section. We have revised the discussion to include a summary of the key findings and to eliminate any overstatements or speculations. Regarding the comment on MUS81 foci, we would like to clarify that our study indeed demonstrates the colocalization of MUS81 foci with hard-to-replicate regions, specifically at telomeres, and to some extent, also centromeres. This finding has been further verified in two cell lines (Figure 4G-H). Additionally, we have also provided evidence of the effect of the MUS81-interaction deficient RECQ4 mutant on MUS81 formation in two cell lines (Figure 4E and F). We believe these results support our claims and provide solid foundation for our conclusions.

In conclusion, although the authors have described a potentially interesting interaction between RECQ4 and MUS81, the data presented here cannot support many of the conclusions stated due to inappropriate experimental design and data of limited quality. In my opinion, this manuscript is not ready for publication in Nature Communications.

Response: We appreciate the reviewer’s feedback and constructive criticism, which have been instrumental in improving our study. In response, we have conducted additional experiments to strengthen the quality of our data and provide more robust support for our conclusions. Additionally, we have clarified the text to address concerns regarding experimental design and data interpretation.

Reviewer #3 (Remarks to the Author):

In this manuscript Ashraf et al show that the RECQ4 helicase interacts with MUS81 and activates its nuclease activity and that this plays a role in chromosomal segregation. This is an interesting study that describes an unidentified role of RECQ4. Nevertheless, I have major concerns with the manuscript, and at its current state there are several issues that need to be addressed in order for this work to be published in Nature Communications.

Major comments:

1) According to Fig 1E, RECQ4 (1-400) and RECQ4 (WT) have similar effect on cleavage of the 3'-flap DNA substrate, which is close to 100% product formation. I understand that the %product formation is the % ratio of the bottom band to the sum of the top and bottom bands. If indeed this is the case, the representative images of Fig 1 do not depict that, as the RECQ4 (1-400) is clearly cleaving less of the substrate than the RECQ4 (WT) at 5, 25 and 50 nM. The same also applies for RECQ4 and RECQ5 samples in Fig 1A. The authors need to clarify what they measure as % product formation in the materials and methods and also make sure that their measurements are correct and that the representative images support their plots.

Response: We apologise for any confusion and lack of clarity. We have now included other representative images for RECQ5 in Figure 1A and RECQ4 (WT) as well as RECQ4 (1-400) in Figure 1D) in the revised version of the manuscript. We have also provided a detailed description of the quantification of % product formation in the corresponding Material and Method section.

2) I am not convinced that the statement ‘Our results revealed that the interaction between RECQ4 and MUS81 was observed mainly in the S phase and partially also in the G2/M phase (Figures 2C and S2C).’ is completely true. The reasons are:

i) The IP experiments in Figure 2A and 2C are lacking an IgG control. The authors need to repeat their IP experiments and include this.

Response: We have repeated the experiment and included a bead control, as we used anti-GFP trap beads for co-immunoprecipitation. The new data confirm our previous observations and are now included in the revised manuscript (Figures 2A and C, S2D and 6D).

ii) The MUS81 band in the G2/M phase lane of the IP:GFP is very small to support a significant interaction and the authors rightly state that there is a partial interaction. Since, the M phase lane of the IP:GFP is clearly underloaded I am wondering if it was indeed properly equally loaded, whether it would produce a MUS81 band similarly small as the G2/M phase sample. If that is true, then the ensuing conclusion would change. The authors need to produce an IP that the samples are equally loaded.

Response: We appreciate the reviewer’s comment, we have repeated the experiment to ensure equal loading and modified the statement in the results section. Our new results confirmed that the interaction between RECQ4 and MUS81 was predominantly observed in the G1 and S phases, with partial interaction observed in the G2/M phase (Figures 2C and S2E).

iii) What the authors considered to be the G1/S phase sample is mostly G1. This changes the argument that the interaction happens primarily in S phase. The authors need to rewrite this part of the text.

Response: We acknowledge the reviewer’s concern and have revised the corresponding statement in the results section. The corrected statement now reflects that the interaction was predominantly observed in the G1 and S phases, with partial interaction observed in the G2/M phase (see page 4, lines 173-175).

iv) The authors could also immunoblot for cell cycle markers such as CDT1 (specific to G1), CYCLIN A (specific to S/G2) and CENPF (specific to G2) to further support the successful synchronisation of their samples.

The authors need to address these concerns.

Response: We have repeated the synchronisation experiment as described for reviewer #2 (page 5, Figure 2C), included new FACS analysis (Figure S2E), as well as western blot for CENPF (Figure 2C) in the revised version of the manuscript.

3) It would be interesting to examine how the siRNA-RECQ4-induced micronuclei and anaphase bridges are affected by simultaneous overexpression of WT-RECQ4 and RECQ4- Δ 3.

Response: We thank the reviewer for this suggestion. In response, we extended our study using Flp-In T-REx U2OS cell line to examine the effect of expressing exogenous RECQ4- Δ 3 in the presence or absence of endogenous wild-type RECQ4, by employing siControl or siRECQ4, respectively. Our data indicate that presence of wild-type RECQ4 effectively compensate for the segregation defects observed in cell expressing only RECQ4- Δ 3, specifically in terms of the average number of bulky bridges and UFBs (Figure S5D-E). These findings are consistent

with results from fibroblasts derived from clinically unaffected individual, which expressed higher level of wild-type RECQ4 protein compared to clinically affected fibroblasts (Figure S5A-C).

4) I find the association between RECQ4 and ALT to be very weak. The authors state ‘...we also compared UFBs in the ALT-negative HT1080 cell line and observed almost no difference in fold changes for CUB and TUB (Figure 3F-G)’. This statement is completely wrong and misleading to the reader. The presented plots (Figures 3F and 3G) to support this statement clearly show that knockdown of RECQL4 by siRNA induces an increase of CUBs and TUBs that the authors show to be statistically significant. The authors need to rewrite this section of the manuscript and address the ensuing conclusions about effect of RECQ4 on cells that use ALT, that they deduce based on their wrong statement. In addition, if the authors want to explore a correlation between their RECQ4 findings and ALT status, they should test several ALT positive and telomerase positive cell lines. Using just U2OS and HT1080 is not enough to reach any legitimate conclusions.

Response: We apologize for the unclear and misleading statement regarding the association between RECQ4 and ALT, which we have now corrected in the revised manuscript. We have rewritten this section to accurately reflect the statistically significant increase in both CUBs and TUBs upon RECQ4 knockdown in the ALT-negative HT1080 cell line (see page 5, lines 203-205; Figures 3F-G and S3J-K). In addition, ALT-positive U2OS cells exhibit higher fold change for TUBs compared to CUBs and FUBs in H1080 (see page 5, lines 205-208; Figures S3D-E, and S3I-K).

Additionally, as suggested by the reviewer, we have extended our studies to include a panel of ALT-positive and ALT-negative cell lines. This comprehensive analysis is now included in Figures 4A-D, 4F, 4J-K, S4A-B, and S4F of the revised manuscript. These data provide a more robust basis for our conclusions regarding the effect of RECQ4 on cells that use ALT mechanism.

5) In order for the authors to investigate a possible role of RECQ4 and ALT phenotype they need to test the effect of RECQ4 knockdown with and without simultaneous overexpression of WT-RECQ4 and RECQ4- Δ 3 in various ALT-related phenotypes (ALT-associated PML bodies, presence of extrachromosomal telomeric DNA circles and telomeric sister chromatid exchanges) in ALT-positive as well as in telomerase-positive cell lines

Response: To address this point, we extended our study cells by transiently transfecting LM216J cells with plasmids expressing RECQ4-WT and RECQ4- Δ 3, and subsequently monitored MUS81 foci. Our findings indicate that the expression of RECQ4-WT successfully rescued the decrease in MUS81 foci observed upon RECQ4 knockdown, whereas expression of RECQ4- Δ 3 did not (Figure 4F and S4F).

Furthermore, we also investigated the effect of RECQ4 knockdown, both with and without simultaneous expression of RECQ4-WT and RECQ4- Δ 3, on the formation of ALT-associated PML bodies (APBs) (Figure 4J-K). These additional experiments, detailed in the revised manuscript, further confirm that RECQ4 depletion impacts APB formation, a defect that can be complemented by RECQ4-WT expression but not by RECQ4- Δ 3. Our new data thus support role of RECQ4-MUS81 interaction in ALT-related phenotypes. The ALT-related phenotypes are not detectable in telomerase-positive cells lines.

6) Based on the text, I deduce that in Figure 4C, RECQ4 has been deleted/knocked down. If this is indeed true, then it needs to be clearly stated in the text. If that is not the case, then I do not see proof for a correlation between RECQ4, MUS81 and ALT status.

Response: We apologise for possible unclarity. We have now explicitly stated that status of RECQ4 in these experiments. Figure 4G and the newly included Figure S4A-B demonstrate that MUS81 foci are indeed readily detected in ALT-positive cells while they are absent in ALT-negative cells. In addition, these MUS81 foci, at least in the two tested ALT-positive cell lines (U2OS and LM216J), are dependent on RECQ4 and its interaction with MUS81 (Figure 4E-F). Furthermore, we show that these foci exhibit very high colocalization with telomeres in both U2OS and LM216J cell lines (Figure 4G-I).

In addition, as a control, the authors should show that MUS81 foci can indeed successfully form in response to DSBs in the HT1080 and LM216T cell lines. In this way they can be sure that the lack of MUS81 foci in these cell lines is not due to a particular defect these cell lines might have.

Response: We have tried to induce MUS81 foci formation in both ALT-positive (U2OS, LM216J) and ALT-negative (HeLa) cells using various agents, including camptothecin, cisplatin, aphidicolin, hydroxyurea, and pyridostatin. However, none of these treatments resulted in the induction of MUS81 foci. There are indeed conflicting reports regarding MUS81 foci formation: some studies have shown an increase in MUS81 foci after replication stress (Chappidi *et al* 2020, PMID: 31759821; Noumora *et al* 2007, PMID: 17903171; Saugar *et al* 2017, PMID: 28813668), while others have reported no increase (Elbakry *et al* 2021, PMID: 33431668; Deshpande *et al* 2022, PMID: 36099913 and Panichnantakul *et al* 2021, PMID: 34139663).

7) The IP experiments in Figure 6D are lacking an IgG control. The authors need to include this.

Also, the immunoblots for GFP should be the same for all samples and not cropped for the GFP control and the WT-RECQ4-GFP, RECQ4- Δ 3-GFP and RTS-CA-GFP samples. In this way, the difference in the molecular weight heights between these samples is clearly shown.

Response: We appreciate the reviewer's comment regarding the need for appropriate control and the presentation of immunoblot data. To address this, we have repeated the IP experiment, this time including a bead-only control, as we used GFP trap beads for co-immunoprecipitation. The new data confirm our previous conclusion and are now included into the revised manuscript (Figure 6D).

Additionally, we have provided uncropped EGFP immunoblot images (Figure R3) to clearly demonstrate the differences in molecular weight between EGFP control, EGFP-RECQ4-WT, EGFP-RECQ4- Δ 3 and EGFP-RTS-CA samples. For clarity and space limitations in the main figures, we would like to continue to use cropped images, however, the uncropped images for all immunoblots will be uploaded to the publisher site and available to research community.

Figure R3. Uncropped EGFP immunoblots corresponding to Figure 6D. Immunoprecipitation (IP) from whole cell extracts expressing EGFP, EGFP-RECQ4-WT, EGFP-RECQ4-Δ3 and EGFP-RTS-CA mutants. Proteins from each cell extract (500 μg) were incubated with GFP beads for one hour and then resolved on SDS-PAGE, followed by western blotting using EGFP antibodies.

8) The authors should test the effect of the RECQ4-RTS-CA mutation on the MUS81-EME1 endonuclease activity.

Response: We have expressed and purified RECQ4-RTS-CA mutant protein and tested its effect on the MUS81-EME1 endonuclease activity. While we observe ability of this mutant to stimulate the activity of MUS81-EME1 endonuclease, however, it indeed shows reproducibly partial defect compared RECQ4-WT. The results of these experiments are now included in the revised manuscript and Figure S4G-H.

9) In order to provide clinical significance, the authors engineered U2OS cells carrying the RECQ4-RTS-CA mutation and performed several experiments to test i) the interaction between RECQ4-MUS81 (Fig 6D), ii) FUBs (Fig 6E), iii) CUBs (Fig 6F), iv) TUBs (Fig 6G) and v) MUS81 foci (Fig 6H). It would be much more informative and clinically relevant if the authors perform these experiments directly on the patient RTS-CU and 252 RTS-CA fibroblasts (and normal fibroblasts as control) that they have acquired, in the same way they did in experiments in Figures 6A-C. On top of these experiments, I would also ask the authors to assess bulky bridges in the patient-derived RTS fibroblasts.

Response: We have conducted the following experiments directly on RTS-CU and RTS-CA fibroblasts: analysis of UFBs (FUB, CUB and TUB in Figures 6A, B, and C, respectively); analysis of micronuclei (Figure S5B); and bulky bridges (Figure S5C). Unfortunately, we could not perform the co-IP experiments in fibroblast as there are no IP-compatible RECQ4 antibodies currently available on the market. Nevertheless, our results reveal increased levels of damage in fibroblast derived from a clinically affected individual (see page 7, lines 282-292). In addition, while we attempted to monitor MUS81 foci in the patient-derived fibroblasts, we were unable to detect spontaneous or DNA damage-induced Mus81 foci in these cells.

10) I am very confused with the immunoblot of Fig S2B.

In the text the authors claim: ‘...we established stable cell lines allowing simultaneous depletion of endogenous RECQ4 and expression of siRNA-resistant exogenous EGFP-tagged RECQ4-WT and RECQ4-Δ3 in Flp-In T-REx U2OS cells (Figure S2A-B)’.

And in the figure legend one reads ‘(B) Western blot confirming the expression of the various constructs EGFP, EGFP-RECQ4-WT, EGFP-RECQ4-Δ3, and EGFP1125 RECQ4-RTS-CA in Flp-In T-REx U2OS cells, alongside endogenous RECQ4 knockdown using SiRecQ4 or SiControl shown for reference.’

According to the information in Materials and Methods the engineered U2OS express pAIO plasmids that simultaneously express an shRNA against RECQ4 and siRNA-resistant RECQ4

(WT, $\Delta 3$ or RTS-CA).

This is not supported by the western blot in Figure S2B, where you can clearly see that in the WT and $\Delta 3$ lanes, the endogenous RECQ4 is present, but in the RTS-CA lane, endogenous REC4 is absent.

Also, the authors claim that on top of the expression of the shRNA against RECQ4 of the pAIO system the authors transfect the cells an siRNA against RECQ4 to achieve knockdown. How do the authors control that this siRNA will not knockdown the various RECQ4 plasmids (WT, $\Delta 3$ or RTS-CA) that are simultaneously expressed in the pAIO system that are only siRNA-resistant but not shRNA-resistant?

The authors need to clarify what do the engineered U2OS cells express, how RECQ4 knockdown is achieved and what is shown in the immunoblot of Fig S2B.

Response: The bands observed in the lanes for EGFP-RECQ4-WT and EGFP-RECQ4- $\Delta 3$ proteins in previous Figure S2B are likely degradation products of these proteins. To confirm this, we repeated the experiment using 6% SDS-PAGE gel for better separation of protein bands, followed by Western blot analysis (Figure S2C). This updated blot confirms that the endogenous RECQ4 band is absent in the siRECQ4, WT, $\Delta 3$, and RTS-CA samples, indicating successful knockdown of endogenous RECQ4. We also provide a western blot using GFP antibodies that show similar multiple bands in cell expressing exogenous RECQ4 constructs (Figure R4).

Figure R4: Western blots confirming the expression of EGFP, EGFP-RECQ4-WT, EGFP-RECQ4- $\Delta 3$, and EGFP-RECQ4-RTS-CA constructs in Flp-In T-REx U2OS cells, treated with siControl or siRECQ4 to deplete endogenous RECQ4. Whole-cell extracts were separated by SDS-PAGE and analysed by western blotting with anti-RECQ4 (A) and anti-GFP (B) antibodies.

We apologise for the lack of clarity describing the expression of engineered U2OS cells. We have modified the text in the Material and Method section to clarify the text. Briefly, upon doxycycline induction, our engineered U2OS cells express both the shRNA targeting endogenous RECQ4 and the exogenous RECQ4 constructs (WT, $\Delta 3$, and RTS-CA). The exogenous RECQ4 sequences are codon optimised to be resistant to the shRNA and siRECQ4 also used for the knockdown to ensure maximal downregulation of endogenous RECQ4.

Minor comments:

1) The authors should make sure to have at least 3 independent biological repeats in all of their experiments and also to add statistical significance tests in them.

Response: We have now included the results of triplicate experiments and their statistical significance for all remaining figures (Figures 5D, 6C).

2) Figure 5B does not provide any substantial information to support the plot in Figure 4A. The only thing the readers see are some cells in anaphase, but none type of bridges (bulky or not) are visible. The authors should provide representative magnified images of what they consider to be bulky bridges.

Response: We apologize for not providing more clear images. We have now included representative magnified images of the bulky bridges in Figure 5B to provide substantial support for the plot in Figure 5A.

REVIEWER COMMENTS

Reviewer #1 (Remarks to the Author):

In this revision entitled "RECQ4-MUS8 interaction contributes to safeguarding hard-to-replicate regions to alleviate genome instability and Rothmund-Thomson syndrome development" by Ashraf, R et al, the authors appropriately responded to my comments. My only remaining issue is the interaction of RQ4 and MUS81 shown in Fig. 2C is quite weak and the MUS81 blot is of low quality. I would only recommend a better MUS81 immunoblot. Regardless, I now recommend this paper for publication in Nature Communications.

Response: Thank you very much for recognising our efforts to address your comments. In response to your concern regarding the strength of the interaction in Figure 2C, we would like to clarify that this signal is specific to a particular cell cycle fraction, unlike the interactions shown in Figures 2A, 6D, and S2E, which represents unsynchronised cell populations across all cell cycle phases. Additionally, the requirement to isolate individual cell cycle fractions, along with a limiting number of cells in the M phase, further explains the lower intensity of the MUS81 band in Figure 2C compared to other figures.

Reviewer #2 (Remarks to the Author):

This revised manuscript is substantially improved. I am happy with most of the experimental and textual revisions. However, there are a few points, listed below, need to be addressed before this manuscript could be published:

1) The data in Figure 1A are crucial for the demonstration that RECQ4 gives the highest stimulation of MUS81-EME1 amongst the RECQ helicases. Therefore, it is important for the authors to provide evidence to show that the proteins used in this assay are equally pure and equally active as DNA helicases. The stimulation should be analyzed using an equivalent number of 'unwinding units' for each helicase. Also, the authors have given the references for the source of these proteins in the rebuttal letter, and it is clear now that the proteins used in this assay are a mixture of those obtained externally and those produced in-house. This makes it even more important to show that the purity of these proteins is similar, and that they have not been affected during transport or storage etc. In addition, the references for the source of the proteins used need to be provided in the manuscript text as well.

Response: We have previously demonstrated that helicase activity is not required for the stimulation of MUS81-EME1 (Chavdarova et al. 2015 and Di Marco et al. 2017), so we believe normalisation to unwinding units is not necessary in this context. However, as requested, we have now provided an SDS-PAGE analysis of the proteins used in these assays (Figure S1A), along with unwinding assays shown below (Figure R1), to verify their corresponding activities and ensure consistency across samples. Additionally, we have also

included references for the sources of the proteins in both the Methods section and Acknowledgments for clarity and transparency.

Figure R1: Verification of activity of proteins used in this study. **(A)** Branch migration activity of BLM. A mobile Holliday junction DNA substrate (6 nM) was incubated with the indicated amounts of BLM for 15 min at 37 °C in a buffer containing 25 mM Tris pH 7.5, 1 mM DTT, 0.1 mg/mL BSA, 50 mM KCl, 7.5 mM creatine phosphate, 11.25 µg/mL creatine kinase, 2.5 mM MgCl₂ and 2.5 mM ATP. The reaction was stopped by the addition of SDS and proteinase K. The samples were resolved in a 12% native PAGE. The gels were scanned using an image scanner FLA-9000 Starion (Fujifilm). **(B)** Helicase activity of RECQ1 assay. A Y-form DNA substrate (5 nM) was incubated with the indicated amount of RecQ1 for 20 min at 37 °C in a buffer containing 20 mM Tris pH 7.5, 5 mM DTT, 0.08 mg/mL BSA, 8% glycerol, 20 mM KCl, 5 mM MgCl₂ and 5 mM ATP. The reaction was stopped by the addition of SDS and proteinase K. The samples were resolved in a 12% native PAGE. The gels were scanned using an image scanner FLA-9000 Starion (Fujifilm). **(C)** Branch migration assay. A mobile fork DNA substrate (3 nM) was incubated with the indicated amounts of RecQ5 for 15 min at 37 °C in a buffer containing 25 mM Tris pH 7.5, 5 mM DTT, 0.1 mg/mL BSA, 30 mM KCl, 7.5 mM creatine phosphate, 11.25 µg/mL creatine kinase, 5 mM MgCl₂ and 2.5 mM ATP. The reaction was stopped by the addition of SDS and proteinase K. The samples were resolved in a 12% native PAGE. The gels were scanned using Molecular Imager PharoSFX (Bio-Rad).

2) In the current Figures 4 and S4, the authors have included a panel of ALT+ve and ATL-ve cell lines in the MUS81 focus study. Two issues with this new analysis need to be addressed: a) The references or sources, and culture conditions for all of the cell lines used need to be included in the Materials & Methods (currently, this information is only shown for some of the cell lines).

b) In Fig S4A-B, the IF staining of MUS81 analysis has a very strong background, and it is actually very difficult to separate the foci from the background. We have to take on faith that the foci amongst the mass of background staining are really the MUS81 foci? More strangely, the MUS81 staining in HT1080 cells was mostly outside of the nucleus. There is a concern that this IF did not work well in this assay. For example, the MUS81 IF images in Figure S4F look quite clear and distinct. The authors need to address the discrepancies between different experiments and provide quantification of MUS81 foci in all the cell lines analyzed in Fig S4A-B. In addition, in Figure S4A, a quantification of MUS81 foci in all the cell lines analyzed need to be provided.

Response:

- a) We have now included sources and culture conditions for all cell lines in the Materials and Methods section.
- b) We apologise for the initial quality of the IF images in Figure S4A-B. The observed differences in clarity between these images and those from Figure S4F (now Figure S4G) were due to the use of plastic plates for microscopy in Figure S4A-B, compared to glass coverslips in Figure S4F (now Figure S4G). To address this, we repeated the experiment on glass coverslips, which significantly improved the image quality for both the representative images (Figure S4A-B) and the quantitative analysis (Figure 4A). Additionally, we provided the validation of the MUS81 foci specificity by performing experiments with siRNA against MUS81, confirming that the observed foci represent MUS81 localization (Figure R2).

Figure R2: Representative immunofluorescence images of MUS81 foci in U2OS cells treated with siControl and siMUS81 for 48 hours.

3) The data shown in this manuscript support the notion that RECQ4-MUS81 interaction contributes to telomere maintenance but are not so strong with regard to ‘other hard-to-replicate regions’. Also, it is an overstatement to say that this interaction contributes to ‘Rothmund-Thomson syndrome development’. I would suggest change the title to ‘RECQ4-MUS81 interaction contributes to telomere maintenance in human cells’, so that it reflects more precisely on the content and conclusion drawn from the data. This means the Abstract and Discussion needs to be modified somewhat also.

Response: We have modified the title and soften the direct contribution to RTS “RECQ4-MUS81 interaction contributes to telomere maintenance with implications to Rothmund-

Thomson syndrome". We have also toned down these aspects in the revised manuscript to limit any overstatements.

Minor point: the table of a list of the mutation found in Rothmund-Thomson syndrome Patients are very helpful for the readers to have a sense of the scale of the mutation spectrum of RECQ4 in these patients. This table could be included in the Supplementary data in the manuscript.

Response: Thank you for the suggestion. We have now included a table summarizing the mutations found in Rothmund-Thomson syndrome patients in the Supplementary Data (Figure S2A) to provide readers with a comprehensive overview of the RECQ4 mutation spectrum in these cases.

Reviewer #3 (Remarks to the Author):

This revised version is a great improvement from the initial submission. Nevertheless, there are a few of my initial concerns that have not been addressed adequately, in order to recommend this revised manuscript for publication in Nature Communications.

Response: Thank you for your positive feedback on the revised manuscript. We appreciate your continued insights, and we have carefully addressed the remaining concerns to ensure the manuscript meets the required standards for publication in *Nature Communications*.

Major comments:

1) According to Fig 1E, RECQ4 (1-400) and RECQ4 (WT) have similar effect on cleavage of the 3'-flap DNA substrate, which is close to 100% product formation. I understand that the %product formation is the % ratio of the bottom band to the sum of the top and bottom bands. If indeed this is the case, the representative images of Fig 1 do not depict that, as the RECQ4 (1-400) is clearly cleaving less of the substrate than the RECQ4 (WT) at 5, 25 and 50 nM. The same also applies for RECQ4 and RECQ5 samples in Fig 1A. The authors need to clarify what they measure as % product formation in the materials and methods and also make sure that their measurements are correct and that the representative images support their plots.

Response: We apologise for any confusion and lack of clarity. We have now included other representative images for RECQ5 in Figure 1A and RECQ4 (WT) as well as RECQ4 (1-400) in Figure 1D) in the revised version of the manuscript. We have also provided a detailed description of the quantification of % product formation in the corresponding Material and Method section.

I thank the authors for addressing my comment.

2) I am not convinced that the statement 'Our results revealed that the interaction between RECQ4 and MUS81 was observed mainly in the S phase and partially also in the G2/M phase (Figures 2C and S2C).' is completely true. The reasons are: i) The IP experiments in Figure 2A and 2C are lacking an IgG control. The authors need to repeat their IP experiments and include this.

Response: We have repeated the experiment and included a bead control, as we used anti-

GFP trap beads for co-immunoprecipitation. The new data confirm our previous observations and are now included in the revised manuscript (Figures 2A and C, S2D and 6D).

I thank the authors for addressing my comment.

ii) The MUS81 band in the G2/M phase lane of the IP:GFP is very small to support a significant interaction and the authors rightly state that there is a partial interaction. Since, the M phase lane of the IP:GFP is clearly underloaded I am wondering if it was indeed properly equally loaded, whether it would produce a MUS81 band similarly small as the G2/M phase sample. If that is true, then the ensuing conclusion would change. The authors need to produce an IP that the samples are equally loaded.

Response: We appreciate the reviewer's comment, we have repeated the experiment to ensure equal loading and modified the statement in the results section. Our new results confirmed that the interaction between RECQ4 and MUS81 was predominantly observed in the G1 and S phases, with partial interaction observed in the G2/M phase (Figures 2C and S2E).

I thank the authors for addressing my comment.

iii) What the authors considered to be the G1/S phase sample is mostly G1. This changes the argument that the interaction happens primarily in S phase. The authors need to rewrite this part of the text.

Response: We acknowledge the reviewer's concern and have revised the corresponding statement in the results section. The corrected statement now reflects that the interaction was predominantly observed in the G1 and S phases, with partial interaction observed in the G2/M phase (see page 4, lines 173-175).

I thank the authors for addressing my comment.

iv) The authors could also immunoblot for cell cycle markers such as CDT1 (specific to G1), CYCLIN A (specific to S/G2) and CENPF (specific to G2) to further support the successful synchronisation of their samples. The authors need to address these concerns.

Response: We have repeated the synchronisation experiment as described for reviewer #2 (page 5, Figure 2C), included new FACS analysis (Figure S2E), as well as western blot for CENPF (Figure 2C) in the revised version of the manuscript.

I thank the authors for addressing my comment.

3) It would be interesting to examine how the siRNA-RECQ4-induced micronuclei and anaphase bridges are affected by simultaneous overexpression of WT-RECQ4 and RECQ4- Δ 3.

Response: We thank the reviewer for this suggestion. In response, we extended our study using Flp-In T-REx U2OS cell line to examine the effect of expressing exogenous RECQ4- Δ 3 in the presence or absence of endogenous wild-type RECQ4, by employing siControl or siRECQ4, respectively. Our data indicate that presence of wild-type RECQ4 effectively compensate for the segregation defects observed in cell expressing only RECQ4- Δ 3, specifically in terms of the average number of bulky bridges and UFBs (Figure S5D-E). These

findings are consistent with results from fibroblasts derived from clinically unaffected individual, which expressed higher level of wild-type RECQ4 protein compared to clinically affected fibroblasts (Figure S5A-C).

The authors should make it clearer that in the plots in Figures S5D-E, that the grey bars are representing cells overexpressing the wild-type RECQ4 and the brown ones are representing cells overexpressing RECQ4- Δ 3. The way the plot is depicted and the explanation in the figure legend is confusing.

Response: Thank you for highlighting this point. We apologise for the confusion caused by the previous figure legend. We have now corrected and clarified the corresponding figure legend (Figures S5E-F in the revised manuscript) to accurately describe the status of RECQ4 in the grey, black, and brown bars. In this revised legend, we specify that the grey bars represent cells expressing endogenous RECQ4 and overexpression of control EGFP, the black bars represent cells with endogenous RECQ4 depleted and overexpression of control EGFP, and the brown bars depict cells with endogenous RECQ4 and simultaneous overexpression of EGFP-RECQ4- Δ 3. This should provide a clearer interpretation of the data.

4) I find the association between RECQ4 and ALT to be very weak. The authors state ‘...we also compared UFBs in the ALT-negative HT1080 cell line and observed almost no difference in fold changes for CUB and TUB (Figure 3F-G)’. This statement is completely wrong and misleading to the reader. The presented plots (Figures 3F and 3G) to support this statement clearly show that knockdown of RECQL4 by siRNA induces an increase of CUBs and TUBs that the authors show to be statistically significant. The authors need to rewrite this section of the manuscript and address the ensuing conclusions about effect of RECQ4 on cells that use ALT, that they deduce based on their wrong statement. In addition, if the authors want to explore a correlation between their RECQ4 findings and ALT status, they should test several ALT positive and telomerase positive cell lines. Using just U2OS and HT1080 is not enough to reach any legitimate conclusions.

Response: We apologize for the unclear and misleading statement regarding the association between RECQ4 and ALT, which we have now corrected in the revised manuscript. We have rewritten this section to accurately reflect the statistically significant increase in both CUBs and TUBs upon RECQ4 knockdown in the ALT-negative HT1080 cell line (see page 5, lines 203-205; Figures 3F-G and S3J-K). In addition, ALT-positive U2OS cells exhibit higher fold change for TUBs compared to CUBs and FUBs in H1080 (see page 5, lines 205-208; Figures S3D-E, and S3I-K).

Additionally, as suggested by the reviewer, we have extended our studies to include a panel of ALT-positive and ALT-negative cell lines. This comprehensive analysis is now included in Figures 4A-D, 4F, 4J-K, S4A-B, and S4F of the revised manuscript. These data provide a more robust basis for our conclusions regarding the effect of RECQ4 on cells that use ALT mechanism.

The authors should provide a western blot of MUS81 for the various ALT+ and telomerase positive cell lines they use in Figure 4A.

The data in Figure 4A must be explained more thoroughly. The authors state that MUS81 foci were completely absent from the telomerase+ cell lines, but this needs to be shown

with some sort of analysis. Therefore, the authors should provide MUS81 foci quantification analysis data for the data presented in Figure 4A.

Response: We have included a western blot assessing levels of MUS81 protein in tested cell lines (Figure S4C) and provided quantification of MUS81 foci (Figure 4A).

5) In order for the authors to investigate a possible role of RECQ4 and ALT phenotype they need to test the effect of RECQ4 knockdown with and without simultaneous overexpression of WT-RECQ4 and RECQ4- Δ 3 in various ALT-related phenotypes (ALT-associated PML bodies, presence of extrachromosomal telomeric DNA circles and telomeric sister chromatid exchanges) in ALT-positive as well as in telomerase-positive cell lines

Response: To address this point, we extended our study cells by transiently transfecting LM216J cells with plasmids expressing RECQ4-WT and RECQ4- Δ 3, and subsequently monitored MUS81 foci. Our findings indicate that the expression of RECQ4-WT successfully rescued the decrease in MUS81 foci observed upon RECQ4 knockdown, whereas expression of RECQ4- Δ 3 did not (Figure 4F and S4F).

Furthermore, we also investigated the effect of RECQ4 knockdown, both with and without simultaneous expression of RECQ4-WT and RECQ4- Δ 3, on the formation of ALT-associated PML bodies (APBs) (Figure 4J-K). These additional experiments, detailed in the revised manuscript, further confirm that RECQ4 depletion impacts APB formation, a defect that can be complemented by RECQ4-WT expression but not by RECQ4- Δ 3. Our new data thus support role of RECQ4-MUS81 interaction in ALT-related phenotypes. The ALT-related phenotypes are not detectable in telomerase-positive cells lines.

Since the authors only assessed APBs and not the other ALT-related phenotypes I suggested (presence of extrachromosomal telomeric DNA circles and telomeric sister chromatid exchanges) in only one ALT+ cell line and none telomerase+ cell lines (which would have been the negative control), I suggest they significantly tone down any claims about an RECQ4 having an effect on ALT. Without the experiments I suggested, any definitive association of RECQ4 and ALT is farfetched and misleading.

Response: Given the time frame for this revision, completing additional ALT-related phenotypes with sufficient quality and replications for publication was not feasible. However, we did observe RECQ4-dependent reduction of c-circles in USO2 cells. Nevertheless, we have toned down our claims about the association of the RECQ4-MUS81 interaction with ALT status in the discussion section.

6) Based on the text, I deduce that in Figure 4C, RECQ4 has been deleted/knocked down. If this is indeed true, then it needs to be clearly stated in the text. If that is not the case, then I do not see proof for a correlation between RECQ4, MUS81 and ALT status.

Response: We apologise for possible unclarity. We have now explicitly stated that status of RECQ4 in these experiments. Figure 4G and the newly included Figure S4A-B demonstrate that MUS81 foci are indeed readily detected in ALT-positive cells while they are absent in ALT-negative cells. In addition, these MUS81 foci, at least in the two tested ALT-positive cell lines (U2OS and LM216J), are dependent on RECQ4 and its interaction with MUS81 (Figure 4E-F). Furthermore, we show that these foci exhibit very high colocalization with telomeres

in both U2OS and LM216J cell lines (Figure 4G-I).

As mentioned in a previous comment, the authors should provide a western blot of MUS81, as well as a quantification analysis of MUS81 foci for the various ALT+ and telomerase positive cell lines they use in Figure 4A.

Response: We have included a western blot assessing levels of MUS81 protein in tested cell lines (Figure S4C) and provided quantification of MUS81 foci (Figure 4A).

In addition, as a control, the authors should show that MUS81 foci can indeed successfully form in response to DSBs in the HT1080 and LM216T cell lines. In this way they can be sure that the lack of MUS81 foci in these cell lines is not due to a particular defect these cell lines might have.

Response: We have tried to induce MUS81 foci formation in both ALT-positive (U2OS, LM216J) and ALT-negative (HeLa) cells using various agents, including camptothecin, cisplatin, aphidicolin, hydroxyurea, and pyridostatin. However, none of these treatments resulted in the induction of MUS81 foci. There are indeed conflicting reports regarding MUS81 foci formation: some studies have shown an increase in MUS81 foci after replication stress (Chappidi et al 2020, PMID: 31759821; Noumora et al 2007, PMID: 17903171; Saugar et al 2017, PMID: 28813668), while others have reported no increase (Elbakry et al 2021, PMID: 33431668; Deshpande et al 2022, PMID: 36099913 and Panichnantakul et al 2021, PMID: 34139663).

This again makes it important that the authors assess MUS81 levels in these cell lines. Since the authors rely their conclusions on MUS81 foci it is imperative that MUS81 controls have been assessed before making any claims.

Response: We have included a western blot assessing levels of MUS81 protein in tested cell lines (Figure S4C) and provided quantification of MUS81 foci (Figure 4A).

7) The IP experiments in Figure 6D are lacking an IgG control. The authors need to include this. Also, the immunoblots for GFP should be the same for all samples and not cropped for the GFP control and the WT-RECQ4-GFP, RECQ4- Δ 3-GFP and RTS-CA-GFP samples. In this way, the difference in the molecular weight heights between these samples is clearly shown.

Response: We appreciate the reviewer's comment regarding the need for appropriate control and the presentation of immunoblot data. To address this, we have repeated the IP experiment, this time including a bead-only control, as we used GFP trap beads for co-immunoprecipitation. The new data confirm our previous conclusion and are now included into the revised manuscript (Figure 6D).

I thank the authors for addressing my comment.

Additionally, we have provided uncropped EGFP immunoblot images (Figure R3) to clearly demonstrate the differences in molecular weight between EGFP control, EGFP-RECQ4-WT, EGFP-RECQ4- Δ 3 and EGFP-RTS-CA samples. For clarity and space limitations in the main figures, we would like to continue to use cropped images, however, the uncropped images for all immunoblots will be uploaded to the publisher site and available to research

community.

If the authors want to continue using the cropped GFP western blots, they should at least indicate on the side of the cropped blot what is depicted when they write 'EGFP', otherwise the reader will be confused with two different cropped blots being described as 'EGFP'.

Also, I would kindly ask the authors to include the uncropped GFP western blots in the supplementary data of the manuscript.

Response: We have clarified the labels on the western blots by specifying the individual RECQ4 forms instead of using "EGFP". Additionally, as requested, we have included the uncropped GFP western blots in the corresponding supplementary data (Figure S2D and S2F-G).

8) The authors should test the effect of the RECQ4-RTS-CA mutation on the MUS81-EME1 endonuclease activity.

Response: We have expressed and purified RECQ4-RTS-CA mutant protein and tested its effect on the MUS81-EME1 endonuclease activity. While we observe ability of this mutant to stimulate the activity of MUS81-EME1 endonuclease, however, it indeed shows reproducibly partial defect compared RECQ4-WT. The results of these experiments are now included in the revised manuscript and Figure S4G-H.

I thank the authors for addressing my comment.

9) In order to provide clinical significance, the authors engineered U2OS cells carrying the RECQ4-RTS-CA mutation and performed several experiments to test i) the interaction between RECQ4-MUS81 (Fig 6D), ii) FUBs (Fig 6E), iii) CUBs (Fig 6F), iv) TUBs (Fig 6G) and v) MUS81 foci (Fig 6H). It would be much more informative and clinically relevant if the authors perform these experiments directly on the patient RTS-CU and 252 RTS-CA fibroblasts (and normal fibroblasts as control) that they have acquired, in the same way they did in experiments in Figures 6A-C. On top of these experiments, I would also ask the authors to assess bulky bridges in the patient-derived RTS fibroblasts.

Response: We have conducted the following experiments directly on RTS-CU and RTS-CA fibroblasts: analysis of UFBs (FUB, CUB and TUB in Figures 6A, B, and C, respectively); analysis of micronuclei (Figure S5B); and bulky bridges (Figure S5C). Unfortunately, we could not perform the co-IP experiments in fibroblast as there are no IP-compatible RECQ4 antibodies currently available on the market. Nevertheless, our results reveal increased levels of damage in fibroblast derived from a clinically affected individual (see page 7, lines 282-292). In addition, while we attempted to monitor MUS81 foci in the patient-derived fibroblasts, we were unable to detect spontaneous or DNA damage-induced Mus81 foci in these cells.

I thank the authors for addressing my comment.

10) I am very confused with the immunoblot of Fig S2B. In the text the authors claim: '...we established stable cell lines allowing simultaneous depletion of endogenous RECQ4 and expression of siRNA-resistant exogenous EGFP-tagged RECQ4-WT and RECQ4- Δ 3 in Flp-In T-REx U2OS cells (Figure S2A-B)'. And in the figure legend one reads '(B) Western blot confirming the expression of the various constructs EGFP, EGFP-RECQ4-WT, EGFP-RECQ4- Δ 3,

and EGFP1125 RECQ4-RTS-CA in Flp-In T-REx U2OS cells, alongside endogenous RECQ4 knockdown using SiRecQ4 or SiControl shown for reference. According to the information in Materials and Methods the engineered U2OS express pAIO plasmids that simultaneously express an shRNA against RECQ4 and siRNA-resistant RECQ4 (WT, $\Delta 3$ or RTS-CA). This is not supported by the western blot in Figure S2B, where you can clearly see that in the WT and $\Delta 3$ lanes, the endogenous RECQ4 is present, but in the RTS-CA lane, endogenous REC4 is absent. Also, the authors claim that on top of the expression of the shRNA against RECQ4 of the pAIO system the authors transfect the cells an siRNA against RECQ4 to achieve knockdown. How do the authors control that this siRNA will not knockdown the various RECQ4 plasmids (WT, $\Delta 3$ or RTS-CA) that are simultaneously expressed in the pAIO system that are only siRNA-resistant but not shRNA-resistant? The authors need to clarify what do the engineered U2OS cells express, how RECQ4 knockdown is achieved and what is shown in the immunoblot of Fig S2B.

Response: The bands observed in the lanes for EGFP-RECQ4-WT and EGFP-RECQ4- $\Delta 3$ proteins in previous Figure S2B are likely degradation products of these proteins. To confirm this, we repeated the experiment using 6% SDS-PAGE gel for better separation of protein bands, followed by Western blot analysis (Figure S2C). This updated blot confirms that the endogenous RECQ4 band is absent in the siRECQ4, WT, $\Delta 3$, and RTS-CA samples, indicating successful knockdown of endogenous RECQ4. We also provide a western blot using GFP antibodies that show similar multiple bands in cell expressing exogenous RECQ4 constructs (Figure R4).

We apologise for the lack of clarity describing the expression of engineered U2OS cells. We have modified the text in the Material and Method section to clarify the text. Briefly, upon doxycycline induction, our engineered U2OS cells express both the shRNA targeting endogenous RECQ4 and the exogenous RECQ4 constructs (WT, $\Delta 3$, and RTS-CA). The exogenous RECQ4 sequences are codon optimised to be resistant to the shRNA and siRECQ4 also used for the knockdown to ensure maximal downregulation of endogenous RECQ4.

I thank the authors for addressing my comment.

Minor comments: 1) The authors should make sure to have at least 3 independent biological repeats in all of their experiments and also to add statistical significance tests in them.

Response: We have now included the results of triplicate experiments and their statistical significance for all remaining figures (Figures 5D, 6C).

I thank the authors for addressing my comment.

2) Figure 5B does not provide any substantial information to support the plot in Figure 4A. The only thing the readers see are some cells in anaphase, but none type of bridges (bulky or not) are visible. The authors should provide representative magnified images of what they consider to be bulky bridges.

Response: We apologize for not providing more clear images. We have now included representative magnified images of the bulky bridges in Figure 5B to provide substantial support for the plot in Figure 5A.

I thank the authors for addressing my comment. I would kindly ask that the new provided images are presented at a higher magnification.

Response: We have now provided new images along with magnified examples of anaphase cells with and without the bulky bridges.

REVIEWER COMMENTS

Reviewer #2 (Remarks to the Author):

This 2nd revised manuscript by Ashraf R. et al has addressed all the three points raised last time.

I am happy with the content and quality of the figures provided, namely Figure S1A, Figure R1, Figure R2 and Figure S5A. I am also happy with the changes in the Title, Abstract, Material/methods and Discussion about following my comment.

The remaining minor modifications I would like to ask for before its publication in Nature Communication are: to include Figures R1 and R2 in the manuscript. This is because, regarding Figure R1, it is essential to show that the proteins used in this study has equal helicase activity and not just by their action in previous studies; regarding Figure R2, it would be an excellent addition to show MUS81 foci changes with siRNA MUS81 KD in a cell line with zoomed images. Both figures will bring more credibility to this manuscript. Please note, after including these two figures in the MS, the text regarding figure orders, Materials/methods, etc will need to be changed accordingly. I am happy to read a final version again.

Response: Thank you very much for recognising our efforts to address your comments. As per your request, we have of incorporated Figures R1 and R2 into the manuscript. Specifically:

- The content of Figure R1 has been included as part of Supplementary Figure 1B-D.
- The content of Figure R2 has been included as part of Supplementary Figure 4A.
- We have also updated the figure numbering, figure legends, and any relevant text in the manuscript accordingly.

Reviewer #3 (Remarks to the Author):

This revised version is a great improvement from the previous submission. My concerns have been met and I am happy to recommend this manuscript for publication in Nature Communications.

Response: We would like to thank you for recognising our efforts to address your concerns.